# Estimating the potential to prevent locally acquired HIV infections in a UNAIDS Fast-Track City, Amsterdam

**Alexandra Blenkinsop[1,2]\*, Mélodie Monod[1], Ard van Sighem[3], Nikos Pantazis[4], Daniela Bezemer[3], Eline Op de Coul[5], Thijs van de Laar[6,7], Christophe Fraser[8], Maria Prins[9], Peter Reiss[2,10], Godelieve J de Bree[2,11], Oliver Ratmann[1]\*, On behalf of HIV Transmission Elimination AMsterdam (H-TEAM) collaboration**

[1]Department of Mathematics, Imperial College London, London, United Kingdom; [2]Amsterdam Institute for Global Health and Development, Amsterdam, Netherlands; [3]Stichting HIV Monitoring, Amsterdam, Netherlands; [4]Department of Hygiene, Epidemiology and Medical Statistics, University of Athens, Athens, Greece; [5]Center for Infectious Diseases Prevention and Control, National Institute for Public Health and the Environment (RIVM), Bilthoven, Netherlands; [6]Department of Donor Medicine Research, Sanquin, Amsterdam, Netherlands; [7]Department of Medical Microbiology, Onze Lieve Vrouwe Gasthuis, Amsterdam, Netherlands; [8]Big Data Institute, Nuffield Department of Medicine, University of Oxford, Oxford, United Kingdom; [9]Academic Medical Center, Amsterdam, Netherlands; [10]Department of Global Health, Amsterdam University Medical Centers, Amsterdam, Netherlands; [11]Division of Infectious Diseases, Department of Internal Medicine, Amsterdam Infection and Immunity Institute, Amsterdam, Netherlands

**\*For correspondence:**
a.blenkinsop@imperial.ac.uk (AB);
oliver.ratmann05@imperial.ac.uk (OR)

## Abstract

**Background:** More than 300 cities including the city of Amsterdam in the Netherlands have joined the UNAIDS Fast-Track Cities initiative, committing to accelerate their HIV response and end the AIDS epidemic in cities by 2030. To support this commitment, we aimed to estimate the number and proportion of Amsterdam HIV infections that originated within the city, from Amsterdam residents. We also aimed to estimate the proportion of recent HIV infections during the 5-year period 2014–2018 in Amsterdam that remained undiagnosed.

**Methods:** We located diagnosed HIV infections in Amsterdam using postcode data (PC4) at time of registration in the ATHENA observational HIV cohort, and used HIV sequence data to reconstruct phylogeographically distinct, partially observed Amsterdam transmission chains. Individual-level infection times were estimated from biomarker data, and used to date the phylogenetically observed transmission chains as well as to estimate undiagnosed proportions among recent infections. A Bayesian Negative Binomial branching process model was used to estimate the number, size, and growth of the unobserved Amsterdam transmission chains from the partially observed phylogenetic data.

**Results:** Between 1 January 2014 and 1 May 2019, there were 846 HIV diagnoses in Amsterdam residents, of whom 516 (61%) were estimated to have been infected in 2014–2018. The rate of new Amsterdam diagnoses since 2014 (104 per 100,000) remained higher than the national rates excluding Amsterdam (24 per 100,000), and in this sense Amsterdam remained a HIV hotspot in the Netherlands. An estimated 14% [12–16%] of infections in Amsterdan MSM in 2014–2018 remained undiagnosed by 1 May 2019, and 41% [35–48%] in Amsterdam heterosexuals, with variation by region of birth. An estimated 67% [60–74%] of Amsterdam MSM infections in 2014–2018 had an

Amsterdam resident as source, and 56% [41–70%] in Amsterdam heterosexuals, with heterogeneity by region of birth. Of the locally acquired infections, an estimated 43% [37–49%] were in foreign-born MSM, 41% [35–47%] in Dutch-born MSM, 10% [6–18%] in foreign-born heterosexuals, and 5% [2–9%] in Dutch-born heterosexuals. We estimate the majority of Amsterdam MSM infections in 2014–2018 originated in transmission chains that pre-existed by 2014.

**Conclusions:** This combined phylogenetic, epidemiologic, and modelling analysis in the UNAIDS Fast-Track City Amsterdam indicates that there remains considerable potential to prevent HIV infections among Amsterdam residents through city-level interventions. The burden of locally acquired infection remains concentrated in MSM, and both Dutch-born and foreign-born MSM would likely benefit most from intensified city-level interventions.

**Funding:** This study received funding as part of the H-TEAM initiative from Aidsfonds (project number P29701). The H-TEAM initiative is being supported by Aidsfonds (grant number: 2013169, P29701, P60803), Stichting Amsterdam Dinner Foundation, Bristol-Myers Squibb International Corp. (study number: AI424-541), Gilead Sciences Europe Ltd (grant number: PA-HIV-PREP-16-0024), Gilead Sciences (protocol numbers: CO-NL-276-4222, CO-US-276-1712, CO-NL-985-6195), and M.A.C AIDS Fund.

## Editor's evaluation

Congratulations on this impressive paper which combines clinical biomarker data, patient specific data and viral genetics data to estimate the proportion of HIV infections occurring within key subgroups of the population in Amsterdam. The work is methodologically impressive and also may be of high utility for understanding the spread of HIV and other viral infections through the population.

## Introduction

Human immunodeficiency virus (HIV) is concentrated in metropolitan areas (*Joint United Nations Programme on HIV/AIDS, 2014*). In response, as of March 2021 over 300 cities have joined the Fast-Track Cities initiative (www.fast-trackcities.org) by signing the Paris Declaration, committing to end the AIDS epidemic by 2030, by addressing disparities in access to basic health and social services, social justice and economic opportunities (*UNAIDS, 2019*). Several of these fast-track cities have successfully developed strategies which best address the needs of the local epidemic, including London's HIV Prevention Programme and early ART initiation, and New York's Status Neutral Prevention and Treatment Cycle (*Public Health England, 2018*; *Myers et al., 2018*). A central milestone in this agenda is to characterise the number of HIV infections that are acquired from sources within cities and are thus preventable through local interventions, as well as to identify the primary risk groups with infections from local sources.

In the Netherlands, Amsterdam is the city with the greatest HIV burden nationally, reflecting in part large communities of MSM and foreign-born individuals. Amsterdam has a long history of a collaborative HIV approach in combating the epidemic and joined the UNAIDS Fast-Track Cities initiative on 1 December 2014. City-level HIV responses were galvanised in the HIV Transmission Elimination Amsterdam project (H-Team) that same year (*de Bree et al., 2019*). The H-Team fast-track response, amongst others, focussed on outreach activities, encouraging repeat testing every 3–6 months to identify acute and early HIV infection, followed by immediate initiation of combination antiretroviral therapy (c-ART) in newly diagnosed patients, and roll-out of pre-exposure prophylaxis (PrEP) in populations at increased risk of HIV infection (*den Daas et al., 2018*; *Bartelsman et al., 2017*; *Hoornenborg et al., 2019*; *Dijkstra et al., 2019*). Prior to the COVID-19 pandemic, the number of annual HIV diagnoses in Amsterdam residents has consistently declined from ~300 new city-level HIV diagnoses in 2010 to ~120 in 2018, primarily in Dutch-born and foreign-born MSM. Given these achievements, it is now unclear how many of the remaining new infections are locally acquired and could thus still be locally averted. Late diagnoses remain common and are a particular concern in this effort, both for individual health and the risk that unnoticed transmission chains pose to public health.

Here, we build on Amsterdam's combined case and genomic surveillance data to reconstruct transmission chains at city level, defined as a single introduction of HIV into Amsterdam residents, followed

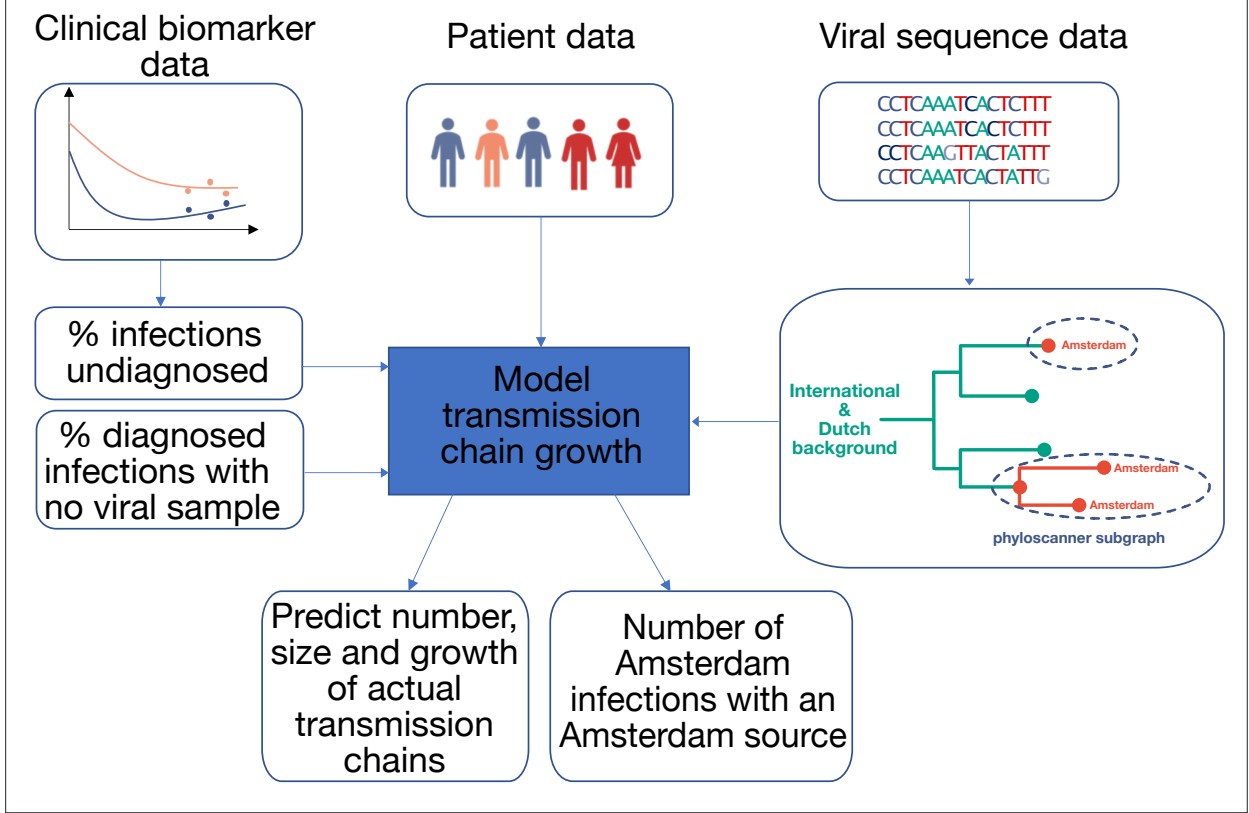

**Figure 1.** Approach to analysis. Input data includes patient baseline data at registration, clinical biomarker data and viral sequence data. Biomarker data is used to estimate infection times, the proportion of undiagnosed infections, and thus the total population size of people living with HIV. HIV sequence data is used to reconstruct phylogenetic trees. Groups of Amsterdam residents with distinct virus are determined phylogeographically with phyloscanner, and without considering genetic distances or bootstrap support. Each such group of Amsterdam residents with distinct virus is interpreted as the partially observed part of a distinct transmission chain among Amsterdam residents, and analysed in calendar time based on the infection times estimated from individual biomarker data, as well as clinical data on viral suppression. The partial observations are used to infer the number, size and growth of the actual transmission chains among Amsterdam residents, and derive key epidemic quantities of interest.

by a direct infection chain among Amsterdam residents (*Figure 1*). We exploit clinical patient data to estimate times of HIV infection at individual level, which provides crucial temporal information for interpreting the observed transmission chains. This allows us to estimate the extent of undiagnosed infections at the forefront of the cities' transmission chains, among infections that are estimated to have occured since Amsterdam joined the Fast-Track Cities network in 2014. We then characterise the growth and origins of Amsterdam transmission chains in 2014–2018, and quantify in particular the proportion of Amsterdam infections in this time period that had an Amsterdam resident as source, and could have been locally averted.

## Materials and methods
### Demographic and clinical cohort data comprising city-level infections

Data were obtained from the prospective ATHENA cohort of all people living with HIV (PLHIV) in care in the Netherlands, including patient demographics and longitudinal CD4, HIV viral load, viral sequence, and treatment data (see Appendix 1, Section 2) (*Boender et al., 2018*). Sequencing methods are described previously (*Bezemer et al., 2004*). Cohort data are near complete in the sense that 2% of individuals opted out of participating in the ATHENA study, and 5.2% of individuals who entered ATHENA were lost to follow-up (*Boender et al., 2018*; *Sighem et al., 2020*). We geolocated diagnosed infections to Amsterdam based on patients' postcode of residence at time of first registration in ATHENA or the most recent registration update, which includes PLHIV that changed residence to

Amsterdam at a registration update (4%), PLHIV that changed residence to another Dutch municipality after first registration (4%), and PLHIV that were consistently resident in Amsterdam (92%).

Participants were stratified by region of birth: MSM (The Netherlands; Western Europe, North America, Oceania; Eastern and Central Europe; South America and the Caribbean; Other), and heterosexual individuals (The Netherlands; South America and the Caribbean; Sub-Saharan Africa; Other), resulting in 9 risk groups in total. Throughout, we denote transmission group (Amsterdam MSM or heterosexuals) by $t$, and geographic region of birth by $r$.

We here focus on city-level transmission chains growing in the period from 1 January 2014 to 31 December 2018, which for brevity we refer to as 2014–2018. Available demographic, clinical, and viral sequence data were obtained for HIV diagnoses in Amsterdam from the ATHENA database version closed on 1 May 2019.

## Estimating HIV infection dates and undiagnosed infections

Using longitudinal viral load and CD4 count data and further demographic and clinical information, we estimated time from infection to diagnosis for all HIV diagnosed patients with a Bayesian approach (*Pantazis et al., 2019*). Briefly, data from the CASCADE collaboration on 19,788 observed HIV seroconverters were used to parameterize a bivariate normal linear model of the joint time evolution of HIV viral load and CD4 cell count decline since time of infection in the context of additional covariates (sex, region of origin, mode of infection, age at time of diagnosis). Then we used the trained model to estimate infection times from longitudinal biomarker data for Amsterdam patients, with an average of four viral load observations and six CD4 cell count observations per patient. We next reconstructed characteristic time-to-diagnosis distributions for each of the nine Amsterdam risk groups (MSM/heterosexual, and region of birth) with a Bayesian hierarchical model from the individual-level estimates, modelling the individual-level estimates with a Weibull distribution. To avoid censoring of infection-to-diagnosis times, we focused analyses on the subset of infections in 2010–2012 which were diagnosed by 1 May 2019 since most infections in this window would have been diagnosed by the close of study, and assume as supported by mathematical models that time-to-diagnosis did not change substantially in 2010–2019 (*Sighem, 2017*; *Sighem et al., 2017*). The model was implemented with Stan version 2.21 (*Carpenter et al., 2017*). Full details are provided in Appendix 1, Section 3.

We then calculated the proportion of infections in each year $y = 2014, ..., 2018$ in each of the 9 Amsterdam risk groups that were not diagnosed by database closure (which we denote by $\delta_{try}$) from the fitted model. To adjust for trends in incidence over time, the annual estimates were weighted by the estimated number of HIV infections in each year among Amsterdam MSM and heterosexual individuals without stratifiction by inmigrant status, according to the European Centre for Disease Control and Prevention (ECDC) HIV modelling tool for Amsterdam, version 1.3.0 (*Stockholm: European Centre for Disease Prevention and Control, 2017*) through weights,

$$\omega_{ty} = \frac{N_{ty}^{Inf-ECDC}}{\sum_{z \in Y} N_{tz}^{Inf-ECDC}} , \tag{1}$$

where y=2014,...,2018 and $N_{ty}^{Inf-ECDC}$ are the estimated total number of infections in year $y$ in Amsterdam MSM or heterosexuals. We then obtained an overall estimate of the proportion of undiagnosed infections in 2014–2018, $\delta_{tr}$ , by applying these weights to the yearly proportions through

$$\delta_{tr} = \sum_{y \in Y} \omega_{ty} \, \delta_{try} \tag{2}$$

Recognizing the limitations in applying weights that do not account for differences by place of birth, we used in sensitivity analyses as weights the observed trends in the number of annual HIV diagnoses in the corresponding Amsterdam risk group. The total number of Amsterdam infections in 2014–2018 including the undiagnosed (which we denote by $N_{tr}^{Inf}$) was next estimated by dividing the number of diagnosed Amsterdam infections in 2014–2018 (which we denote by $N_{tr}^{D}$) with the estimated proportion of diagnosed individuals,

$$N_{tr}^{Inf} = \frac{N_{tr}^{D}}{1-\delta_{tr}} \tag{3}$$

# Phylogenetic reconstruction of transmission chains among Amsterdam residents

To reconstruct distinct HIV transmission chains among Amsterdam residents, we used the first available partial HIV-1 *polymerase* (*pol*) sequence from Amsterdam PLHIV, Dutch PLHIV from outside Amsterdam, and ~82,000 *pol* sequences from non-Dutch PLHIV. The non-Dutch viral sequences were retrieved from the Los Alamos HIV-1 sequence database subject to a length of at least 1300 in the *pol* gene on March 2, 2020 (www.hiv.lanl.gov). The basic local alignment search tool (BLAST v2.10.0) was used to select the top 20 closest background sequences to any Dutch sequence (*Altschul et al., 1990*). All sequences were subtyped using Comet v2.3 (*Struck et al., 2014*). Sequences with an uncertain subtype classification using Comet were analysed with Rega v3.0 (*Pineda-Peña et al., 2013*). Any remaining sequences for which a subtype could not be resolved were discarded from further analysis (n=122). Subtype-specific alignments were generated with *Virulign* (*Libin et al., 2019*) (Appendix 1 Section 4.1) and sequences from other subtypes were added as outgroup for the purpose of phylogenetic rooting. The final alignments were trimmed to positions 2253–3870 in the reference genome HXB2 (*Ratner et al., 1985*).

Subtype-specific HIV phylogenetic trees were generated for alignments with at least 50 Amsterdam sequences (subtypes and recombinant forms B, 01AE, 02AG, C, D, G, A1 or 06 cpx) using FastTree v2.1.8 (*Price et al., 2010*) rooted at the outgroup, and the outgroup taxa were then pruned from the phylogeny. Next, we attributed to all viral lineages in the phylogenies a 'state' label that included information on the transmission risk group (MSM, heterosexual, other) and location with *phyloscanner* version 1.8.0 (*Wymant et al., 2018*); see *Bezemer et al., 2022* for details. Locations were classified into Amsterdam (for ATHENA patients with an Amsterdam postcode at time of registration or a registration update), the Netherlands (for other ATHENA patients), and the 9 world regions Africa, Western Europe, Eastern Europe and Central Asia, North America, Latin America and the Caribbean, Dutch Caribbean and Suriname, Middle East and North Africa, South and South-East Asia and Oceania (for non-Dutch sequences).

In the labelled phylogeny, the lineage labels jump backwards in time, for example from Amsterdam MSM associated with a lineage ending in a tip observed in Amsterdam MSM to Western Europe. Thus, we can group lineages according to the same label between jumps, and we follow *Wymant et al., 2018* in referring to these groups as *phyloscanner* subgraphs. We assumed that we have sufficient background sequences such that no additional background sequences would further separate transmission chains among Amsterdam residents into more distinct chains. A subtle but important related point is that with the available location data at time of registration or a registration update, we are only able to phylogenetically reconstruct transmission chains by residence status rather than the location at which transmission actually occurred. For example, two Amsterdam residents appear in the same *phyloscanner* subgraph if they infected each other during a short-term visit in another Dutch, European or global location, if they were both infected from a common source during such a short-term visit and the source remained unsampled, if they infected each other before they began their residence in Amsterdam, or after they moved to another Dutch municipality. Diagnosed Amsterdam patients in the same subgraph were then interpreted as belonging to the same transmission chain, and the estimated state of the root of the subgraph was interpreted as the geographical origin of the transmission chain. Throughout, we refer to the subgraphs also as the phylogenetically observed (parts of) transmission chains. Using this approach, we note that unlike most phylogenetic clustering analyses (*Burns et al., 2017*), every infected patient with a sequence is included in one subgraph, and all partially observed transmission chains of size one are included in the analysis to ensure that the entire distribution of observed transmission chains is represented in the analysis (*Bezemer et al., 2022*). To capture phylogenetic uncertainty, phylogenetic analyses were repeated on 100 bootstrap replicates drawn from each subtype alignment, and transmission chains were enumerated across these replicate analyses.

We classified phylogenetically reconstructed transmission chains by the infection dates that we estimated from each patient's diagnosis date, risk group, age, CD4 trajectory and viral load trajectory. Chains were classified as 'pre-existing' if at least one of its members had a posterior median infection date before 2014, and as 'emerging' if all members had a posterior median infection date after January 1, 2014.

## Virally unsuppressed transmission chains

For all pre-existing chains, we determined the number of infectious individuals at the start of 2014 from viral load data. Specifically, we defined patients as suppressed by 2014 if their last viral load measurement before 2014 was below 100 copies/ml, and count for each pre-existing chain its suppressed and unsuppressed members by 2014.

## Estimating the growth of city-level transmission chains

Because of the large number of late presenters and incomplete sequence coverage in diagnosed patients, the phylogenetically observed transmission chains are incomplete and statistical models were required to estimate the growth and origins of Amsterdam transmission chains. We here extended the Bayesian branching process model of *Bezemer et al., 2022* to estimate the growth of pre-existing transmission chains. Specifically, given $m = 1, ..., M$ index cases of a chain that pre-existed, the final size distribution of stuttering transmission chains is under a Negative Binomial branching process model given by

$$c\left(i|\mu m, \phi m\right) = \tfrac{m}{m+i}\text{NegBin}\left(i|\mu m, \phi m\right) \tag{4}$$

where NegBin is the Negative Binomial distribution characterised by mean $\mu m$ and dispersion parameter $\phi m$, $i = 0, 1, 2, ...$ is the number of new cases, and μ < 1. Incomplete sampling of new cases can be accommodated via

$$c_{obs}\left(i|m, \mu, \phi, \rho\right) = \sum_{k=1}^{\infty} \text{Bin}(i|k,p)c(k|m, \mu, \rho) = \sum_{k=1}^{\infty} \text{Bin}(i|k, \rho)\tfrac{m}{m+k}\text{NegBin}\left(k|\mu m, \phi m\right) \ , \tag{5}$$

where $\rho$ denotes the probability that a new case in 2014–2018 is diagnosed and has a viral sequence sampled by database closure. In the model, the index cases are assumed to be infectious and defined by the number of unsuppressed members by 2014 in a pre-existing chain, adjusted for the sampling probability of such members. We further capped the infinite sum in (3) in the model, recognizing that the summands rapidly tend to zero. The corresponding equation for emergent transmission chains (since 2014 as defined above) is similar,

$$\tilde{c}_{obs}\left(n|m = 1, \mu, \phi, \rho\right) = \frac{\sum_{z=n}^{\infty} Bin\left(n|z,\rho\right)\frac{1}{z}NegBin\left(z-1|\mu,\phi\right)}{1-\sum_{z=n}^{\infty}\left(\left(1-\rho\right)^z\frac{1}{z}NegBin\left(z-1|\mu,\phi\right)\right)} \ , \tag{6}$$

where $n = 1, 2, ...$ are the total number of observed cases in an emerging chain. We then denote with $x_s$ and $\tilde{x}_s$ respectively the observed growth distributions for the phylogenetically observed, pre-existing and emergent transmission chains in the phylogeny of subtype/ recombinant form, and for either Amsterdam MSM or heterosexuals, which we denote by $s$. Here, $x_s$ is a matrix with rows indicating the number of index cases and columns indicating the number of new cases, and $\tilde{x}_s$ is a row vector with rows indicating the total number of cases in emerging chains. For ease of reading, we suppress the subscripts where possible from now on. The likelihood then comprises the growth distributions of emerging chains, pre-existing chains that continued to grow, and pre-existing chains with unsuppressed members that did not grow, with the following log-likelihood,

$$l(x, \tilde{x}|\mu, \phi, \rho) = \sum_{m=1}^{M} \sum_{i=0}^{I} x_{mi}\log c_{obs}\left(i|m, \mu, \phi, \rho\right) + \sum_{n=1}^{N} \tilde{x}_n\log \tilde{c}_{obs}\left(n|m = 1, \mu, \phi, \rho\right) , \tag{7}$$

where $M$ is the largest number of index cases observed across the chains after adjusting for sampling, $I$ is the largest number of new cases observed in pre-existing chains and $N$ is the largest number of new cases observed in emergent chains, including the first case. Pre-existing chains for which all members were suppressed by 2014 and which did not grow were not included, because these chains had no unsuppressed index case. Due to small counts, we grouped the observed growth distributions for the phylogenetically observed transmission chains for non-B subtypes together before fitting the model. We fitted the branching process model under a Bayesian framework with Stan version 2.21 to the observed growth distributions among MSM, borrowing information across subtypes B and non-B, and similarly for heterosexuals. The primary output of the model are posterior predictive distributions on the number, size and growth of the actual transmission chains among Amsterdam residents, both for MSM and heterosexuals, and by viral subtype. This includes emerging chains that were entirely unsampled. Full details are provided in Appendix 1, Section 6.

## Derived statistical estimates

Given estimates of the number and growth of both pre-existing and emergent transmission chains, it is straightforward to derive estimates of the proportion of HIV infections among Amsterdam residents in 2014–2018 that had an Amsterdam resident as source (which we denote by $\gamma$ and refer to as the proportion of locally acquired infections). This is because all infections originating from an individual living in Amsterdam had a local source, except the index cases in the emerging chains that were introduced from outside of Amsterdam. Ignoring population subgroups for the derivation, we have

$$\gamma = \frac{N^I - \alpha N^C}{N^I},$$
(8)

where $N^I$ is the estimated number of new infections between 2014 and 2018 in Amsterdam residents, $N^C$ is the estimated number of transmission chains which emerged between 2014 and 2018 and $\alpha$ is the estimated proportion of emergent transmission chains with an Amsterdam origin. Since each transmission chain has one index case, $\alpha N^C$ is the estimated number of infections with non-Amsterdam origin, and $N^I - \alpha N^C$ is the estimated number of infections that had an Amsterdam resident as a source.

Using *Equation 8*, we were able to obtain estimates (8) for Amsterdam MSM residents and Amsterdam heterosexual residents, and for each phylogeny, that is stratified further by each of the major subtypes and recombinant forms (which we denote by $\gamma_s$). To obtain estimates stratified by the nine Amsterdam risk groups of interest (where $t$ denotes transmission group MSM or heterosexual and $r$ denotes geographic region of birth), we calculated weighted averages of the $\gamma_{ts}$ across chains and subtypes, with the weight determined as the proportion of the infected individuals in transmission group $t$ (i.e. either MSM or heterosexuals) from region of birth $r$ that are infected with subtype/recombinant form s. Specifically,

$$\gamma_{tr} = \sum_{s \epsilon S} v_{tsr} \gamma_{ts}, \,$$
(9)

where the proportions $v_{tsr}$ are for brevity defined in Appendix 1 Section 7. We interpret $\gamma_{tr}$ as the proportion of Amsterdam infections in transmission risk group $t$, from geographic region $r$, that have the potential to be preventable through local interventions.

## Ethics

As from 2002 ATHENA is managed by Stichting HIV Monitoring, the institution appointed by the Dutch Ministry of Public health, Welfare and Sport for the monitoring of people living with HIV in the Netherlands. People entering HIV care receive written material about participation in the ATHENA cohort and are informed by their treating physician on the purpose of data collection, thereafter they can consent verbally or elect to opt-out. Data are pseudonymised before being provided to investigators and may be used for scientific purposes. A designated data protection officer safeguards compliance with the European General Data Protection Regulation (*Boender et al., 2018*).

## Results

### Substantial declines in HIV diagnoses and infections in Amsterdam

Between 1 January 2014 and 1 May 2019, there were 846 HIV diagnoses in Amsterdam residents who self-identified as MSM (75%) or heterosexual (20%). Of the remaining diagnoses, 1 (<1%) was among injecting drug users (IDU), 12 (1%) were through other modes of transmission and 30 (3%) had an unknown mode of transmission. A total of 275 (33%) of the diagnoses in MSM and heterosexuals presented with a CD4 count below 350, with late presentation being higher among heterosexuals. All diagnosed patients had biomarker data available to estimate time to diagnosis, and 516 of 846 (61%) were estimated to have been infected between 2014 and 2018 based on the posterior median infection time estimate (*Table 1*). In the preceding 5-year period 2009–2013, there were 1436 HIV diagnoses in Amsterdam and a similar proportion of these presented late (567, 39%). There were 1128 diagnoses with estimated infection in 2009–2013, suggesting a substantial reduction in infections in 2014–2018. Yet, the rate of new Amsterdam diagnoses since 2014 (104 per 100,000) remained higher than the national rates excluding Amsterdam (24 per 100,000), and in this sense Amsterdam remains a HIV hotspot in the Netherlands.

**Table 1.** HIV infections among Amsterdam residents in 2014-2018.

| Risk group | Observed HIV diagnoses in Amsterdam residents in 2014-May 2019 (n) | Observed HIV diagnoses in Amsterdam residents in 2014-May 2019 with CD4 <350 (n) | Observed HIV diagnoses in Amsterdam residents, estimated to have been infected in 2014–2018 (n) | Estimated undiagnosed HIV infections in Amsterdam residents until May 2019 (%) | Estimated HIV infections in Amsterdam residents in 2014–2018 (n) |
|---|---|---|---|---|---|
| Total | 846 | 275 | 516 | 19% [17–21%] | 636 [620-656] |
| MSM (all) | 671 | 192 | 446 | 14% [12–16%] | 516 [506-529] |
| MSM (Dutch-born) | 298 | 103 | 190 | 11% [9–13%] | 214 [209-219] |
| MSM (Born in W. Europe, N. America and Oceania) | 100 | 12 | 80 | 9% [6–14%] | 88 [85-93] |
| MSM (Born in E. and C. Europe) | 51 | 8 | 32 | 16% [11–24%] | 38 [36-42] |
| MSM (Born in S. America and the Caribbean) | 124 | 38 | 83 | 17% [13–22%] | 100 [95-107] |
| MSM (Born in any other country) | 98 | 31 | 61 | 20% [14–27%] | 76 [71-83] |
| Heterosexuals (all) | 175 | 83 | 70 | 41% [35–48%] | 119 [107-135] |
| Heterosexuals (Dutch-born) | 51 | 19 | 23 | 30% [21–44%] | 33 [29-41] |
| Heterosexuals (Born in Sub-Saharan Africa) | 67 | 36 | 17 | 57% [47–67%] | 40 [32-51] |
| Heterosexuals (Born in S. America and the Caribbean) | 37 | 18 | 21 | 28% [19–42%] | 29 [26-36] |
| Heterosexuals (Born in any other country) | 20 | 10 | 9 | 40% [25–57%] | 15 [12-21] |

Posterior estimated median time from infection to diagnosis [95% CI].

## Nine of ten Amsterdam diagnoses and infections are in MSM

A total of 190 (37%) Amsterdam diagnoses with estimated infection in 2014–2018 were in Dutch-born MSM, 256 (50%) in foreign-born MSM, 23 (4%) in Dutch-born men and women identifying as heterosexuals, and 47 (9%) in foreign-born heterosexuals. Thus, the large majority of Amsterdam diagnoses with infection dates between 2014 and 2018 were in foreign-born and Dutch-born MSM, and an important question that we address below is if these diagnoses also likely had an Amsterdam source.

Overall, we find the individual-level time-to-diagnosis estimates varied substantially within each of the 9 Amsterdam risk groups shown in *Table 1* (see also *Appendix 1—figures 1 and 2*). The posterior median time-to-diagnosis estimates among individuals were 14 months longer in heterosexuals than in MSM, 9 months longer in Dutch-born heterosexuals than Dutch-born MSM, and 19 months longer in foreign-born heterosexuals than foreign-born MSM (*Appendix 1—figure 3*). These substantial diagnosis delays continue to undermine the long-term prognosis of infected individuals and transmission prevention efforts.

## High proportion of infections since 2014 that remained undiagnosed by May 2019

Local estimates of the continuum of care indicate that Amsterdam has surpassed the 95-95-95 targets, with an estimated 5% of all people in Amsterdam living with HIV that remained undiagnosed by the

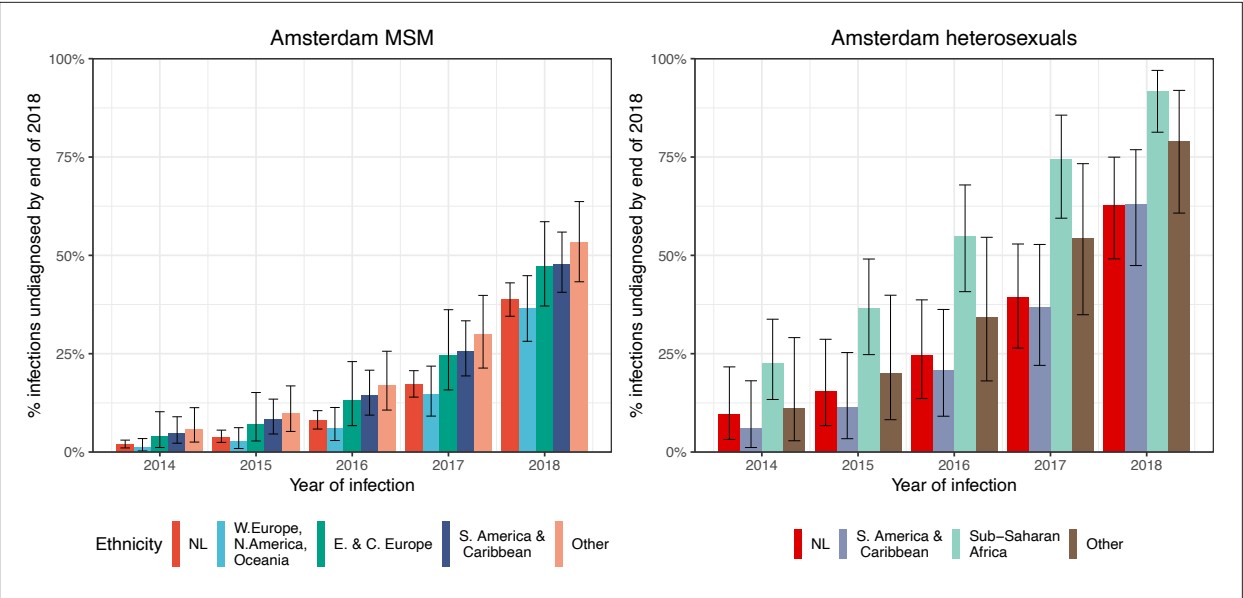

**Figure 2.** HIV infections in Amsterdam residents in 2014–2018 that remained undiagnosed by 1 May 2019. Posterior median estimates are shown as bars and 95% credible intervals as error bars. Estimates generated from time-to-diagnosis estimates for 535 MSM and 97 heterosexuals.

end of 2019 (*Sighem et al., 2020*; *UNAIDS, 2019*). Based on the time-to-diagnosis estimates in our cohort, we can focus here at the forefront of ongoing transmission chains and quantify the proportion of recent Amsterdam infections in 2014–2018 that remained undiagnosed by 1 May 2019. *Figure 2* shows that the estimated undiagnosed proportions are considerably higher when we focus on infections acquired since 2014. Accounting for declining diagnosis and infection trends (see Materials and methods), an estimated 14% [12–16%] of infections in Amsterdan MSM in 2014–2018 remained undiagnosed, and 41% [35–48%] in Amsterdam heterosexuals (*Table 1*). The highest proportion of undiagnosed Amsterdam infections in 2014–2018 are in heterosexuals born in Sub-Saharan Africa, with 57% [47–67%].

While the bivariate model of biomarker data that underpins the individual-level time-to-diagnosis estimates has been validated (*Pantazis et al., 2019*), our estimates of the proportion of undiagnosed infections in 2014–2018 depend further on the trends in the number of infections in each year as shown in *Equation 2*. The main analysis is based on trends in HIV infections in Amsterdam MSM and heterosexuals that were estimated with the ECDC HIV Modelling Tool for Amsterdam. The ECDC estimates account for late diagnoses, but aggregate over region of birth. Recognizing this limitation, in sensitivity analyses we used instead trends in directly observed Amsterdam diagnoses, which apply to each Amsterdam risk group but do not account for confounding due to late diagnoses. In the sensitivity analysis, we estimate that 14% [13–17%] of infections in Amsterdam MSM in 2014–2018 remained undiagnosed, and 34% [28–41%] in Amsterdam heterosexuals. Further details are presented in Appendix 1, Section 3.3–3.5.

## More than 1800 distinct transmission chains among Amsterdam residents

We next adopted viral phylogenetic methods to understand how the diagnosed Amsterdam infections since 2014 are distributed across Amsterdam's HIV transmission networks. A total 378 of the 516 (73%) individuals had a *pol* sequence available, of whom 341 were of the major subtypes or recombinant forms that are circulating in Amsterdam (B, 01AE, 02AG, C, D, G, A1 and 06 cpx). 37 individuals were excluded from further analysis as their subtype identification was inconclusive, or they were associated with other subtypes or recombinant forms with fewer than 50 sequences in Amsterdam. *Appendix 1—table 1* summarises the characteristics of the study population, and those with a sequence available. We reconstructed viral phylogenies using the HIV sequence data from these individuals combined with viral sequences from 3647 Amsterdam diagnoses with estimated

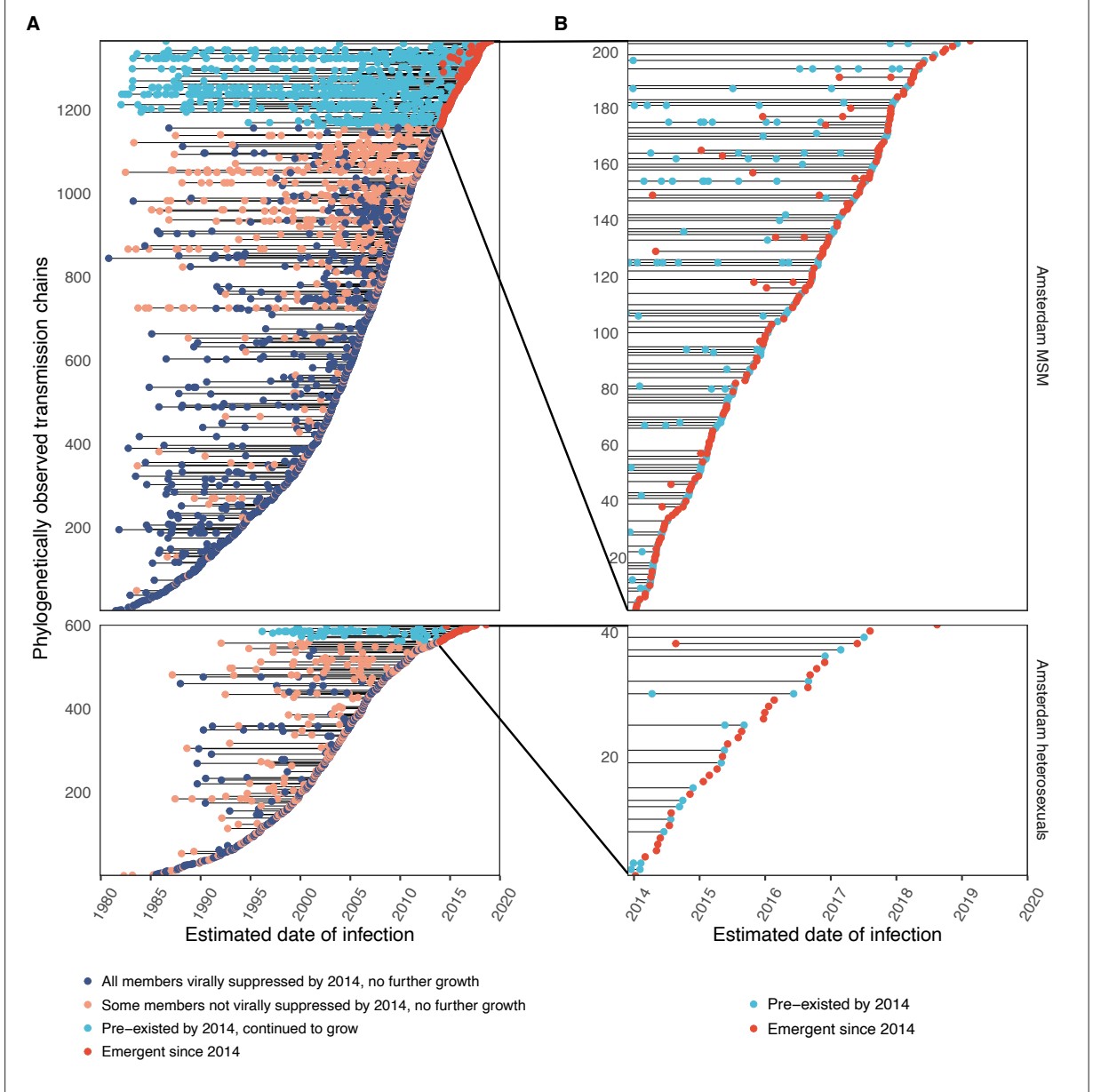

**Figure 3.** Phylogenetically observed parts of Amsterdam transmission chains. (**A**) All chains. Horizontal lines connect individuals in reconstructed transmission chains in Amsterdam by chains which had no new case since 2014, and those which continued to grow or emerged, among MSM (top) and heterosexuals (bottom), in order of last diagnosis per chain. (**B**) Subset of chains in which at least one individual was estimated to have been infected since 2014. Data are presented as in subfigure A.

infection prior to 2014, 6087 diagnosed individuals from the Netherlands outside Amsterdam, and 14,222 viral sequences from outside the Netherlands that were genetically closest to those circulating in the Netherlands (*Appendix 1—figures 4–25*). Key statistics based on the bootstrap analysis are reported in *Appendix 1—Tables 2 and 3*.

We identified across the major HIV-1 subtypes and circulating recombinant forms 1829 distinct viral phylogenetic subgraphs that comprised at least one diagnosed Amsterdam infection prior to 2014, which we refer to as the phylogenetically observed pre-existing transmission chains (*Figure 3* and *Appendix 1—figure 26*). There were 1253 pre-existing chains in MSM, of which 949 (76%) had all members virally suppressed as of 2014, and of those 906 (95%) had no new member in 2014–2018. The remaining 5% of subgraphs likely grew from unsuppressed index individuals that did not have an HIV sequence sampled. In heterosexuals, there were 576 pre-existing chains, of which 401 (70%)

had all members virally suppressed as of 2014, and of those 391 (98%) had no new member in 2014–2018. The proportion of unsuppressed subgraphs in Amsterdam heterosexuals was indeed statistically significantly lower than in Amsterdam MSM, but not strongly so (p-value 0.02, one-sided chi-square test). To summarise, transmission appears to have stopped since 2014 in almost all phylogenetically observed pre-existing chains that had all their observed members suppressed by 2014.

## Growth of the phylogenetically observed parts of city-level transmission chains

Considering growth, 89 (7%) of the 1253 phylogenetically observed pre-existing chains in Amsterdam MSM had at least one new member diagnosed in 2014–2018, and 114 chains emerged (*Table 2* and *Figure 3*). In Amsterdam heterosexuals, 15 (3%) of the 576 phylogenetically observed pre-existing chains had at least one new member diagnosed in 2014–2018, and 26 chains emerged. The emerging chains thus outnumbered the growing pre-existing chains in both Amsterdam MSM and heterosexuals. However, the observed phylogenetic data are challenging to interpret directly because larger proportions of recent infections remain undiagnosed, approximately half of diagnosed individuals did not have a sequence sampled, and small chains are more likely to remain entirely unobserved (see Materials and methods).

## Emerging transmission chains outnumber pre-existing, growing transmission chains

We next used a Bayesian branching process growth model to predict the size and growth of the actual transmission chains (see Materials and methods and Appendix 1, Section 6). Model fit to the observed growth distributions was very good (*Appendix 1—figure 27*). We estimate that there are substantially more emerging chains in Amsterdam since 2014 than phylogenetically observed, 172 [154-195] in MSM and 58 [42-83] in heterosexuals, reflecting that emergent chains have a high probability to be entirely unobserved when growth is below the epidemic reproduction threshold of one (*Table 2*). Thus, the estimated actual, emerging chains outnumber the growing pre-existing chains in both Amsterdam MSM and heterosexuals more strongly than the phylogenetic data suggest.

In terms of proportions, an estimated 61% [55–67%] of the growing chains among Amsterdam MSM were emerging, and 69% [56–81%] of the growing chains among Amsterdam heterosexuals. We estimate further that 47% [39–55%] of the estimated infections among Amsterdam MSM in 2014–2018 were in emerging chains, and 61% [45–77%] of the estimated infections among Amsterdam heterosexuals (*Table 3*). Thus, on average the pre-existing chains contributed more new cases in 2014–2018 to Amsterdam infections than the emerging chains.

## Proportion of locally preventable infections

From the emerging transmission chains, we can directly estimate the proportion of Amsterdam infections since 2014 that had an Amsterdam source (see Materials and methods). We interpret these infections as locally preventable, because they are within the reach of the HIV prevention efforts in Amsterdam. In Amsterdam MSM, an estimated 67% [60–74%] of infections in 2014–2018 were locally preventable, with little variation by region of birth (*Figure 4*, proportions next to error bars). In Amsterdam heterosexuals, an estimated 56% [41–70%] of infections in 2014–2018 were locally preventable, with more variation by region of birth, though we caution that the underlying sample sizes were small.

We next multiplied the proportions of locally preventable infections with the estimated number of infections in 2014–2018 in each of the 9 Amsterdam risk groups to obtain estimates of the absolute number of locally preventable infections in Amsterdam in 2014–2018 in each risk group (*Figure 4*, y-axis). Of the estimated 415 [316-542] locally preventable Amsterdam infections in 2014–2018, an estimated 178 [129-243] (43% [37–49%]) were in foreign-born MSM, 171 [124-231] (41% [35–47%]) in Dutch-born MSM, 45 [24-82] (10% [6–18%]) in foreign-born heterosexuals, and 21 [10-39] (5% [2–9%]) in Dutch-born heterosexuals.

**Table 2.** Growth distribution of transmission chains among Amsterdam residents in 2014–2018.

| Transmission group | New cases | Observed* Pre-existing chains (N) | Observed* Pre-existing chains (%) | Observed* Emerging chains (N) | Observed* Emerging chains (%) | Predicted† Pre-existing chains (N) | Predicted† Pre-existing chains (%) | Predicted† Emerging chains (N) | Predicted† Emerging chains (%) |
|---|---|---|---|---|---|---|---|---|---|
| Amsterdam MSM | 0 | 220 | 71.2% | - | - | 198 [175-221] | 64.1% [56.6–71.5%] | - | - |
| | 1 | 59 | 19.1% | 94 | 82.5% | 52 [37-69] | 16.8% [12.0–22.3%] | 137 [118-158] | 79.7% [72.3–86.1%] |
| | 2 | 15 | 4.9% | 11 | 9.6% | 23 [14-35] | 7.4% [4.5–11.3%] | 19 [11-30] | 11.2% [6.3–17.0%] |
| | 3 | 6 | 1.9% | 7 | 6.1% | 13 [6-20] | 4.2% [1.9–6.5%] | 7 [2-13] | 4.1% [1.2–7.6%] |
| | 4 | 3 | 1.0% | 2 | 1.8% | 7 [3-14] | 2.3% [1.0–4.5%] | 3 [0-8] | 1.8% [0.0–4.3%] |
| | 5 | 2 | 0.6% | 0 | 0.0% | 4 [1-10] | 1.3% [0.3–3.2%] | 2 [0-5] | 1.1% [0.0–2.9%] |
| | 6 | 0 | 0.0% | 0 | 0.0% | 3 [0-7] | 1.0% [0.0–2.3%] | 1 [0-4] | 0.6% [0.0–2.1%] |
| | 7+ | 4 | 1.3% | 0 | 0.0% | 7 [2-14] | 2.3% [0.6–4.5%] | 2 [0-6] | 1.1% [0.0–3.2%] |
| | Total that grew | 89 | | 114 | | 111 [88-134] | | 172 [154-195] | |
| | Total | 309 | | 114 | | 309 [309-309] | | 172 [154-195] | |
| Amsterdam heterosexual | 0 | 150 | 90.9% | - | - | 138 [123-150] | 83.6% [74.5–90.9%] | - | - |
| | 1 | 13 | 7.9% | 25 | 96.2% | 17 [9-28] | 10.3% [5.5–17.0%] | 50 [35-72] | 86.4% [74.1–95.6%] |
| | 2 | 2 | 1.2% | 1 | 3.8% | 5 [1-11] | 3.0% [0.6–6.7%] | 5 [1-12] | 9.3% [2.0–19.0%] |
| | 3 | 0 | 0.0% | 0 | 0.0% | 2 [0-6] | 1.2% [0.0–3.6%] | 1 [0-5] | 2.0% [0.0–7.8%] |
| | 4 | 0 | 0.0% | 0 | 0.0% | 1 [0-3] | 0.6% [0.0–1.8%] | 0 [0-2] | 0.0% [0.0–4.3%] |
| | 5 | 0 | 0.0% | 0 | 0.0% | 0 [0-2] | 0.0% [0.0–1.2%] | 0 [0-2] | 0.0% [0.0–2.6%] |
| | 6 | 0 | 0.0% | 0 | 0.0% | 0 [0-2] | 0.0% [0.0–1.2%] | 0 [0-1] | 0.0% [0.0–2.0%] |
| | 7+ | 0 | 0.0% | 0 | 0.0% | 0 [0-3] | 0.0% [0.0–1.8%] | 0 [0-1] | 0.0% [0.0–2.0%] |
| | Total that grew | 15 | | 26 | | 27 [15-42] | | 58 [42-83] | |
| | Total | 165 | | 26 | | 165 [165-165] | | 58 [42-83] | |

*Parts of the actual Amsterdam transmission chains were observed in viral phylogenies of the major subtypes and circulating recombinant forms (B, 01AE, 02AG, C, D, G, A1 or 06 cpx).
†Predicted based on the Bayesian branching process growth model and accounting for undiagnosed and unsampled individuals.

**Table 3.** Distribution of Amsterdam infections since 2014 in pre-existing and emerging transmission chains.

| | Observed* | | | | | Predicted† | | | | | |
| --- | --- | --- | --- | --- | --- | --- | --- | --- | --- | --- | --- |
| | Total | In pre-existing chains | | In emerging chains | | Total | In pre-existing chains | | In emerging chains | | |
| | (N) | (N) | (%) | (N) | (%) | (N) | (N) | (%) | (N) | (%) | |
| MSM (Dutch) | 145 | 86 | 59.30% | 59 | 40.70% | 254 [202-318] | 136 [95-188] | 53.6% [44.1-62.4%] | 117 [93-147] | 46.4% [37.6-55.9%] | |
| MSM (W. Europe, N. America, Oceania) | 40 | 25 | 62.50% | 15 | 37.50% | 68 [49-91] | 37 [23-56] | 54.8% [40.5-68.1%] | 31 [20-43] | 45.2% [31.9-59.5%] | |
| MSM (E. & C. Europe) | 17 | 9 | 52.90% | 8 | 47.10% | 29 [18-42] | 15 [8-25] | 53.6% [34.2-72.7%] | 13 [7-21] | 46.4% [27.3-65.8%] | |
| MSM (S. America & Caribbean) | 53 | 24 | 45.30% | 29 | 54.70% | 95 [72-126] | 50 [33-74] | 52.8% [40.3-64.8%] | 45 [31-61] | 47.2% [35.2-59.7%] | |
| MSM (Other) | 42 | 14 | 33.30% | 28 | 66.70% | 76 [55-103] | 37 [22-57] | 48.4% [34.4-61.7%] | 39 [26-56] | 51.6% [38.3-65.6%] | |
| **MSM (All)** | 297 | 158 | 53.20% | 139 | 46.80% | 523 [427-647] | 276 [200-377] | 52.8% [44.6-60.7%] | 246 [206-300] | 47.2% [39.3-55.4%] | |
| Heterosexual (Dutch) | 14 | 2 | 14.30% | 12 | 85.70% | 38 [23-59] | 14 [5-29] | 37.8% [17.5-58.9%] | 23 [13-38] | 62.2% [41.1-82.5%] | |
| Heterosexual (Sub-Saharan Africa) | 11 | 4 | 36.40% | 7 | 63.60% | 30 [17-51] | 10 [3-24] | 34.3% [11.3-58.6%] | 20 [11-34] | 65.7% [41.4-88.7%] | |
| Heterosexual (S. America & Caribbean) | 14 | 8 | 57.10% | 6 | 42.90% | 35 [20-58] | 14 [5-33] | 42.9% [18.6-65.8%] | 19 [10-34] | 57.1% [34.2-81.4%] | |
| Heterosexual (Other) | 5 | 3 | 60.0% | 2 | 40.0% | 13 [6-23] | 5 [1-12] | 39.1% [9.1-70.0%] | 8 [3-15] | 60.9% [30.0-90.9%] | |
| **Heterosexual (All)** | 44 | 17 | 38.60% | 27 | 61.40% | 117 [80-173] | 45 [22-83] | 38.7% [22.6-54.9%] | 71 [49-105] | 61.3% [45.1-77.4%] | |

*Parts of the actual Amsterdam transmission chains were observed in viral phylogenies of the major subtypes and circulating recombinant forms (B, 01AE, 02AG, C, D, G, A1 or 06 cpx).

†Predicted based on the Bayesian branching process growth model and accounting for undiagnosed and unsampled individuals.

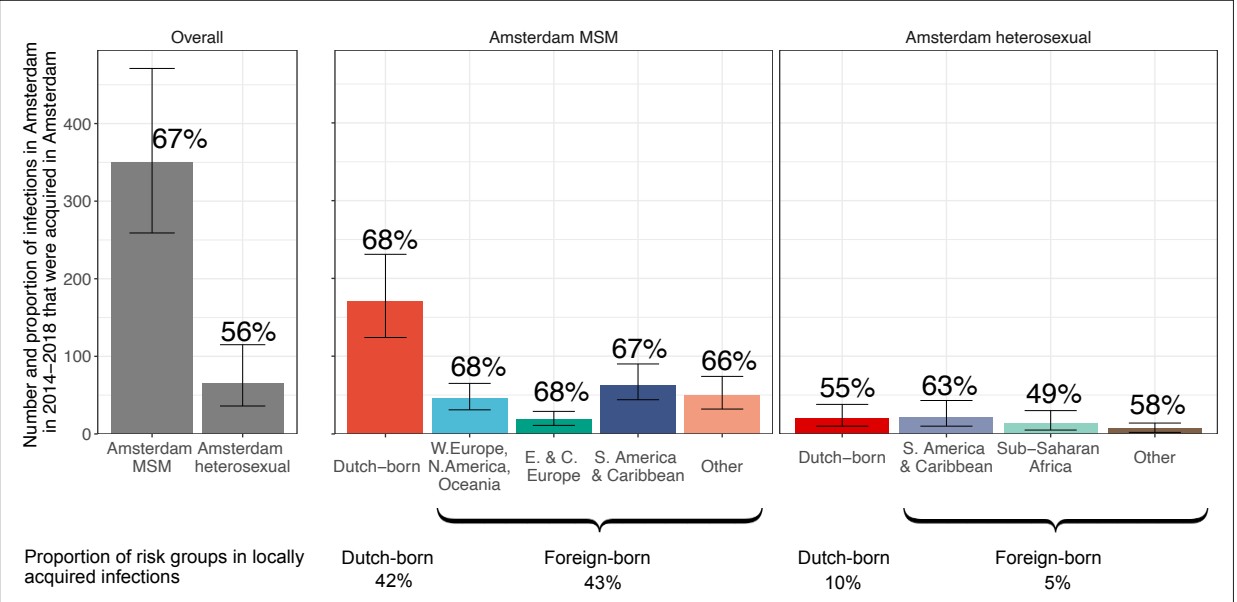

**Figure 4.** Estimated number of locally preventable infections in 2014–2018 along with 95% credible intervals, for MSM and heterosexuals stratified by region of birth. Posterior median estimates of proportion (%) of preventable infections shown above bars. Estimates generated from 203 phylogenetic subgraphs among Amsterdam MSM, containing 297 individuals, and 41 subgraphs among Amsterdam heterosexuals, containing 44 individuals.

## Discussion

More than 300 cities have by the end of 2021 signed the Fast-Track Cities Paris Declaration and committed to end the AIDS epidemic by 2030, addressing disparities in access to basic health and social services, social justice and economic opportunities. The city of Amsterdam reached the UNAIDS Fast-track Cities 95-95-95 targets before the onset of the COVID-19 pandemic, and has seen a decade of declines in city-level HIV diagnoses. Here, we characterised the number, size and growth of HIV transmission chains among Amsterdam residents, and quantified the further potential of preventing HIV infection at city level. It is important to recognize that through the analyses conducted here, the exact location of infection events cannot be identified. Rather, the available location data enable us to identify groups of Amsterdam residents with phylogenetically distinct HIV, which are the inferential basis for estimating the number, size, and growth of the actual unobserved transmission chains among Amsterdam residents. Regardless of the exact infection location, Amsterdam residents live in Amsterdam, and are thus within reach of Amsterdam public health and local prevention interventions.

We can structure our insights in four themes. First, when focusing on the denominator of recent infections that are estimated to have occurred in the 5-year period 2014–2018, the proportions of individuals that remained undiagnosed by early 2019 were high and variable, between 9% and 20% in (self-identified) Amsterdam MSM risk groups, and between 28% and 57% in Amsterdam heterosexual risk groups. These results underscore that strategies aimed at raising awareness of HIV infection, providing easy access to checking symptoms of early HIV infection, encouraging frequent testing, PrEP provision, addressing fears of a positive test and reducing stigma are vital to break the forefront of ongoing HIV transmission chains (https://hebikhiv.nl/en/; *Dijkstra et al., 2017*; *Heijman et al., 2009*; *Burns et al., 2017*; *Myers et al., 2018*). The estimated times to diagnosis document substantial disparities across risk groups in entering HIV care in Amsterdam, and separate efforts have characterised individuals with late diagnoses (*Op de Coul et al., 2016*; *Bil et al., 2019*; *Slurink et al., 2021*). We explored the impact of assumptions on incidence trends to the undiagnosed estimates and found some sensitivities (Appendix 1, Section 3.3), although estimates were all very similar as long as the assumed incidence trends reflected available data. Further sensitivity analyses are reported in Appendix 1 Section 3.4–3.5. We further validated the time-to-diagnosis estimates by comparing the estimated proportion of recent HIV infections (≤6 months) with those estimated in an independent study in Amsterdam using avidity assays (*Slurink et al., 2021*), and found them to be similar

(**Appendix 1—figure 28**). The main limitation of our biomarker approach is thus that at present we cannot account for time trends in time-to-diagnosis.

Second, we documented the growth of Amsterdam HIV transmission chains in which all phylogenetically observed members were virally suppressed by 2014. We find that regardless of risk group, almost all such virally suppressed chains did not grow in the sense that no new infections were phylogenetically observed. These results are unsurprising and mirror the established relationship that treatment for HIV infection, which results in undetectable viral load equals untransmittable virus (**Rodger et al., 2019**).

Third, we initially speculated that with a decade of declining HIV diagnoses in Amsterdam, those infections that still occur might be concentrated in newly seeded, emerging transmission chains. It is challenging to interpret the directly observed data because high proportions of individuals remain undiagnosed and/or are not sequenced, and emerging chains are more likely to be completely undetected. We thus used statistical growth models accounting for unsampled cases, and we estimate in contrast to our initial speculations that 53% of new Amsterdam MSM infections in 2014–2018 grew from chains that existed prior to 2014, and 39% of new Amsterdam heterosexual infections. Following up and tracing back from known transmission chains is easier than discovering emerging chains, and so the many new infections that originate in existing chains have particularly high prevention potential (**Oster et al., 2018**; **Little et al., 2021**; **Dennis et al., 2021**).

Fourth, we quantified the locally preventable infections among Amsterdam residents in 2014–2018, defined as the infections in Amsterdam residents in 2014–2018 who are estimated to have as source another Amsterdam resident. Using the virus' genetic code as an objective marker into infection events, we estimate that regardless of declining diagnoses and incidence, the majority of infections in Amsterdam residents in 2014–2018 remained locally preventable in all risk groups investigated. The statistical strength of evidence into this finding was strong for Amsterdam MSM (all 95% credible intervals for the proportion of locally preventable infections were above 50%), but more moderate for Amsterdam heterosexuals (wider credible intervals including 50%), reflecting that relatively few infections in Amsterdam heterosexuals in 2014–2018 were observed with a viral sequence by early 2019 due to frequent late diagnosis and incomplete viral sequencing. These findings are consistent with data from clinic surveys in migrants across Europe (**Alvarez-Del Arco et al., 2017**), which indicated similar levels of in-country HIV acquisition post migration of 51% in heterosexual women and 58% in heterosexual men.

In summary, our data from 2014 to 2018 indicates considerable potential to prevent HIV infections among Amsterdam residents through city-level interventions, even in the context of substantial improvements in curbing the number of diagnoses and infections in Amsterdam over the past 10 years. Within the similarities in demographics, HIV burden, access to care, and prevention approaches between Amsterdam and many cities in Western Europe and worldwide, our conclusions are relevant to the wider UNAIDS Fast-Track cities, and provide evidence-based support for locally targeted combination HIV prevention interventions in metropolitan areas. COVID-19 has severely disrupted prevention messaging, testing and PrEP services and early pathways to care, making innovative and targeted HIV prevention approaches all the more important.

## Acknowledgements

We thank the steering committee of the Amsterdam HIV transmission initiative and the Machine Learning & Global Health network for earlier comments on this work; and Imperial College Research Computing Service, DOI: 10.14469/h-pc/2232, for providing the computational resources to perform this study.

The H-TEAM initiative is being supported by Aidsfonds (grant number: 2013169, P29701, P60803), Stichting Amsterdam Dinner Foundation, Bristol-Myers Squibb International Corp. (study number: AI424-541), Gilead Sciences Europe Ltd (grant number: PA-HIV-PREP-16-0024), Gilead Sciences (protocol numbers: CO-NL-276-4222, CO-US-276-1712, CO-NL-985-6195), and M.A.C AIDS Fund.

## Additional information

### Competing interests

Ard van Sighem: Funding for managing the ATHENA cohort is supported by a grant from the Dutch Ministry of Health, Welfare and Sport through the Centre for Infectious Disease Control of the National Institute for Public Health and the Environment. Received grants unrelated to this study from European Centre for Disease Prevention and Control paid to author's institution. Nikos Pantazis: Received grants unrelated to this study from ECDC and Gilead Sciences Hellas, paid to author's institution. Received honoraria for presentations unrelated to this study from Gilead Sciences Hellas. Maria Prins: Received unrestricted research grants and speaker/ advisory fees from Gilead Sciences, Abbvie and MSD; all of which were paid to author's institute and were unrelated to this study. Peter Reiss: Has received grants unrelated to this study from Gilead Sciences, ViiV Healthcare and Merck, paid to author's institution. Received Honoraria for lecture from Merck paid to institution. Received Honoraria from Gilead Sciences, ViiV Healthcare and Merck, paid to institution. Godelieve J de Bree: Received honoraria to her Institution for scientific advisory board participations for Gilead Sciences and speaker fees from Gilead Sciences (2019), Takeda (2018-2022) and ExeVir (2020-current). The other authors declare that no competing interests exist.

### Funding

| Funder | Grant reference number | Author |
| --- | --- | --- |
| Aids Fonds | P29701 | Alexandra Blenkinsop<br>Ard van Sighem<br>Oliver Ratmann<br>Godelieve J de Bree |

The funders had no role in study design, data collection and interpretation, or the decision to submit the work for publication.

### Author contributions

Alexandra Blenkinsop, Formal analysis, Methodology, Writing – original draft; Mélodie Monod, Nikos Pantazis, Formal analysis, Writing – review and editing; Ard van Sighem, Data curation, Writing – review and editing; Daniela Bezemer, Data curation, Writing – review and editing, Study oversight; Eline Op de Coul, Writing – review and editing, Study oversight; Thijs van de Laar, Writing – review and editing, Study oversight; Christophe Fraser, Writing – review and editing, Study oversight; Maria Prins, Writing – review and editing, Study oversight; Peter Reiss, Supervision, Writing – review and editing, Study oversight; Godelieve J de Bree, Supervision, Funding acquisition, Writing – review and editing, Study oversight; Oliver Ratmann, Conceptualization, Supervision, Funding acquisition, Methodology, Writing – original draft

### Author ORCIDs

Alexandra Blenkinsop ⬦ http://orcid.org/0000-0002-2328-8671
Ard van Sighem ⬦ http://orcid.org/0000-0002-6656-0516

### Ethics

Human subjects: As from 2002 ATHENA is managed by Stichting HIV Monitoring, the institution appointed by the Dutch Ministry of Public health, Welfare and Sport for the monitoring of people living with HIV in the Netherlands. People entering HIV care receive written material about participation in the ATHENA cohort and are informed by their treating physician on the purpose of data collection, thereafter they can consent verbally or elect to opt-out. Data are pseudonymised before being provided to investigators and may be used for scientific purposes. A designated data protection officer safeguards compliance with the European General Data Protection Regulation (Boender et al. 2018).

### Decision letter and Author response

Decision letter https://doi.org/10.7554/eLife.76487.sa1
Author response https://doi.org/10.7554/eLife.76487.sa2

## Additional files

### Supplementary files
• MDAR checklist

### Data availability
Anonymised data are available in the public Github repository https://github.com/alexblenkinsop/locally.acquired.infections, (copy archived at swh:1:rev:02b5e1150acce280def913a0d2e30ec13a880122). These include aggregated time-to-diagnosis data, and reconstructed phylogenetic trees labelled by one of the 9 Amsterdam risk groups and year of sequence sample. Statistical information or data for separate research purposes from the ATHENA cohort can be requested by submitting a research proposal (https://www.hiv-monitoring.nl/english/research/research-projects/). HIV physicians can review the data of their own treatment centre and compare these data with the full cohort through an online report builder. For correspondence: hiv.monitoring@amc.uva.nl.

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

# Appendix 1

## Supplementary tables and figures

**Appendix 1—table 1.** Patient characteristics for Amsterdam residents with an estimated infection date between 2014 and 2018.

| Strata | | All patients | Patients with a sequence* |
|---|---|---|---|
| Sex | Female | 40 (7.8%) | 24 (7%) |
| | Male | 476 (92.2%) | 317 (93%) |
| Risk group | MSM | 446 (86.4%) | 297 (87.1%) |
| | Heterosexual | 70 (13.6%) | 44 (12.9%) |
| Age group at estimated time of infection | 18–24 | 74 (14.3%) | 48 (14.1%) |
| | 25–34 | 209 (40.5%) | 124 (36.4%) |
| | 35–44 | 113 (21.9%) | 76 (22.3%) |
| | 45–59 | 110 (21.3%) | 87 (25.5%) |
| | 60+ | 10 (1.9%) | 6 (1.8%) |
| Place of birth | Sub-Saharan Africa | 24 (4.8%) | 16 (4.8%) |
| | Asia | 20 (4%) | 13 (3.9%) |
| | Australia & New Zealand | 2 (0.4%) | 2 (0.6%) |
| | Central Europe | 25 (5%) | 16 (4.8%) |
| | Eastern Europe | 8 (1.6%) | 1 (0.3%) |
| | Suriname, Curacao and Aruba | 41 (8.1%) | 32 (9.6%) |
| | South America and Caribbean | 63 (12.5%) | 35 (10.5%) |
| | Middle East and North Africa | 31 (6.1%) | 20 (6%) |
| | Netherlands | 213 (42.2%) | 159 (47.6%) |
| | North America | 23 (4.6%) | 14 (4.2%) |
| | Western Europe | 55 (10.9%) | 26 (7.8%) |
| Estimated time to diagnosis (years) | | 0.4 [0.04–3.2] | 0.41 [0.03–3.25] |

*Patients with sequence of a subtype or circulating recombinant form B, 01AE, 02AG, C, D, G, A1 or 06 cpx

**Appendix 1—table 2.** Number and size of phylogenetically observed transmission chains by transmission risk group and HIV subtype or circulating recombinant form (CRF) for central analysis. 95% confidence intervals are obtained from 100 bootstrap analyses for each subtype alignment.

| Risk group | Subtype or CRF | Total number of chains | Chains of size 1 | Chains of size 2-5 | Chains of size 5-10 | Chains of size ≥10 |
|---|---|---|---|---|---|---|
| Amsterdam MSM | B | 1237 [1259-2097] | 856 [872-1446] | 276 [264-479] | 64 [58-116] | 41 [32-66] |
| | 01AE | 41 [37-46] | 24 [21-32] | 15 [12-17] | 2 [0-3] | 0 [0-1] |
| | 02AG | 26 [21-34] | 17 [14-27] | 7 [2-9] | 1 [0-4] | 1 [0-2] |
| | C | 26 [24-28] | 22 [18-25] | 4 [3-6] | 0 [0-0] | 0 [0-0] |
| | A1 | 21 [18-25] | 13 [10-18] | 6 [4-7] | 0 [0-3] | 2 [0-2] |
| | G | 9 [8-9] | 0 [0-0] | 8 [6-8] | 1 [1-2] | 0 [0-0] |
| | D | 6 [6-6] | 6 [6-6] | 0 [0-0] | 0 [0-0] | 0 [0-0] |
| | 06cpx | 2 [2-2] | 2 [2-2] | 0 [0-0] | 0 [0-0] | 0 [0-0] |

*Appendix 1—table 2 Continued on next page*

*Appendix 1—table 2 Continued*

| Risk group | Subtype or CRF | Total number of chains | Chains of size 1 | Chains of size 2-5 | Chains of size 5-10 | Chains of size ≥10 |
|---|---|---|---|---|---|---|
| Amsterdam heterosexuals | B | 277 [272-482] | 225 [217-392] | 45 [39-77] | 6 [2-9] | 1 [1-3] |
| | 01AE | 23 [20-24] | 19 [15-21] | 4 [3-6] | 0 [0-0] | 0 [0-0] |
| | 02AG | 111 [106-126] | 77 [77-100] | 30 [20-31] | 4 [1-6] | 0 [0-1] |
| | C | 87 [82-89] | 72 [63-75] | 15 [13-19] | 0 [0-1] | 0 [0-0] |
| | A1 | 43 [37-49] | 34 [30-42] | 8 [3-12] | 1 [0-2] | 0 [0-1] |
| | G | 28 [28-33] | 22 [20-29] | 6 [4-8] | 0 [0-0] | 0 [0-0] |
| | D | 16 [15-18] | 12 [10-16] | 4 [2-5] | 0 [0-0] | 0 [0-0] |
| | 06cpx | 17 [14-21] | 12 [8-15] | 4 [2-8] | 1 [0-2] | 0 [0-1] |

**Appendix 1—table 3.** Estimated numbers of phylogenetic transmission chains with ancestral origins in each geographic region from central analysis.

95% confidence intervals obtained from 100 bootstrap analyses for each subtype alignment.

| Subtype or CRF | Estimated ancestral origin | Amsterdam MSM | Amsterdam heterosexuals |
|---|---|---|---|
| B | Amsterdam - other risk group | 16 [8-27] | 73 [59-124] |
| | Netherlands | 699 [721-1238] | 110 [113-199] |
| | Western Europe | 147 [133-253] | 18 [6-24] |
| | Eastern Europe and Central Asia | 27 [21-46] | 1 [1-3] |
| | North America | 84 [71-151] | 7 [4-20] |
| | South America and Caribbean | 21 [16-43] | 1 [1-4] |
| | Middle East and North Africa | 2 [1-5] | - |
| | South and South-East Asia | 3 [2-8] | - |
| | Oceania | 1 [1-3] | - |
| 01AE | Amsterdam - other risk group | - | 2 [1-4] |
| | Netherlands | 11 [5-17] | 10 [5-14] |
| | Middle East and North Africa | 1 [1-1] | - |
| | South and South-East Asia | 21 [14-24] | 8 [3-9] |
| 02AG | Amsterdam - other risk group | - | 5 [3-8] |
| | Netherlands | 11 [6-20] | 29 [20-39] |
| | Sub-Saharan Africa | 4 [1-7] | 39 [29-51] |
| | Western Europe | 5 [1-4] | 2 [1-9] |
| C | Amsterdam - other risk group | 2 [1-3] | 1 [1-2] |
| | Netherlands | 8 [3-9] | 21 [15-26] |
| | Sub-Saharan Africa | 4 [2-7] | 29 [25-39] |
| | Western Europe | 1 [1-3] | 2 [1-7] |
| | South America and Caribbean | 2 [1-3] | 1 [1-1] |
| | South and South-East Asia | 3 [1-3] | 1 [1-2] |
| A1 | Amsterdam - other risk group | 1 [1-2] | 3 [1-5] |
| | Netherlands | 10 [6-13] | 19 [12-24] |
| | Sub-Saharan Africa | 1 [1-2] | 11 [9-17] |
| | Western Europe | 2 [1-3] | - |

*Appendix 1—table 3 Continued on next page*

*Appendix 1—table 3 Continued*

| Subtype or CRF | Estimated ancestral origin | Amsterdam MSM | Amsterdam heterosexuals |
|---|---|---|---|
| | Eastern Europe and Central Asia | 1 [1-2] | - |
| A1 | South and South-East Asia | 3 [1-3] | - |
| | Netherlands | 2 [1-3] | 5 [1-7] |
| | Sub-Saharan Africa | 1 [1-3] | 12 [9-18] |
| | Western Europe | 1 [1-2] | 3 [1-6] |
| G | Eastern Europe and Central Asia | 2 [1-2] | 1 [1-1] |
| | Netherlands | 1 [1-2] | 2 [1-6] |
| D | Sub-Saharan Africa | 2 [1-3] | 9 [5-11] |
| | Netherlands | - | 1 [1-4] |
| | Sub-Saharan Africa | 1 [1-1] | 9 [6-14] |
| | Western Europe | 1 [1-1] | - |

**Appendix 1—table 4.** Viral suppression status of the phylogenetically observed pre-2014 Amsterdam transmission chains.

| Risk group | Subtype | All sampled individuals virally suppressed by 2014* | Pre-2014 chains | Pre-2014 chains that grew | | Individuals (Total) | Individuals (infected before 2014) | | Individuals (infected before 2014 and not virally suppressed) | |
|---|---|---|---|---|---|---|---|---|---|---|
| | | | (n) | (n) | (%) | (n) | (n) | (%) | (n) | (%) |
| Amsterdam MSM | B | Yes | 866 | 35 | 4% | 1432 | 1279 | 89% | 0 | 0% |
| | B | No | 286 | 44 | 15% | 1740 | 1303 | 75% | 352 | 20% |
| | Non-B | Yes | 83 | 8 | 10% | 172 | 119 | 69% | 0 | 0% |
| | Non-B | No | 18 | 2 | 11% | 80 | 51 | 64% | 23 | 29% |
| | Total | | 1253 | 89 | 7% | 3424 | 2752 | 80% | 375 | 11% |
| Amsterdam heterosexual | B | Yes | 180 | 5 | 3% | 218 | 200 | 92% | 0 | 0% |
| | B | No | 85 | 4 | 5% | 284 | 189 | 67% | 90 | 32% |
| | Non-B | Yes | 221 | 5 | 2% | 301 | 281 | 93% | 0 | 0% |
| | Non-B | No | 90 | 1 | 1% | 235 | 142 | 60% | 92 | 39% |
| | Total | | 576 | 15 | 3% | 1038 | 812 | 78% | 182 | 18% |
| Total | | | 1829 | 104 | 6% | 4462 | 3564 | 80% | 557 | 12% |

*Individuals infected prior to 2014, with last viral load measurement before 2014 below 100copies/ml.

**Appendix 1—table 5.** Observed and estimated ancestral origins of phylogenetic subgraphs and estimated complete transmission chains with new cases in 2014-2018.

| Risk group | Subtype | Origin of chains | Observed (N) | Observed (%) | Predicted (N) | Predicted (%) |
|---|---|---|---|---|---|---|
| Amsterdam MSM | B | Amsterdam - other risk group | 1 [1-3] | 0.8% [0.5-2%] | 2 [1-6] | 0.5% [0.2-1.4%] |
| | | Asia | 2 [2-4] | 1.5% [1-2.3%] | 6 [2-12] | 1.5% [0.5-2.8%] |
| | | Eastern Europe and Central Asia | 7 [4-13] | 5% [2.9-7.3%] | 21 [12-30] | 5% [3-7.3%] |
| | | South America and Caribbean | 5 [2-12] | 3.2% [1.5-5.9%] | 14 [8-22] | 3.4% [1.9-5.4%] |

*Appendix 1—table 5 Continued on next page*

Appendix 1—table 5 Continued

| Risk group | Subtype | Origin of chains | Observed (N) | Observed (%) | Predicted (N) | Predicted (%) |
|---|---|---|---|---|---|---|
| | | Middle East and North Africa | 1 [1-2] | 0.8% [0.5-1.3%] | 3 [1-7] | 0.7% [0.2-1.7%] |
| | | Netherlands | 96 [84-159] | 71.1% [64-77.1%] | 294 [272-317] | 71.1% [66.8-75.4%] |
| | | North America | 8 [4-17] | 5.7% [2.5-9.3%] | 23 [15-33] | 5.7% [3.6-8%] |
| | | Oceania | 2 [2-2] | 1% [1-1%] | 1 [1-2] | 0.2% [0.2-0.5%] |
| | | Western Europe | 16 [11-29] | 11.7% [8-15.9%] | 48 [36-61] | 11.6% [8.7-14.9%] |
| | Non-B | Sub-Saharan Africa | 3 [1-5] | 10.7% [3.6-19.6%] | 7 [3-13] | 10.8% [4.2-19%] |
| | | Amsterdam - other risk group | 1 [1-3] | 3.9% [3.3-11.4%] | 2 [1-4] | 2.5% [1.3-6.2%] |
| | | Asia | 8 [6-11] | 31% [22.2-42.3%] | 21 [13-30] | 31.3% [20.3-43.1%] |
| | | Eastern Europe and Central Asia | 1 [1-1] | 3.5% [3.3-3.6%] | 1 [1-2] | 1.5% [1.3-2.8%] |
| | | South America and Caribbean | 1 [1-2] | 4% [3.3-8.2%] | 3 [1-7] | 4.4% [1.4-10%] |
| | | Middle East and North Africa | 1 [1-1] | 3.6% [3.3-4%] | 1 [1-3] | 1.5% [1.3-4.2%] |
| | | Netherlands | 12 [8-16] | 46.4% [32.1-59.5%] | 31 [22-41] | 45.9% [34.2-57.8%] |
| Amsterdam heterosexual | B | Amsterdam - other risk group | 3 [1-7] | 21.4% [7.4-38.5%] | 22 [14-30] | 21.4% [13.8-29.4%] |
| | | Eastern Europe and Central Asia | 1 [1-1] | 7.2% [6.7-7.7%] | 1 [1-2] | 1% [0.9-1.9%] |
| | | Netherlands | 11 [8-17] | 75% [54.8-92%] | 75 [64-89] | 74.8% [66.3-82.8%] |
| | | North America | 1 [1-3] | 6.7% [4.7-10.6%] | 2 [1-4] | 1.9% [0.9-4.2%] |
| | | Western Europe | 1 [1-3] | 7.1% [5.3-20.3%] | 2 [1-6] | 2.1% [0.9-5.5%] |
| | Non-B | Sub-Saharan Africa | 5 [2-8] | 33.3% [9.4-51.9%] | 39 [29-51] | 31.9% [24-40.5%] |
| | | Amsterdam - other risk group | 1 [1-2] | 6.7% [5.4-12.5%] | 9 [3-15] | 7% [2.7-11.8%] |
| | | Asia | 1 [1-1] | 6.7% [5.7-9.8%] | 2 [1-6] | 1.7% [0.8-4.7%] |
| | | Netherlands | 8 [4-12] | 50% [28.9-74.2%] | 62 [50-77] | 50.4% [41.7-59.7%] |
| | | North America | 1 [1-1] | 5.6% [5.6-5.6%] | 1 [1-2] | 0.8% [0.7-1.6%] |

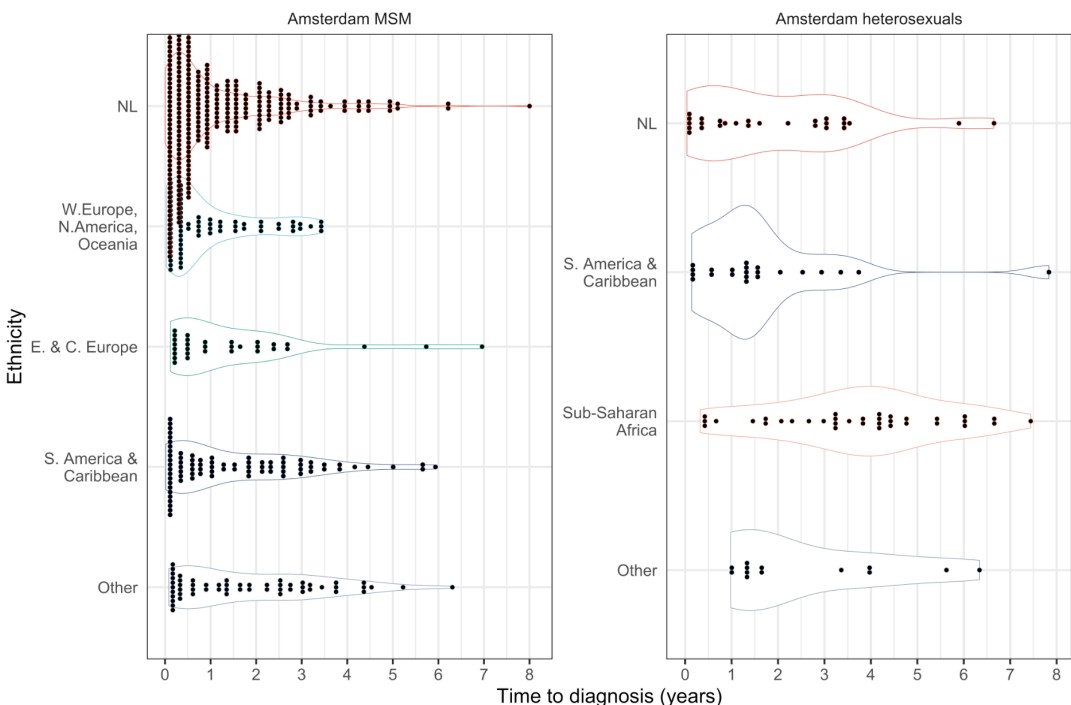

**Appendix 1—figure 1.** Distribution of individual level posterior median estimated times to diagnosis by place of birth, for Amsterdam MSM and heterosexuals.

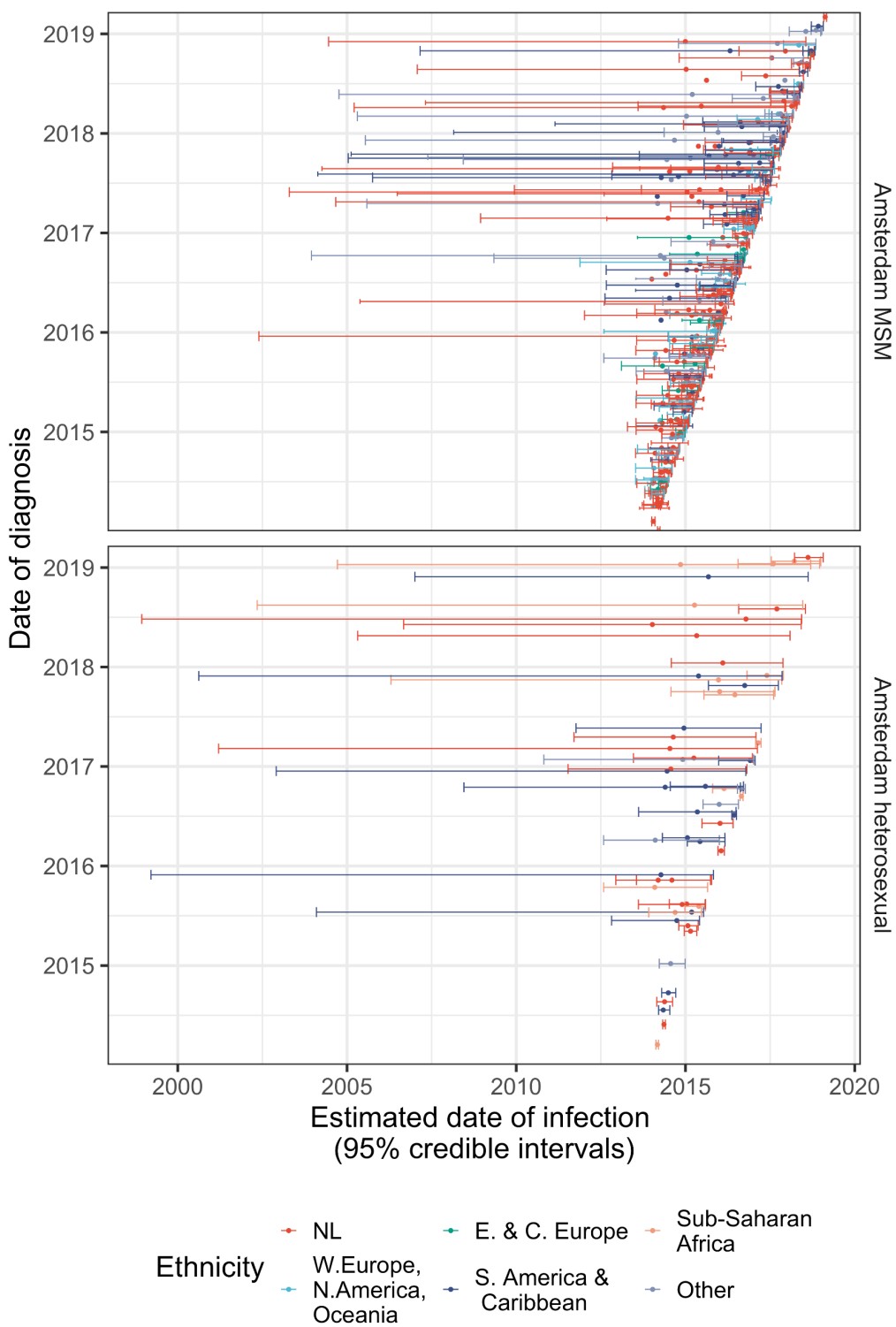

**Appendix 1—figure 2.** Diagnosis date and posterior median estimated infection date (with 95% credible interval) of individuals in Amsterdam diagnosed between January 2014 and May 2019.

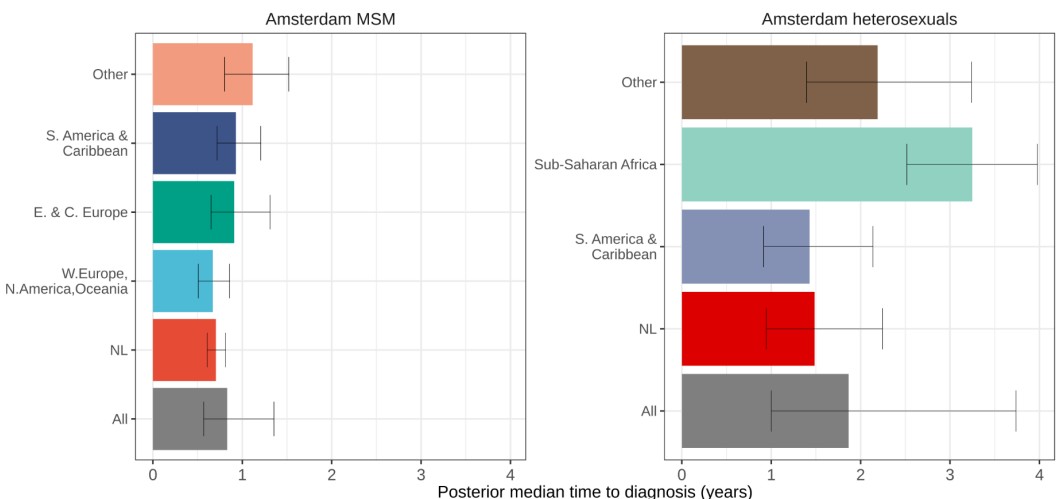

**Appendix 1—figure 3.** Posterior median estimated time to diagnosis (with 95% credible interval) of HIV infections in Amsterdam occurring in 2014-2018, stratified by risk group (MSM and heterosexuals) and place of birth. Estimates generated from time-to-diagnosis estimates for 535 MSM and 97 heterosexuals.

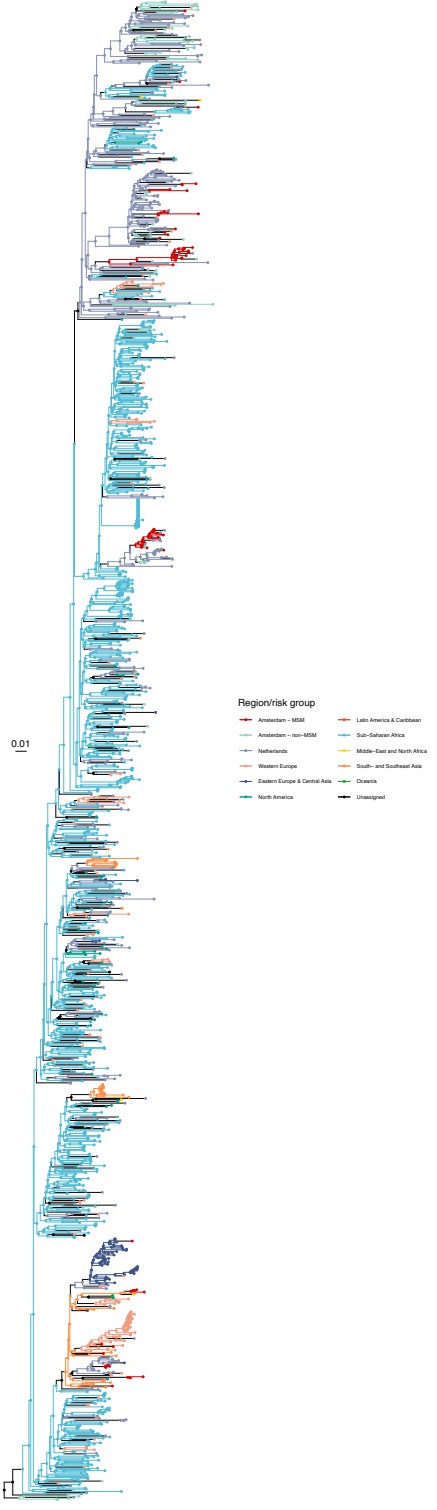

**Appendix 1—figure 4.** Annotated phylogeny of viral sequences of subtype A1 of Amsterdam MSM and background individuals. Colours of tips show the observed states of each observed sequence, and colours of lineages represent inferred states. States were assigned to each sequence as described in *Equation S22*, and represent both transmission group (MSM, non-MSM) and place of birth or residence.

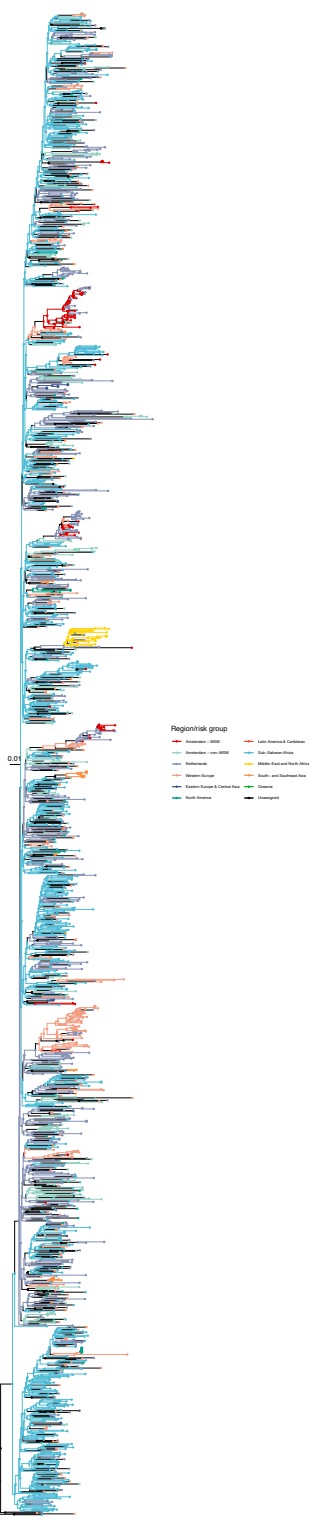

**Appendix 1—figure 5.** Annotated phylogeny of viral sequences of circulating recombinant form 02AG of Amsterdam MSM and background individuals. Colours of tips show the observed states of each observed
*Appendix 1—figure 5 continued on next page*

*Appendix 1—figure 5 continued*
sequence, and colours of lineages represent inferred states. States were assigned to each sequence as described in *Equation S22*, and represent both transmission group (MSM, non-MSM) and place of birth or residence.

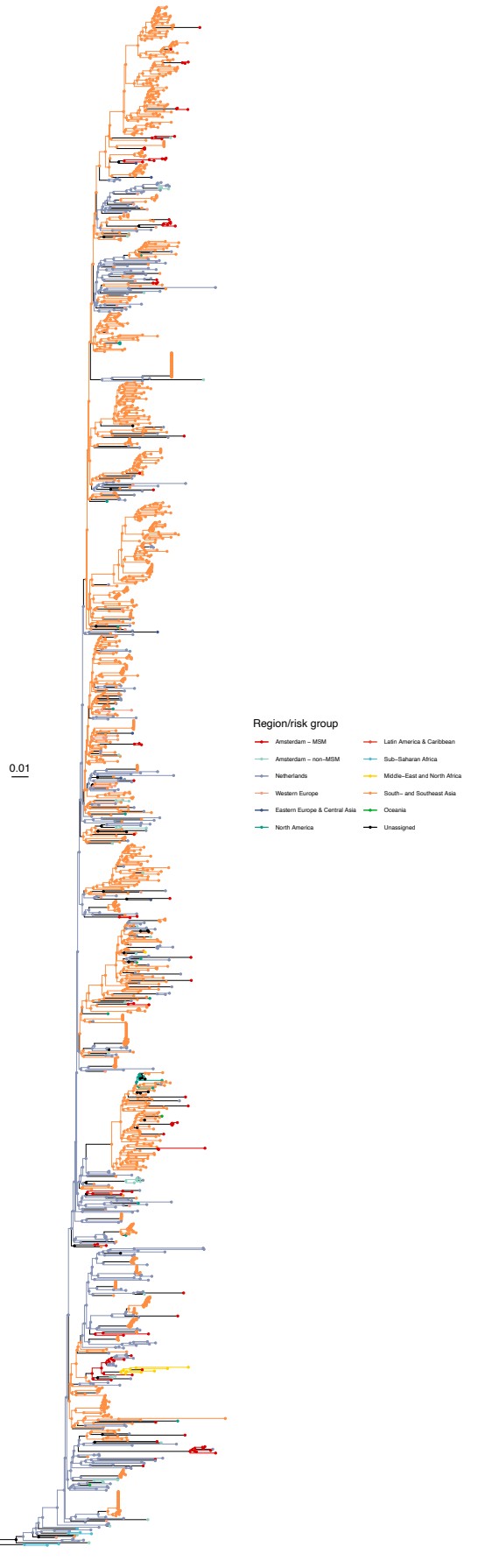

Region/risk group

- Amsterdam – MSM
- Amsterdam – non-MSM
- Netherlands
- Western Europe
- Eastern Europe & Central Asia
- North America
- Latin America & Caribbean
- Sub-Saharan Africa
- Middle-East and North Africa
- South- and Southeast Asia
- Oceania
- Unassigned

0.01

**Appendix 1—figure 6.** Annotated phylogeny of viral sequences of circulating recombinant form 01AE of Amsterdam MSM and background individuals. Colours of tips show the observed states of each observed sequence, and colours of lineages represent inferred states. States were assigned to each sequence as described in *Equation S22*, and represent both transmission group (MSM, non-MSM) and place of birth or residence.

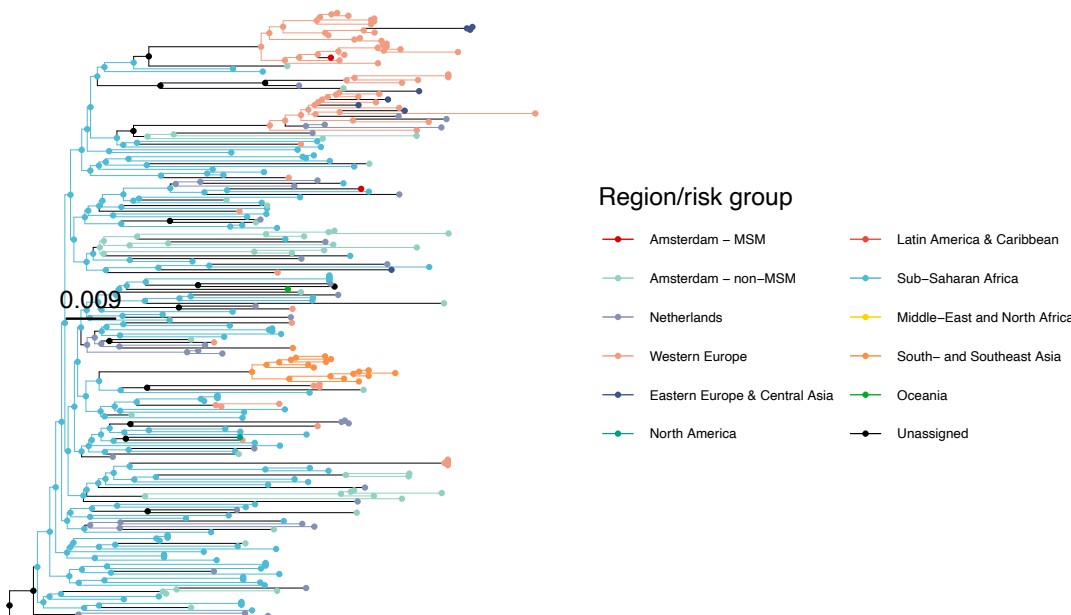

**Appendix 1—figure 7.** Annotated phylogeny of viral sequences of circulating recombinant form 06cpx of Amsterdam MSM and background individuals. Colours of tips show the observed states of each observed sequence, and colours of lineages represent inferred states. States were assigned to each sequence as described in *Equation S22*, and represent both transmission group (MSM, non-MSM) and place of birth or residence.

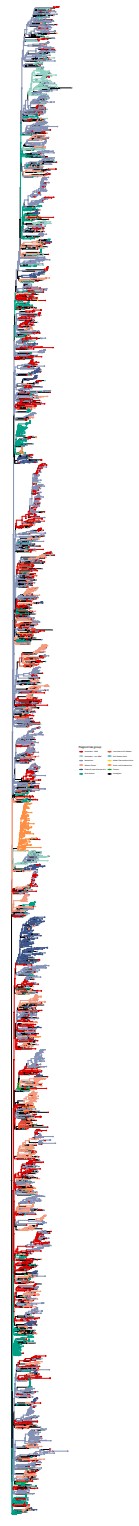

**Appendix 1—figure 8.** Annotated phylogeny of viral sequences of a sub-clade of subtype B of Amsterdam MSM and background individuals. Colours of tips show the observed states of each observed sequence, and colours of lineages represent inferred states. States were assigned to each sequence as described in *Equation S22*, and represent both transmission group (MSM, non-MSM) and place of birth or residence.

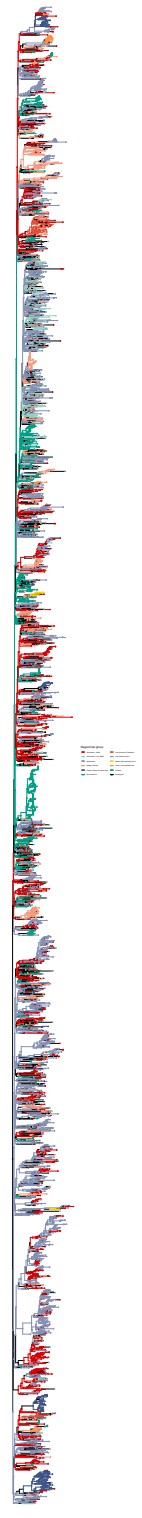

**Appendix 1—figure 9.** Annotated phylogeny of viral sequences of a sub-clade of subtype B of Amsterdam MSM and background individuals. Colours of tips show the observed states of each observed sequence, and colours of lineages represent inferred states. States were assigned to each sequence as described in *Equation S22*, and represent both transmission group (MSM, non-MSM) and place of birth or residence.

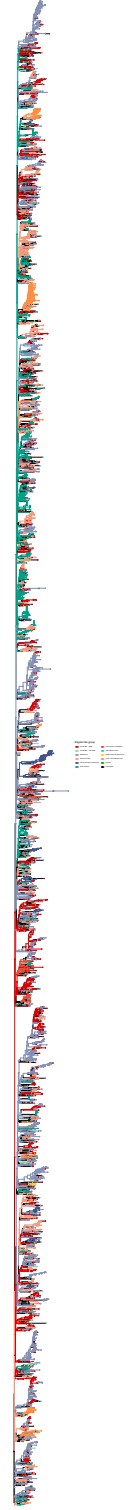

**Appendix 1—figure 10.** Annotated phylogeny of viral sequences of a sub-clade of subtype B of Amsterdam MSM and background individuals. Colours of tips show the observed states of each observed sequence, and colours of lineages represent inferred states. States were assigned to each sequence as described in *Equation S22*, and represent both transmission group (MSM, non-MSM) and place of birth or residence.

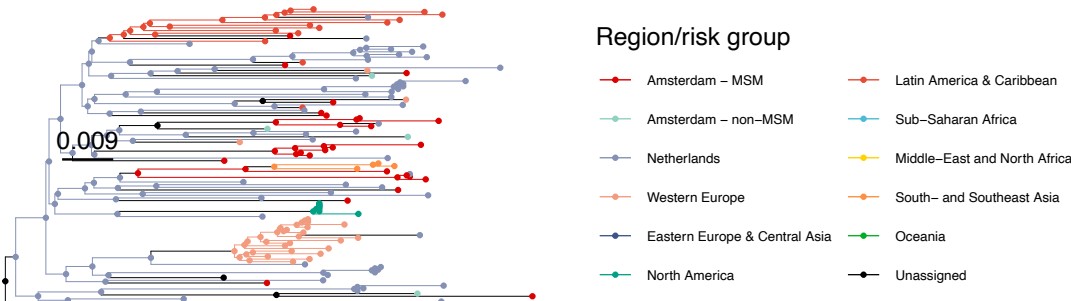

**Appendix 1—figure 11.** Annotated phylogeny of viral sequences of a sub-clade of subtype B of Amsterdam MSM and background individuals. Colours of tips show the observed states of each observed sequence, and colours of lineages represent inferred states. States were assigned to each sequence as described in *Equation S22*, and represent both transmission group (MSM, non-MSM) and place of birth or residence.

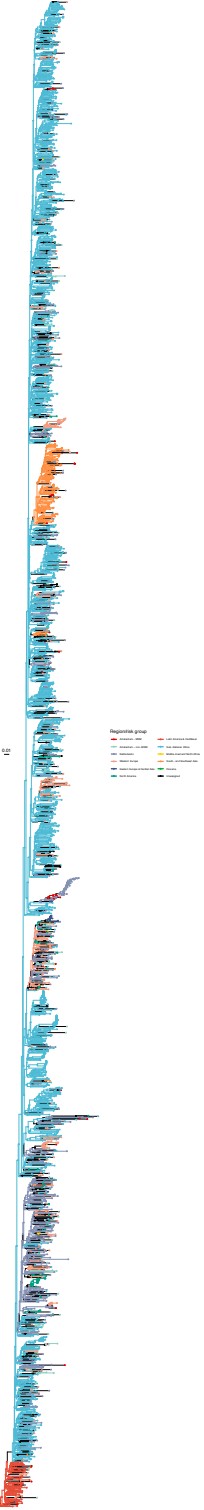

**Appendix 1—figure 12.** Annotated phylogeny of viral sequences of subtype C of Amsterdam MSM and background individuals. Colours of tips show the observed states of each observed sequence, and colours of lineages represent inferred states. States were assigned to each sequence as described in *Equation S22*, and represent both transmission group (MSM, non-MSM) and place of birth or residence.

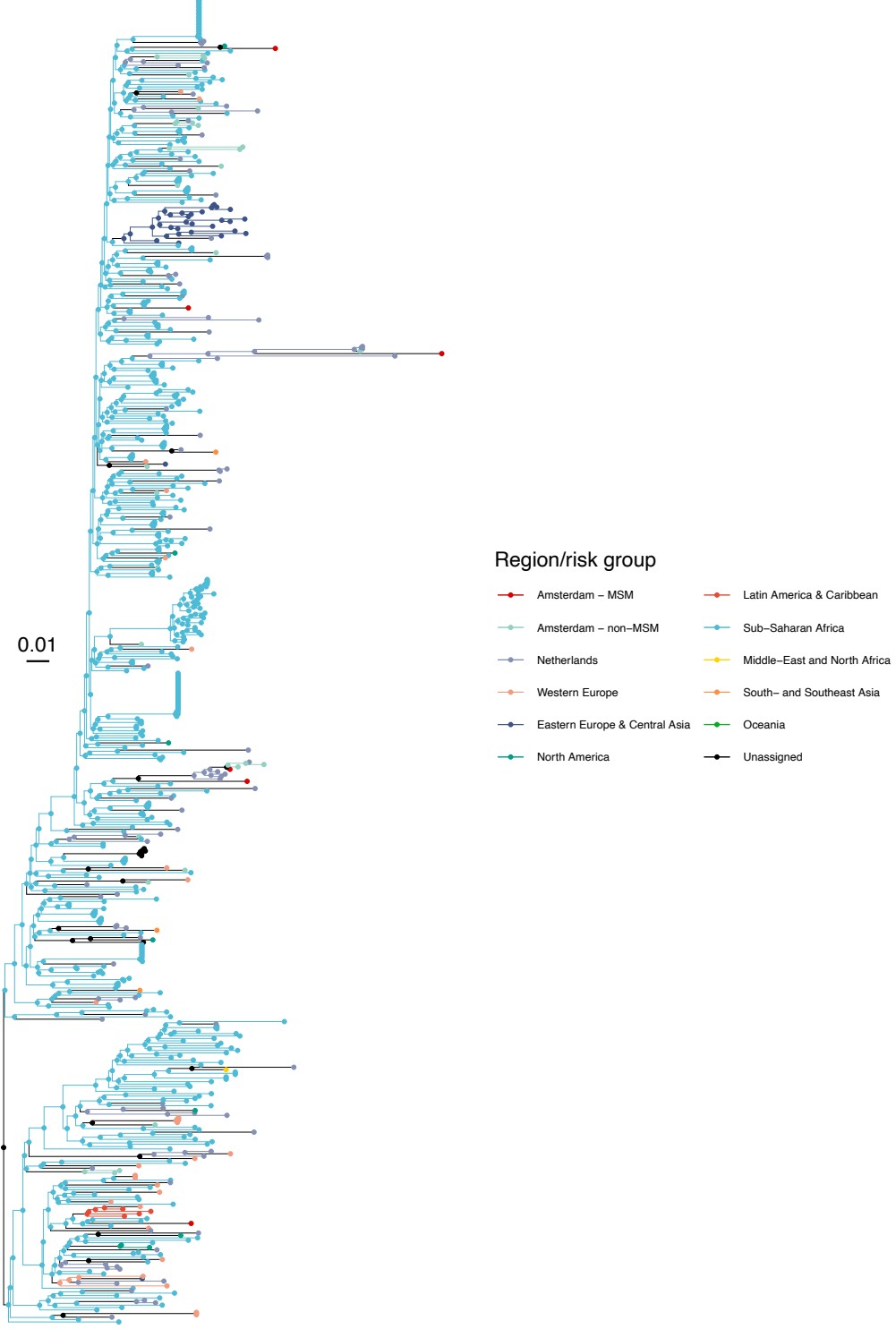

**Appendix 1—figure 13.** Annotated phylogeny of viral sequences of subtype D of Amsterdam MSM and background individuals. Colours of tips show the observed states of each observed sequence, and colours of lineages represent inferred states. States were assigned to each sequence as described in *Equation S22*, and represent both transmission group (MSM, non-MSM) and place of birth or residence.

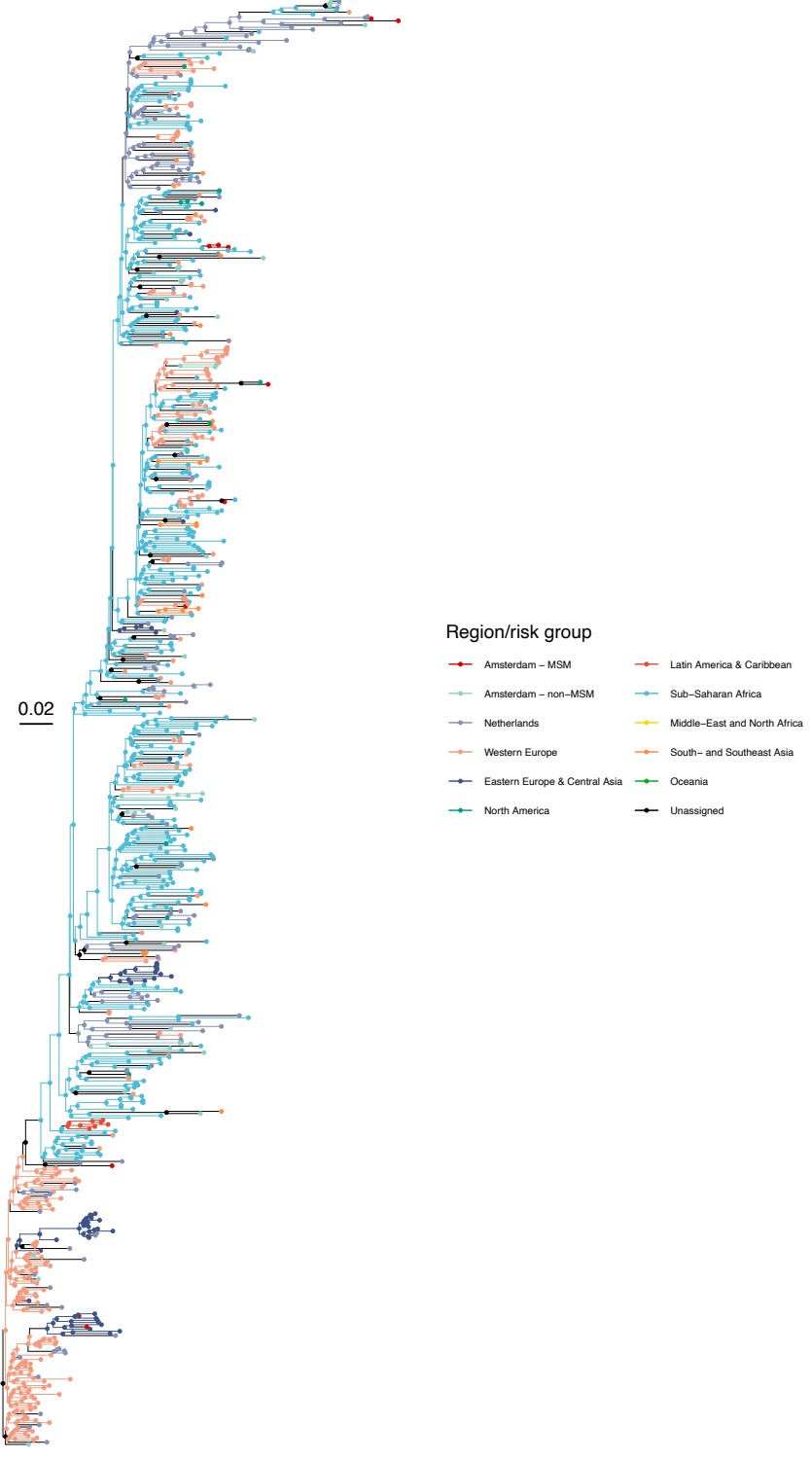

**Appendix 1—figure 14.** Annotated phylogeny of viral sequences of subtype G of Amsterdam MSM and background individuals. Colours of tips show the observed states of each observed sequence, and colours of lineages represent inferred states. States were assigned to each sequence as described in *Equation S22*, and represent both transmission group (MSM, non-MSM) and place of birth or residence.

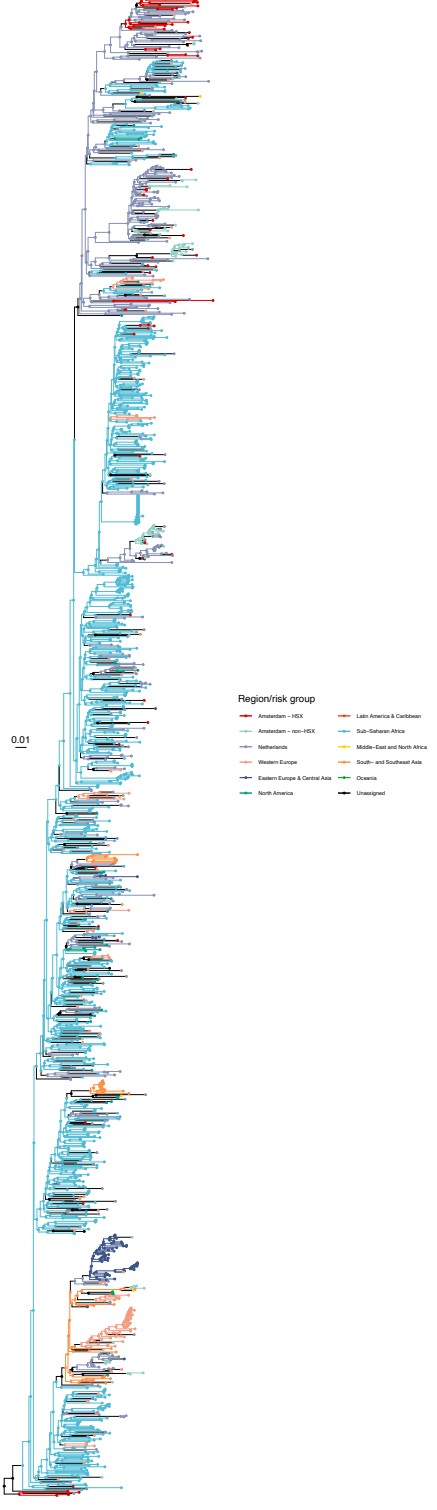

**Appendix 1—figure 15.** Annotated phylogeny of viral sequences of subtype A1 of Amsterdam heterosexual and background individuals. Colours of tips show the observed states of each observed sequence, and colours of lineages represent inferred states. States were assigned to each sequence as described in *Equation S23*, and represent both transmission group (heterosexual, non-heterosexual) and place of birth or residence.

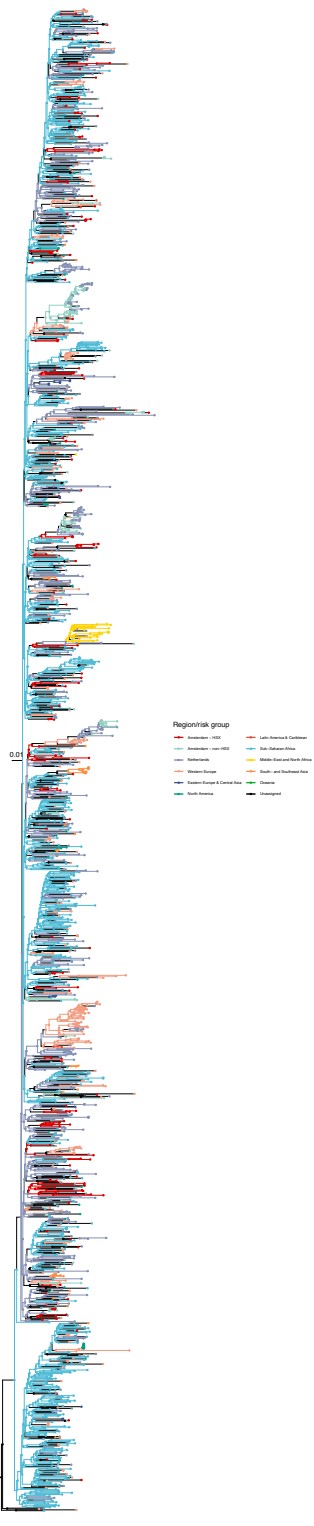

**Appendix 1—figure 16.** Annotated phylogeny of viral sequences of circulating recombinant form 02AG of Amsterdam heterosexual and background individuals. Colours of tips show the observed states of each observed
*Appendix 1—figure 16 continued on next page*

*Appendix 1—figure 16 continued*
sequence, and colours of lineages represent inferred states. States were assigned to each sequence as described in *Equation S23*, and represent both transmission group (heterosexual, non-heterosexual) and place of birth or residence.

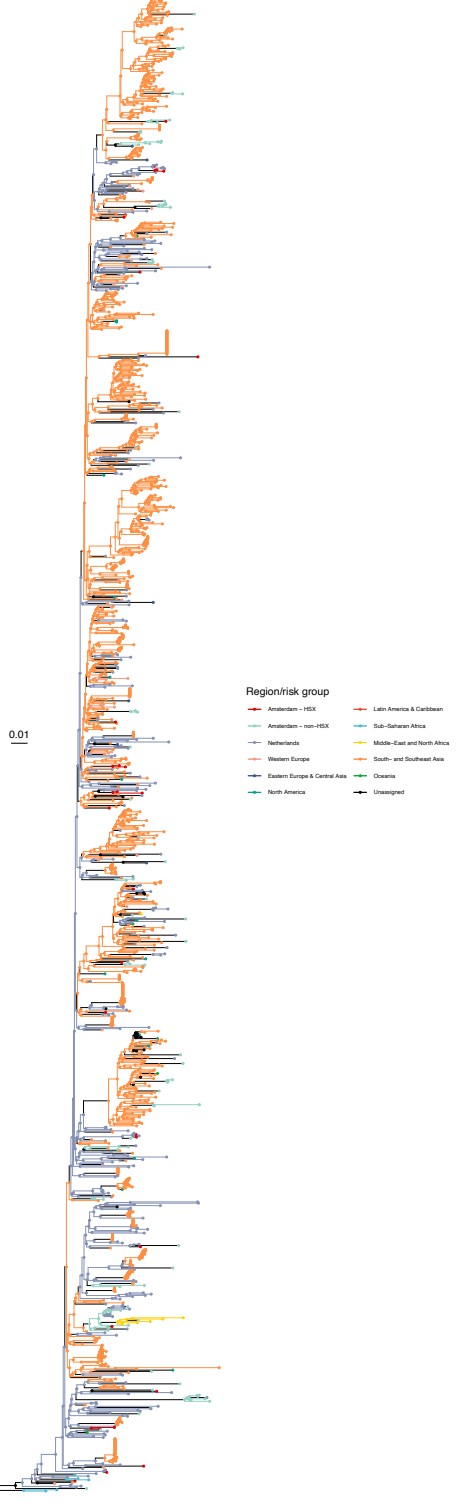

**Appendix 1—figure 17.** Annotated phylogeny of viral sequences of circulating recombinant form 01AE of Amsterdam heterosexual and background individuals. Colours of tips show the observed states of each observed sequence, and colours of lineages represent inferred states. States were assigned to each sequence as described in *Equation S23*, and represent both transmission group (heterosexual, non-heterosexual) and place of birth or residence.

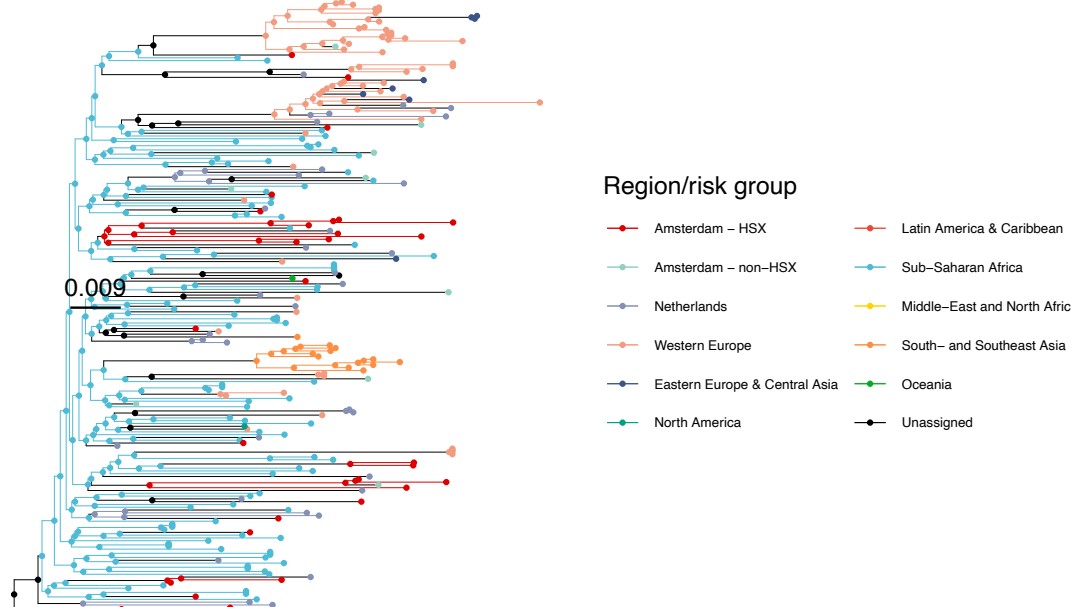

**Appendix 1—figure 18.** Annotated phylogeny of viral sequences of circulating recombinant form 06cpx of Amsterdam heterosexual and background individuals. Colours of tips show the observed states of each observed sequence, and colours of lineages represent inferred states. States were assigned to each sequence as described in *Equation S23*, and represent both transmission group (heterosexual, non-heterosexual) and place of birth or residence.

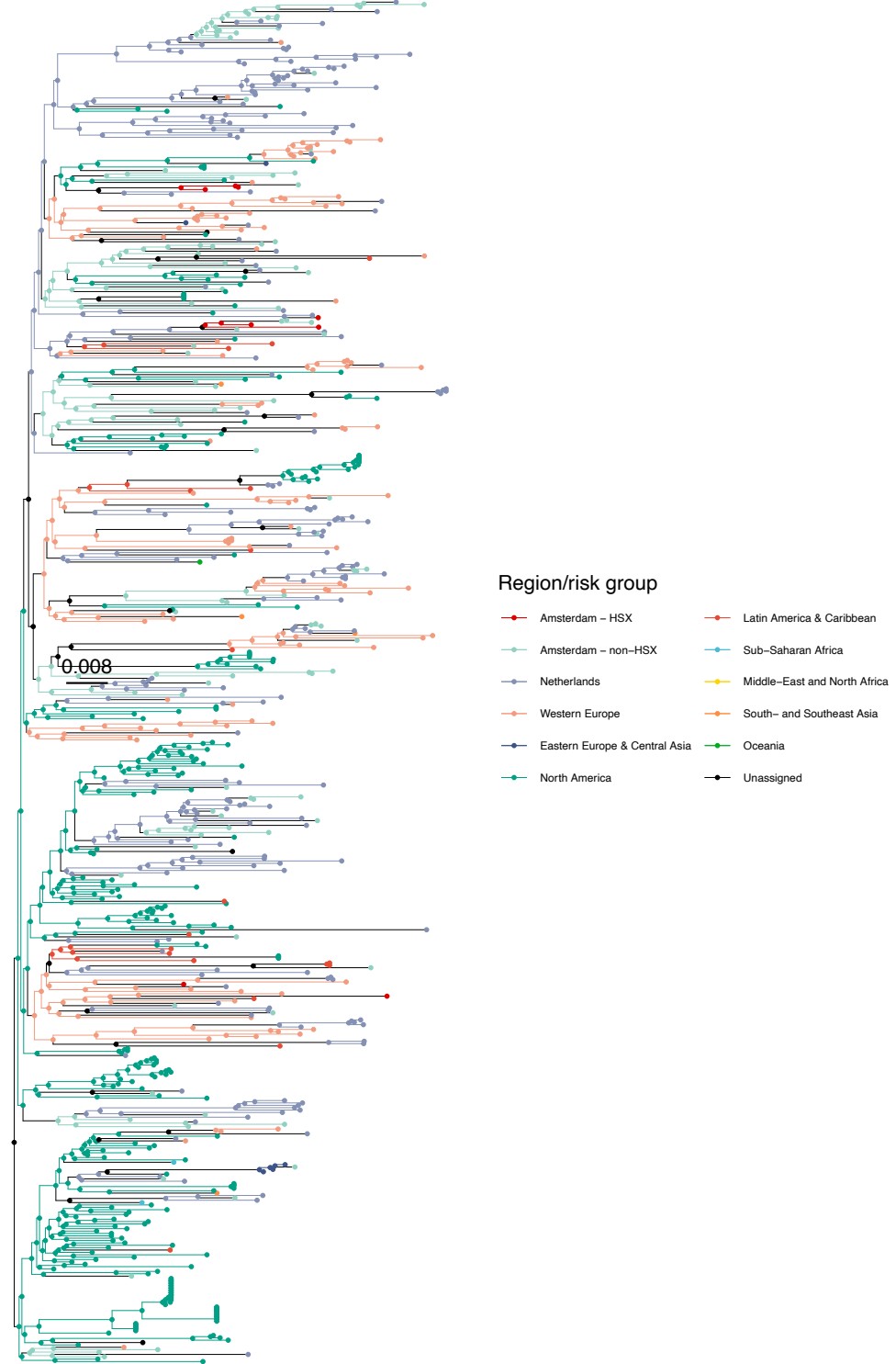

**Appendix 1—figure 19.** Annotated phylogeny of viral sequences of a sub-clade of subtype B of Amsterdam heterosexual and background individuals. Colours of tips show the observed states of each observed sequence, and colours of lineages represent inferred states. States were assigned to each sequence as described in *Equation S23*, and represent both transmission group (heterosexual, non-heterosexual) and place of birth or residence.

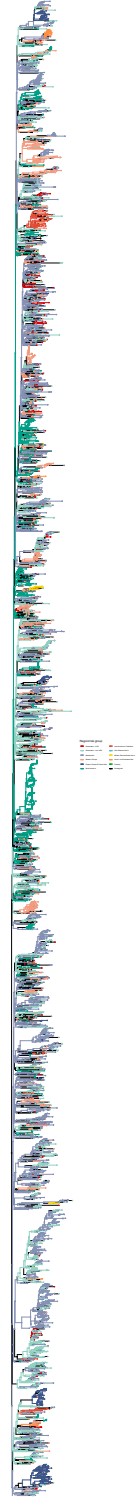

**Appendix 1—figure 20.** Annotated phylogeny of viral sequences of a sub-clade of subtype B of Amsterdam heterosexual and background individuals. Colours of tips show the observed states of each observed sequence, and colours of lineages represent inferred states. States were assigned to each sequence as described in *Equation S23*, and represent both transmission group (heterosexual, non-heterosexual) and place of birth or residence.

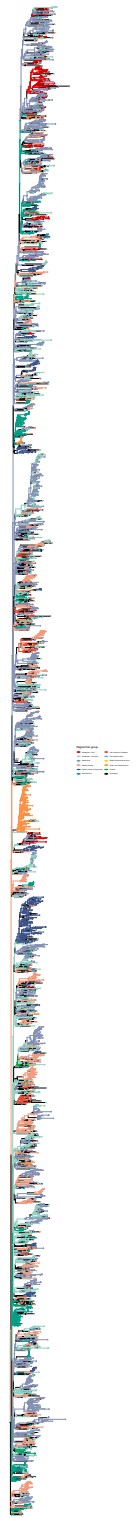

**Appendix 1—figure 21.** Annotated phylogeny of viral sequences of a sub-clade of subtype B of Amsterdam heterosexual and background individuals. Colours of tips show the observed states of each observed sequence, and colours of lineages represent inferred states. States were assigned to each sequence as described in *Equation S23*, and represent both transmission group (heterosexual, non-heterosexual) and place of birth or residence.

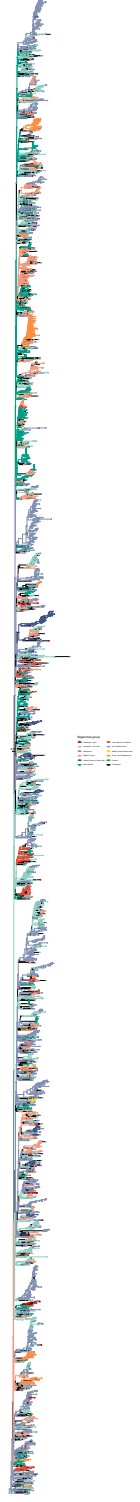

**Appendix 1—figure 22.** Annotated phylogeny of viral sequences of a sub-clade of subtype B of Amsterdam heterosexual and background individuals. Colours of tips show the observed states of each observed sequence, and colours of lineages represent inferred states. States were assigned to each sequence as described in *Equation S23*, and represent both transmission group (heterosexual, non-heterosexual) and place of birth or residence.

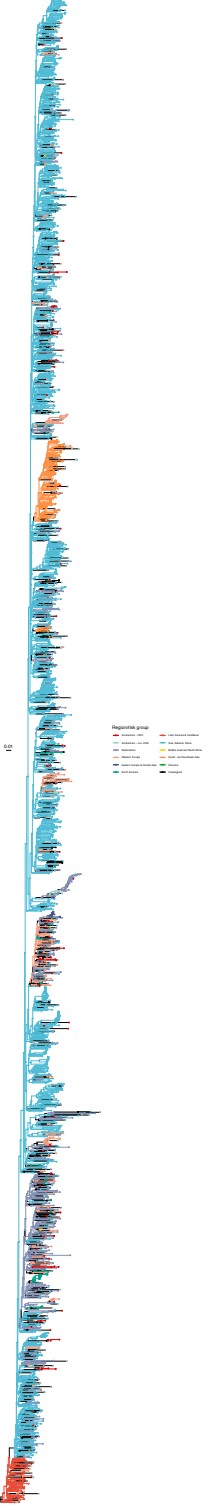

**Appendix 1—figure 23.** Annotated phylogeny of viral sequences of subtype C of Amsterdam heterosexual and background individuals. Colours of tips show the observed states of each observed sequence, and colours of lineages represent inferred states. States were assigned to each sequence as described in *Equation S23*, and represent both transmission group (heterosexual, non-heterosexual) and place of birth or residence.

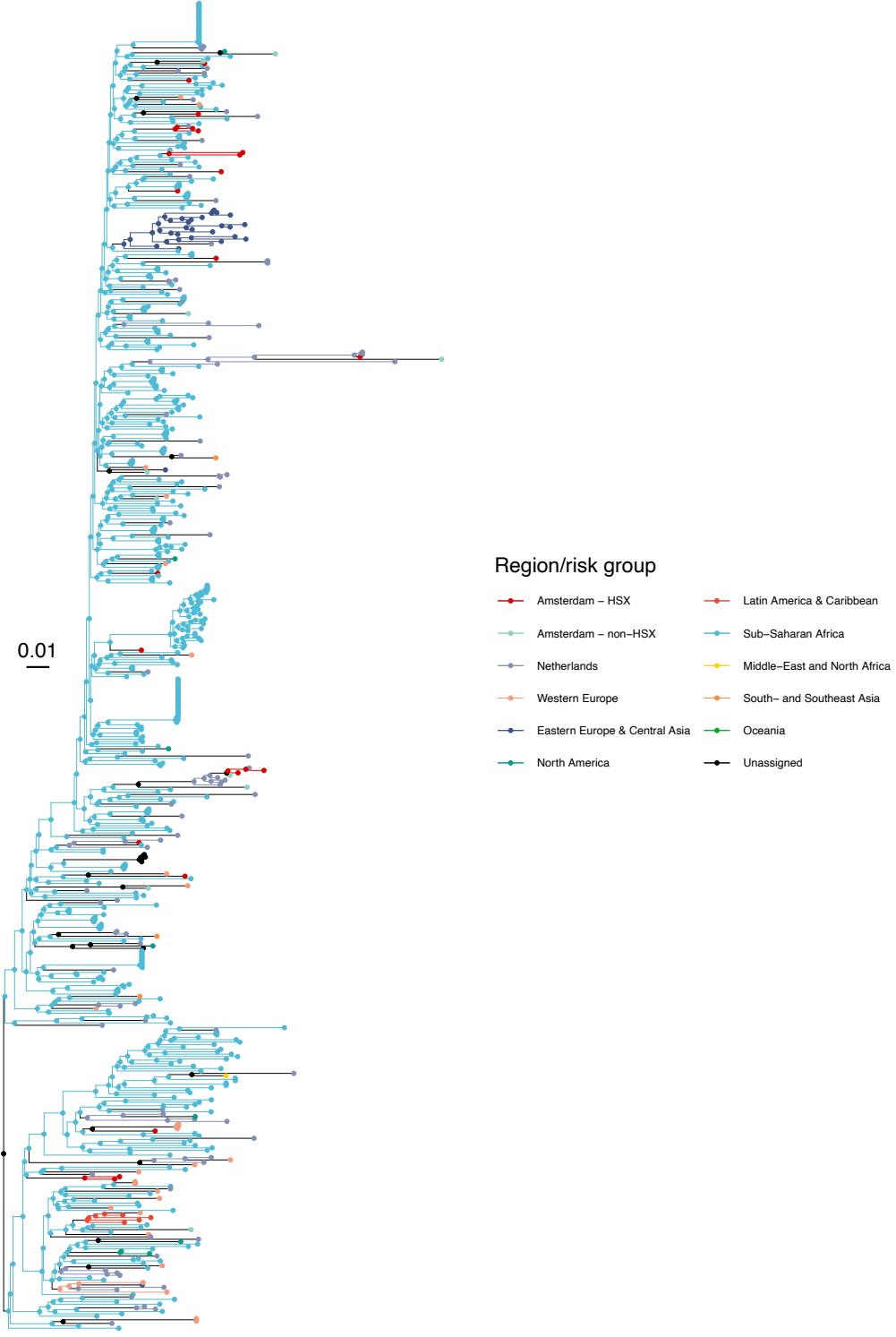

**Appendix 1—figure 24.** Annotated phylogeny of viral sequences of subtype D of Amsterdam heterosexual and background individuals. Colours of tips show the observed states of each observed sequence, and colours of lineages represent inferred states. States were assigned to each sequence as described in *Equation S23*, and represent both transmission group (heterosexual, non-heterosexual) and place of birth or residence.

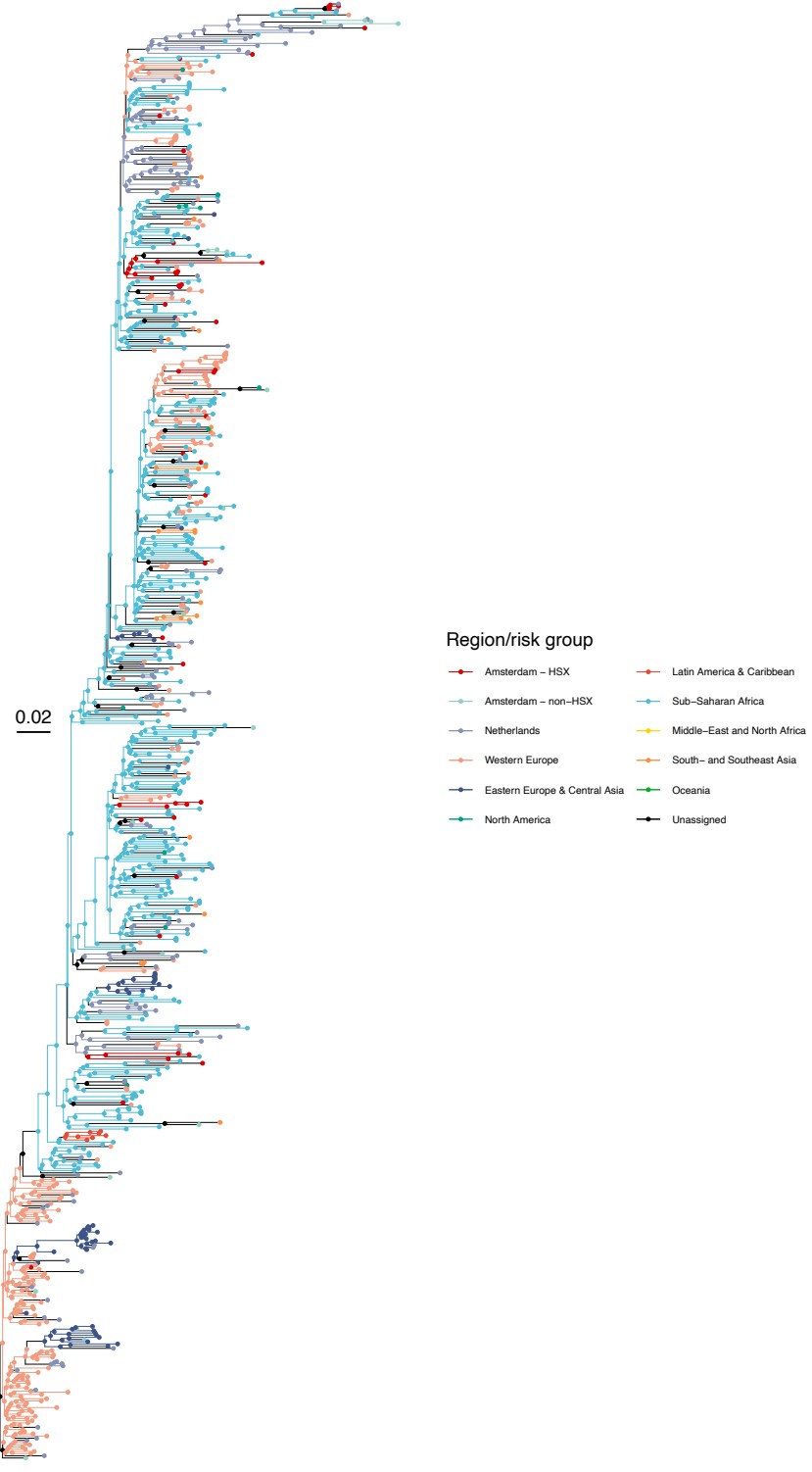

**Appendix 1—figure 25.** Annotated phylogeny of viral sequences of subtype G of Amsterdam heterosexual and background individuals. Colours of tips show the observed states of each observed sequence, and colours of *Appendix 1—figure 25 continued on next page*

*Appendix 1—figure 25 continued*

lineages represent inferred states. States were assigned to each sequence as described in *Equation S23*, and represent both transmission group (heterosexual, non-heterosexual) and place of birth or residence.

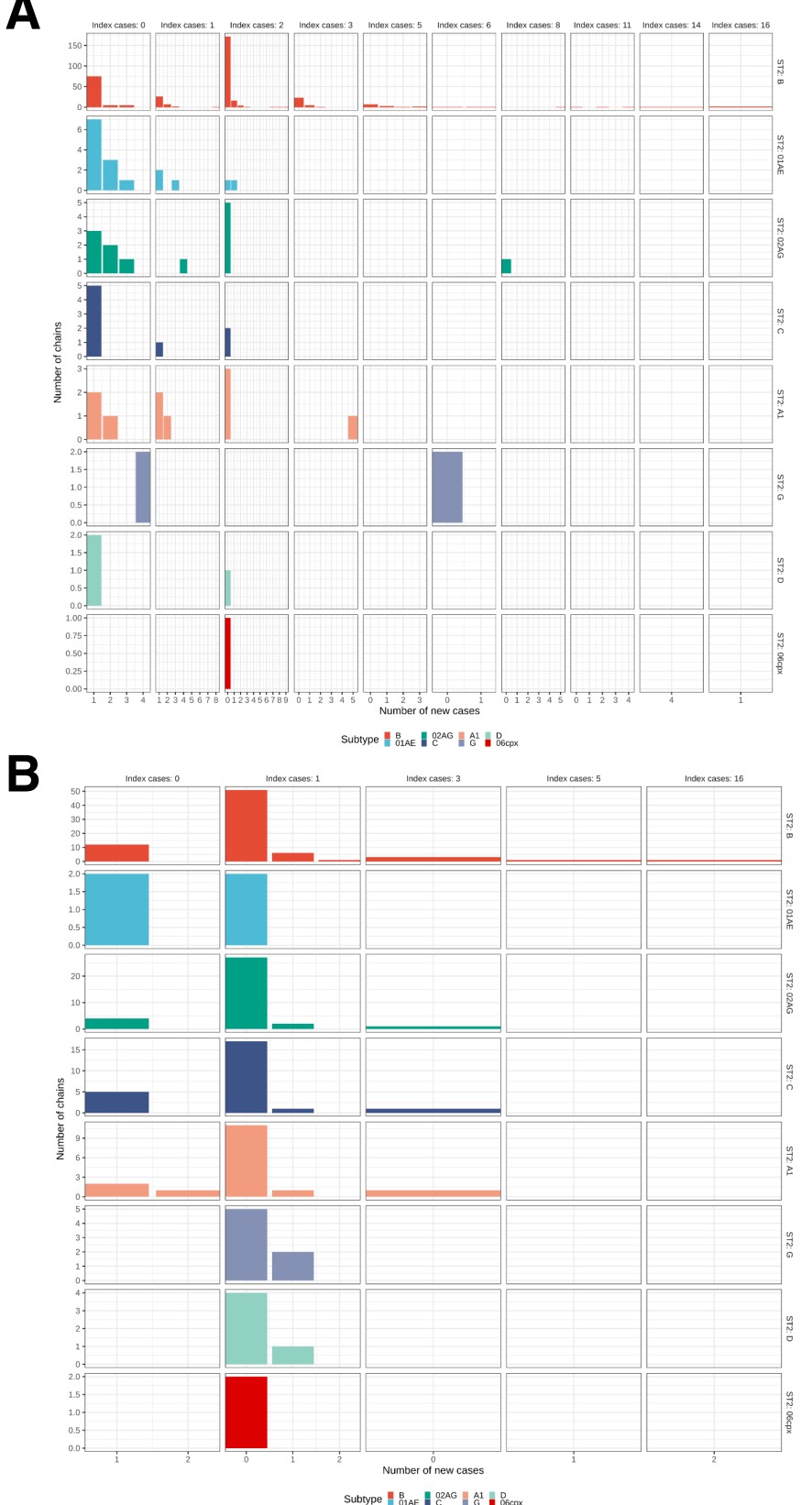

**Appendix 1—figure 26.** Growth of phylogenetically observed subgraphs by subtype. First column (index cases = 0) are for emergent chains, where the index case is among the newly generated cases. (**A**) Subgraphs among Amsterdam MSM. (**B**) Subgraphs among Amsterdam heterosexuals.

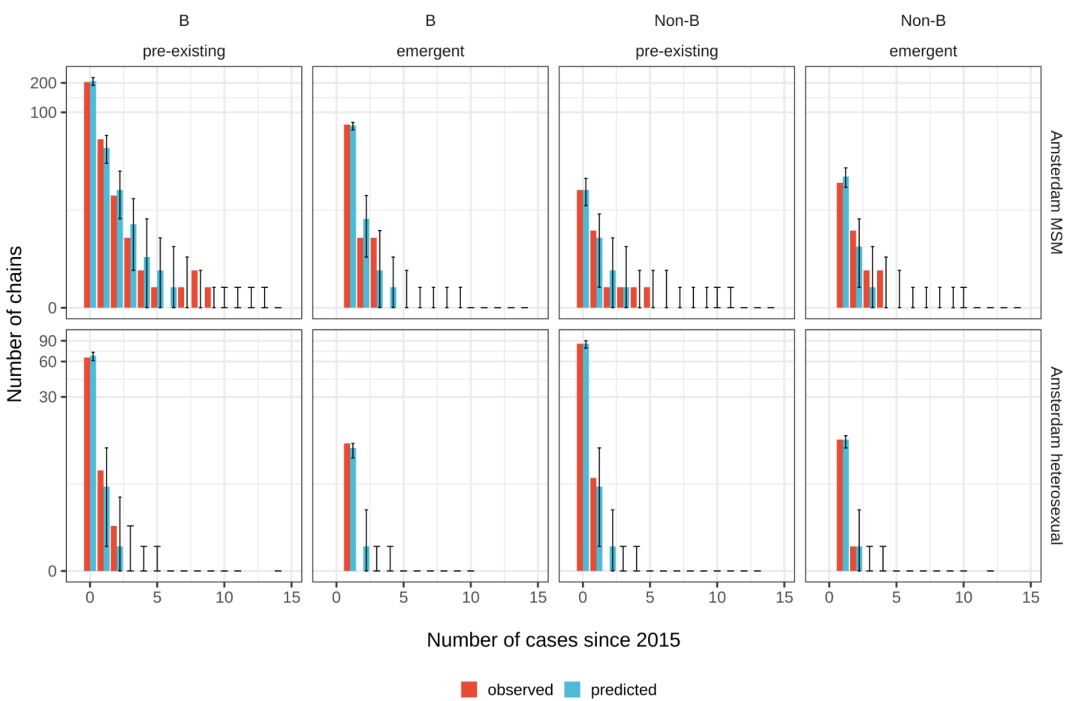

**Appendix 1—figure 27.** Posterior predictive check for Amsterdam MSM (top) and Amsterdam heterosexuals (bottom) for B and non-B subtypes. Estimates generated from 203 phylogenetic subgraphs among Amsterdam MSM, containing 297 individuals, and 41 subgraphs among Amsterdam heterosexuals, containing 44 individuals.

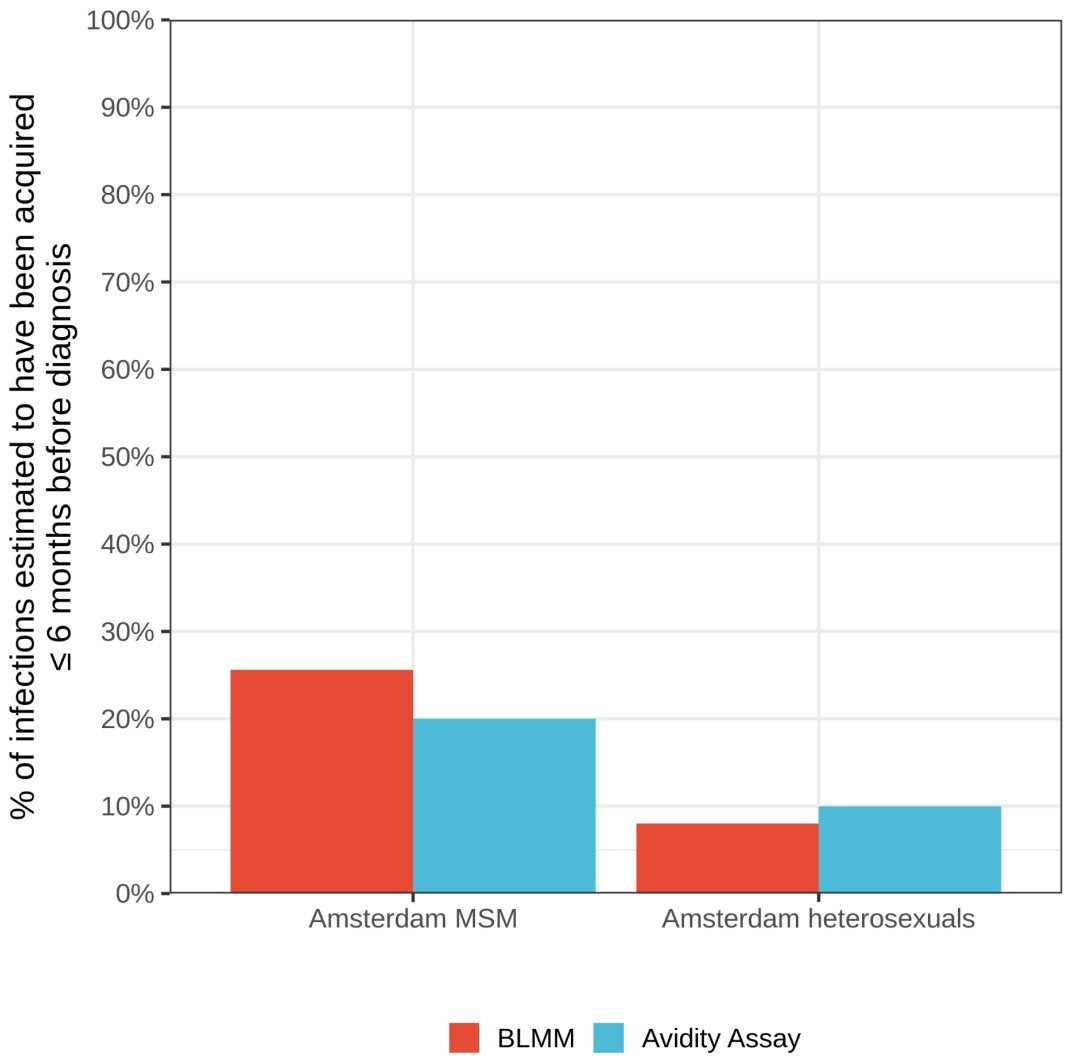

**Appendix 1—figure 28.** Estimates for the proportion of HIV infections acquired within 6 months of diagnosis from the bivariate linear mixed model (BLMM) approach (for infections diagnosed between 2013-2015), compared with estimates obtained from avidity assays in a study by *Slurink et al., 2021* (for infections diagnosed between 2013-2015).

Data were obtained from Stichting HIV Monitoring, collected as part of the open ATHENA cohort of all patients in care in the Netherlands. The dataset includes includes the municipal health service (GGD) region of the patient at the time of registration to the cohort, or at their most recent registration update, based on their the postcode of their place of residency (PC4 code) either at time of registration to the cohort, or at their most recent registration update. PC4 is the most granular administrative city level in Amsterdam, with 12,000 residents on average per PC4 area and a number of residents ranging from 10 to 26,263. *Appendix 1—figure 29* shows a map of the 81 Amsterdam PC4 areas. Amsterdam patients were identified as patients with a first or more recent registration in the Amsterdam GGD region.

The ATHENA database version was closed on March 31st 2019 (*Boender et al., 2018*). We obtained data for 19,204 patients from the Netherlands, with 7,773 of these having an Amsterdam postcode at first or last registration.

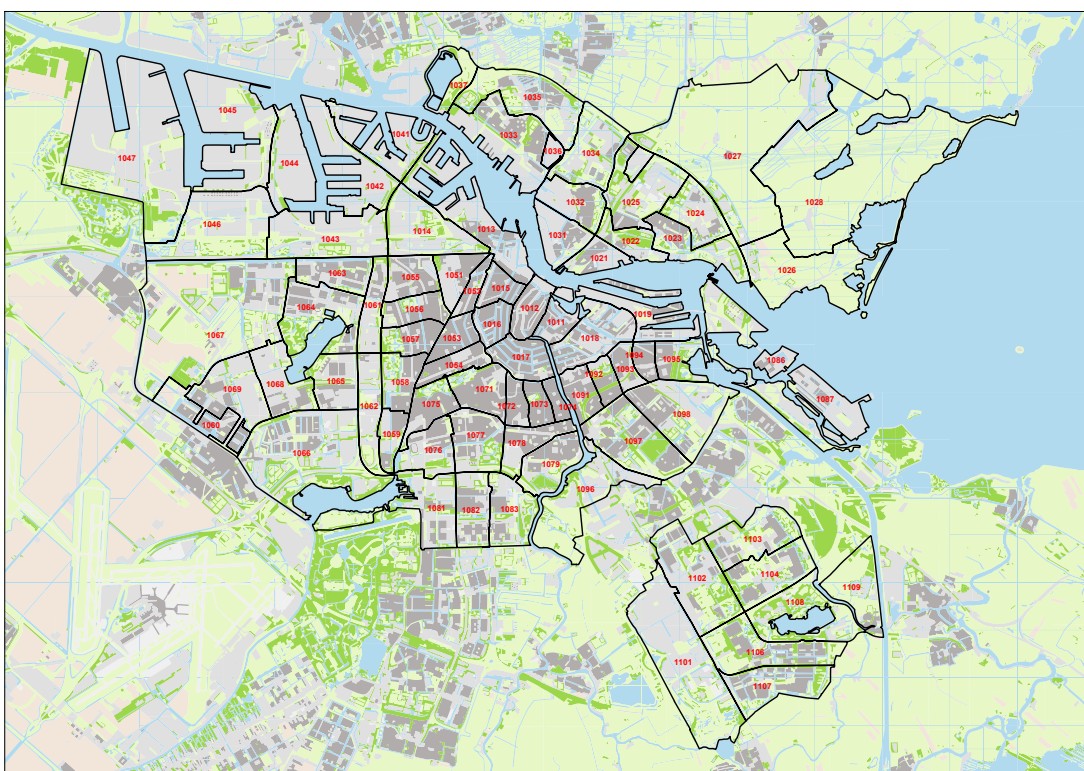

**Appendix 1—figure 29.** Map of Amsterdam postal code (PC4) areas.

We leverage baseline data recorded at registration on year of birth, country of birth, mode of transmission, date of death (if patient has died), date of AIDS diagnosis, date of ART start, date of last HIV negative test and date of first HIV positive test.

We also obtained datasets from the ATHENA cohort of partial HIV-1 polymerase (pol) sequences of Amsterdam patients, including date of sample, and of clinical data collected longitudinally of viral load measurements and CD4 counts.

In the study, we focus on infections estimated to have been acquired between 2014-2018 (see Section 3.1). We also consider MSM and heterosexual transmission groups only, since less than 2% of infections were in other transmission groups. Table 1 summarises patient characteristics for all Amsterdam individuals estimated to have been infected with HIV between 2014-2018, and those who have a viral sequence available. The cohort is predominantly male (92%), and MSM (86%). 41% of individuals were between 25-34 years old at their estimated time of infection. Less than 3% of individuals were estimated to have been infected aged 60 or older. 41% of individuals infected between 2014-2018 were born in the Netherlands, followed by 13% from South America and the Caribbean, which are predominantly individuals from Suriname and the Dutch Caribbean. *Appendix 1—table 1* also reports characteristics of patients with a viral sequence available. Empirically comparing only those with a sequence with the complete Amsterdam cohort of all individuals infected between 2014-2018, indicates that those patients with a sequence are representative of the whole diagnosed population.

For each transmission group, we define each strata by place of birth, according to the main migrant populations in Amsterdam. For Amsterdam MSM and heterosexuals, respectively, these are,

$$\mathcal{M} = \{\text{Netherlands; W.Europe, North America and Oceania; Eastern and Central Europe;} \\ \text{Latin America and the Caribbean; Other}\}, \tag{S1a}$$

$$\mathcal{H} = \{\text{Netherlands; Sub-Saharan Africa; Latin America and the Caribbean; Other}\}. \tag{S1b}$$

Since we focus on infections acquired between 2014-2018, we define the study start and end time by,

$$\psi_{\text{start}} = 2014, \tag{S2a}$$

$$\psi_{\text{end}} = 2018. \tag{S2b}$$

## Estimating HIV infection dates and undiagnosed infections

In this section, we first describe how we fit a model to clinical biomarker data to estimate the time from infection to diagnosis, and consequently the date of infection. Next, we describe how we fit a model to the posterior median estimates of the time-to-diagnosis, to estimate the proportion of Amsterdam infections which remained undiagnosed by the close of the study.

## Estimating HIV infection dates

### 3.1.1 Data

We define the complete cohort of patients registered in Amsterdam by $\mathcal{N}$. We first follow methods in *Pantazis et al., 2019* to estimate time from infection to diagnosis for individual $i \in \mathcal{N}$ by $w_i$. We use an indicator $R_i$ to denote transmission risk group of each individual, where,

$$R_i = \begin{cases} 1, & \text{if } i \text{ is Amsterdam MSM} \\ 0, & \text{if } i \text{ is Amsterdam heterosexual} \end{cases} \tag{S3}$$

We utilise clinical biomarker data for each patient on CD4 counts and viral loads, measured after diagnosis but before onset of AIDS or start of ART. As a caveat, we keep viral load measurements within one week of ART start, and CD4 counts within one month of ART start. This choice is supported by the fact that ART takes time to act. We denote CD4 counts by $y^c$, and viral loads by $y^r$, and encapsulate measurements for all $i$ individuals in a vector,

$$\mathbf{Y}_i^c = (y_{i1}^c, \ldots, y_{in_i^c}^c)^T \text{ and}$$
$$\mathbf{Y}_i^r = (y_{i1}^r, \ldots, y_{in_i^r}^r)^T. \tag{S4}$$

Each measurement is collected at an (unknown) time since infection,

$$\mathbf{t}_i^c = (t_{i1}^c, \ldots, t_{in_i^c}^c)^T \text{ and}$$
$$\mathbf{t}_i^r = (t_{i1}^r, \ldots, t_{in_i^r}^r)^T. \tag{S5}$$

We have clinical data prior to AIDS diagnosis or start of ART for 6,879 (88%) of patients. For the remaining 12% we are unable to estimate the time of infection. We then denote the time between diagnosis and each biomarker measurement by,

$$\mathbf{d}_i^c = (d_{i1}^c, \ldots, d_{in_i^c}^c)^T \text{ and}$$
$$\mathbf{d}_i^r = (d_{i1}^r, \ldots, d_{in_i^r}^r)^T. \tag{S6}$$

From this, we can then express the time from infection to measurement date in *Equation S5* in terms of the estimated date of infection, $w_i$, and the time between diagnosis and each biomarker measurement as follows,

$$t_{ij}^c = d_{ij}^c + w_i \text{ and}$$
$$t_{ij}^r = d_{ij}^r + w_i. \tag{S7}$$

### 3.1.2 Model

We then use a bivariate linear mixed model for the joint distribution of the two biomarkers over time and denote their distribution by,

$$f(\mathbf{y}_i^c, \mathbf{y}_i^r | \mathbf{t}_i^c, \mathbf{t}_i^r), \tag{S8}$$

for the joint distribution of the two biomarkers over time. We place a uniform prior on $w_i$ over $(0, u_i)$, where $u_i$ is the interval between time at risk for each individual and HIV diagnosis. We take the

risk onset date to be the maximum of the time between the last negative and test and diagnosis, and the time between the individual turning 15 years of age and diagnosis.

The posterior distribution of $w_i$ is as follows:

$$f(w_i|\mathbf{y}_i) = \frac{f(\mathbf{y}_i|w_i)f(w_i)}{\int_0^{u_i} f(\mathbf{y}_i|w_i)f(w_i)dw_i}, \tag{S9}$$

from which we estimate the median time from infection to diagnosis for $w_i$, and 95% credible intervals.

### 3.1.3 Estimated quantities

Then, if $T_i^{\text{diagnosis}}$ is the reported diagnosis date for individual $i$, we estimate their infection date, denoted by $T_i^{\text{infection}}$, with,

$$T_i^{\text{infection}} = T_i^{\text{diagnosis}} - w_i. \tag{S10}$$

*Appendix 1—figure 1* shows the distribution of individual median estimates for time-to-diagnosis by the risk groups given by *Equation (S1a) and (S1b)* for MSM and heterosexuals, respectively. Figure 1 plots the diagnosis date against the estimated infection date for all individuals diagnosed between 2014 and the May 2019. 95% credible intervals indicate uncertainty around individual level estimates from the model. We note that treatment guidelines changed in 2015 from starting ART based on CD4 count, which is measured every 6 months, to immediate ART initiation. Since we only consider biomarker measurements taken prior to ART start, as a result we have fewer biomarker measurements per individual for PLHIV diagnosed since 2015, which leads to larger uncertainty around date of infection.

### Estimating the proportion of infections in 2014-2018 that were undiagnosed by May 2019

### 3.2.1 Data

We next sought to estimate the proportion of infections in 2014-2018 that remained undiagnosed by May 2019. The patient data is right-censored, so many recent infections may yet be undiagnosed in the patient data set. For this reason, we considered the subset of Amsterdam diagnoses that we estimated to have been acquired between 2010 and the end of 2012, since most infections acquired in this interval would have been diagnosed by early 2019 given typical disease progression (*Pantaleo et al., 1993*). We first define an indicator $U_i(\tau)$, which is a function of a given year $\tau$, in which,

$$U_i(\tau) = \begin{cases} 1, & \text{if } T_i^{\text{infection}} < \tau \\ 0, & \text{otherwise.} \end{cases} \tag{S11}$$

We then define the synthetic cohort of infections in 2010-2012 by S12.

$$\mathcal{C}^{\text{MSM}} \subseteq \mathcal{N} : R_i = 1 \cap U_i(2010.0) = 0 \cap U_i(2013.0) = 1,$$
$$\mathcal{C}^{\text{HSX}} \subseteq \mathcal{N} : R_i = 0 \cap U_i(2010.0) = 0 \cap U_i(2013.0) = 1. \tag{S12}$$

We then consider individuals $k \in \mathcal{C}^{\text{MSM}}$ and $l \in \mathcal{C}^{\text{HSX}}$. For each transmission group, we defined each strata by place of birth given in *Equation (S1a) and (S1b)*. *Appendix 1—table 6* shows the characteristics of patients used to fit the model.

*Appendix 1—table 6 Continued on next page*

**Appendix 1—table 6.** Patient characteristics for individuals with an estimated infection date between 2010-2012.

| Risk group | Place of birth | Amsterdam infections 2010-2012 | Median estimated time to diagnosis (years) [95% quantiles] |
|---|---|---|---|
| Amsterdam MSM | W.Europe, N.America, Oceania | 72 | 0.42 [0.05-3.41] |
| | E. & C. Europe | 31 | 0.88 [0.13-6.04] |
| | S. America & Caribbean | 81 | 1.04 [0.05-5.57] |
| | Netherlands | 295 | 0.56 [0.04-4.77] |
| | Other | 56 | 1.38 [0.12-4.97] |
| | All | 535 | 0.64 [0.04-4.97] |
| Amsterdam heterosexual | Sub-Saharan Africa | 35 | 3.86 [0.33-6.8] |
| | S. America & Caribbean | 22 | 1.37 [0.14-5.68] |
| | Netherlands | 27 | 1.42 [0.07-6.16] |
| | Other | 13 | 1.6 [0.99-6.12] |
| | All | 97 | 2.22 [0.1-6.67] |

### 3.2.2 Hierarchical model

We fit a hierarchical Weibull model to the estimated times from infection to diagnosis ($w_i$) in Stan, for MSM and heterosexuals separately. For MSM, we denote the function $j(k)$ which takes as value the place of birth of individual $k$, as defined in **equation (S1a)**. We estimate ethnicity-specific shape and scale parameters $\kappa_{j(k)\in\mathcal{M}}$ and $\lambda_{j(k)\in\mathcal{M}}$ which can borrow information from each other through a hierarchical prior distribution. For convenience when choosing priors, we re-parameterised the Weibull distribution in terms of its median and 80% quantile ($\log \chi_{j(k)}^{50}$, $\log \chi_{j(k)}^{80} - \log \chi_{j(k)}^{50}$). The quantile function for the Weibull distribution is given by Equation (S13).

$$Q(p; \kappa_{j(k)}, \lambda_{j(k)}) = \lambda_{j(k)}(-\log(1-p))^{1/\kappa_{j(k)}}. \tag{S13}$$

We then express the parameters of the Weibull distribution as follows:

$$\kappa_{j(k)} = \frac{\log(\log(5)-\log(2))}{\log \chi_{j(k)}^{80} - \log \chi_{j(k)}^{50}},$$
$$\lambda_{j(k)} = \exp\left(\log(\chi_{j(k)}^{50}) - \frac{\log(\log(2))}{\kappa_{j(k)}}\right), \tag{S14}$$

and then specify the Weibull model and its prior distribution as follows,

$$w_{j(k)} \sim \text{Weibull}\left(\frac{\log(\log(5)-\log(2))}{\log \chi_{j(k)}^{80} - \log \chi_{j(k)}^{50}}, \exp\left(\log(\chi_{j(k)}^{50}) - \frac{\log(\log(2))}{\kappa_{j(k)}}\right)\right), \tag{S15a}$$

$$\log(\chi_{j(k)}^{50}) \sim N(\mu_{\log \chi^{50}}, \sigma_{\log \chi^{50}}^2) \tag{S15b}$$

$$\log(\chi_{j(k)}^{80}) - \log(\chi_{j(k)}^{50}) \sim N(\mu_{\log \chi^{80}-\log \chi^{80}}, \sigma_{\log \chi^{80}-\log \chi^{50}}^2) \tag{S15c}$$

$$\mu_{\log \chi^{50}} \sim N(\log(Q(0.5)), 0.5) \tag{S15d}$$

$$\mu_{\log \chi^{80}-\log \chi^{50}} \sim N(\log(Q(0.5)) - \log(Q(0.8)), 0.5) \tag{S15e}$$

$$\sigma_{\log \chi^{50}} \sim \text{Exp}(2) \tag{S15f}$$

$$\log(\sigma_{\log \chi^{80}-\log \chi^{50}}) \sim N(0, 1), \tag{S15g}$$

where $Q(0.5)$ and $Q(0.8)$ are the empirical quantiles from the estimated times to diagnosis, for each transmission group.

The joint posterior distribution of model *Equation S15a* was estimated in rstan with Stan version 2.21 using three Hamiltonian Monte Carlo chains with 2000 samples each including a warmup of 500 samples. The models mixed well, with no correlation between parameters, and had at most one divergence (*Appendix 1—figures 30–33*). The smallest effective sample size across parameters for the MSM model was 1461 and 1059 for the heterosexual model.

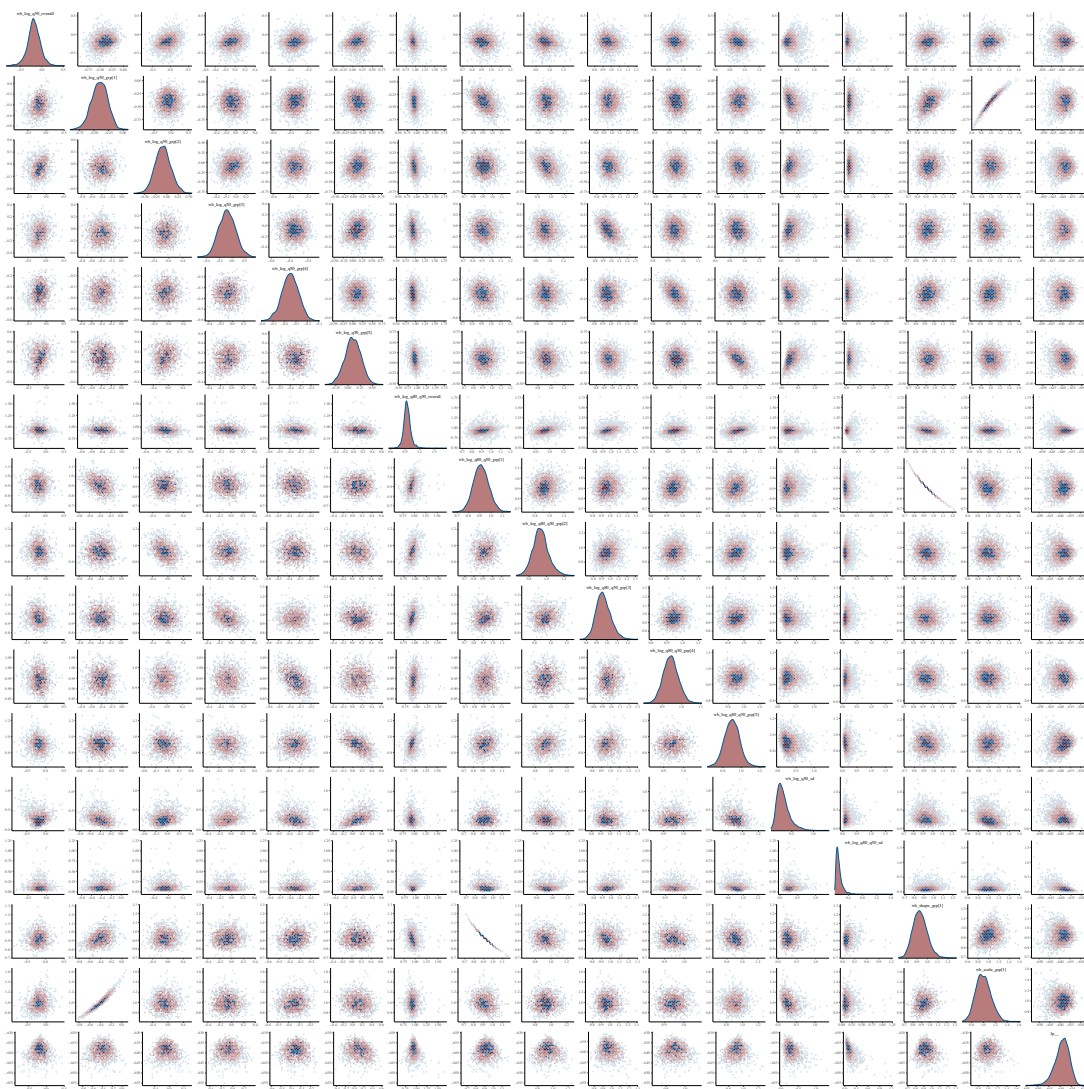

**Appendix 1—figure 30.** Pairs plot of the joint posterior density of the model parameters for MSM time-to-diagnosis model.

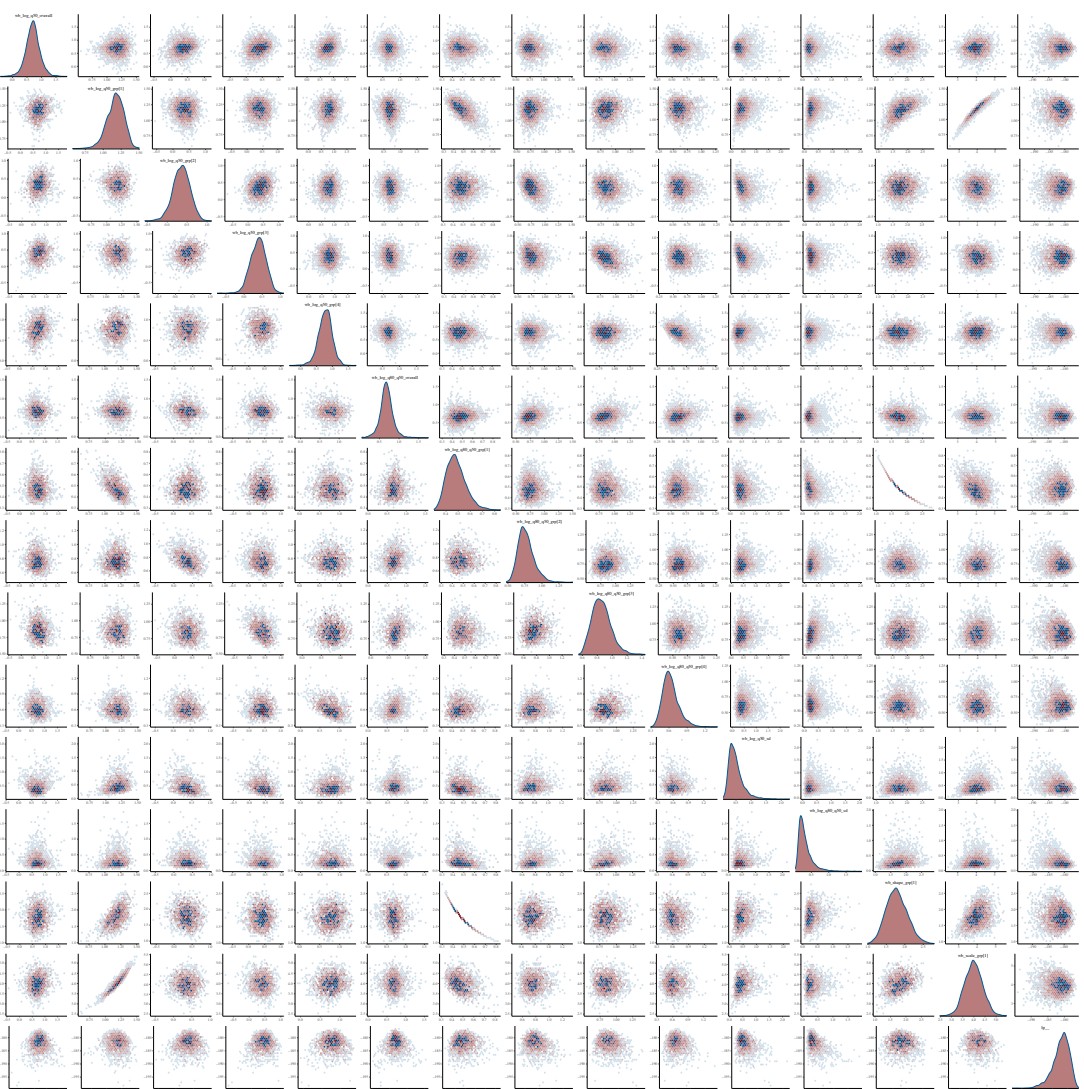

**Appendix 1—figure 31.** Pairs plot of the joint posterior density of the model parameters for heterosexual time-to-diagnosis model.

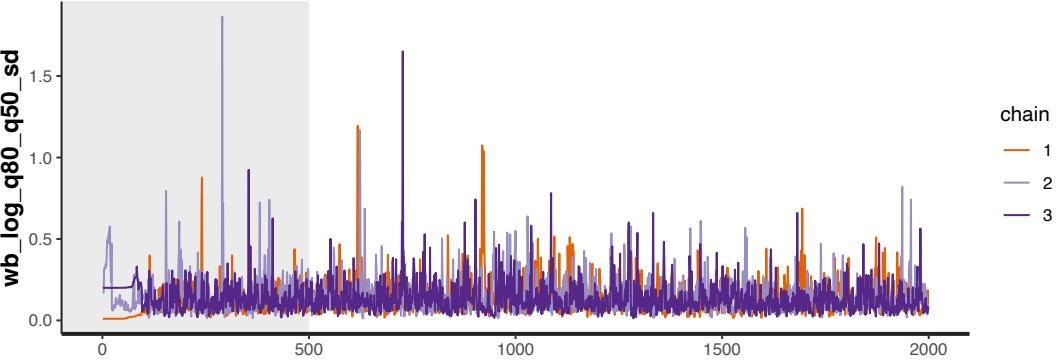

**Appendix 1—figure 32.** Trace plot for parameter with smallest effective sample size in MSM time-to-diagnosis model.

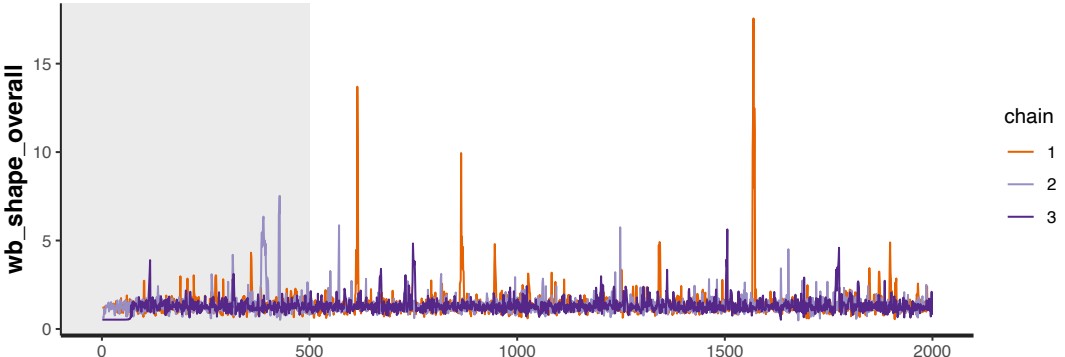

**Appendix 1—figure 33.** Trace plot for parameter with smallest effective sample size in heterosexual time-to-diagnosis model.

### 3.2.3 Estimated quantities

We then estimate, for a given year $y \in \mathcal{Y} = 2014, \ldots, 2018$ of infection, the probability of an MSM individual not being diagnosed by 2019, given their place of birth, as follows:

$$\delta_{j(k),y}^{(k)} = 1 - P\left(w_{j(k)} \leq (2019 - y) \, | \kappa_{j(k)}^{(k)}, \lambda_{j(k)}^{(k)}\right). \tag{S16}$$

To account for changes in infection rate over the study period, we generate weights $\omega_y$ for each year using the distribution of total infections among MSM by year estimated by the ECDC modelling tool (**Stockholm: European Centre for Disease Prevention and Control, 2017**),

$$\omega_y = \frac{N_y^{Inf-ECDC}}{\sum_{z \in \mathcal{Y}} N_z^{Inf-ECDC}}, \tag{S17}$$

where $N_y^{Inf-ECDC}$ are the estimated total infections acquired among MSM in Amsterdam in year $y$. We then calculate the average probability that an individual infected in 2014-2018 remained undiagnosed by 2019 with

$$\delta_{j(k)}^{(k)} = \sum_{y \in \mathcal{Y}} \omega_y \delta_{j(k),y}^{(k)}. \tag{S18}$$

We then denote the number of diagnosed Amsterdam MSM infected in 2014-2018 and born in world region $j(k)$ by $N_{j(k)}^D$. Finally, we can estimate the total number of infections in 2014-2018 in Amsterdam MSM through,

$$N^{Inf(k)} = \sum_{j(k) \in \mathcal{M}} \frac{N_{j(k)}^D}{1 - \delta_{j(k)}^{(k)}}, \tag{S19}$$

and obtain numerical estimates of $N^{Inf(k)}$ via the Monte Carlo samples from the joint posterior and the calculated proportions $\delta_{j(k)}^{(k)}$ of undiagnosed infections. Poster median estimates and 95% credible intervals of (S16)-(S19) are obtained by summarising the set of Monte Carlo samples after the transformations. The model for heterosexuals is formulated analogously.

**Appendix 1—figure 34** shows the estimated Weibull distributions for the time to diagnoses, stratified by MSM and heterosexuals and place of birth. The empirical cumulative distribution functions (CDFs) of the times to diagnoses are for comparison shown as step functions (black). The fits were good, with the empirical CDFs generally lying within the 95% posterior intervals of the fitted CDFs for all risk groups. **Appendix 1—figure 35** summarises the total number of estimated infections acquired between 2014 and 2018, the subset of those that were diagnosed by 2019, and the subset of those which have a viral sequence available. The sequence sampling fraction is shown above each bar.

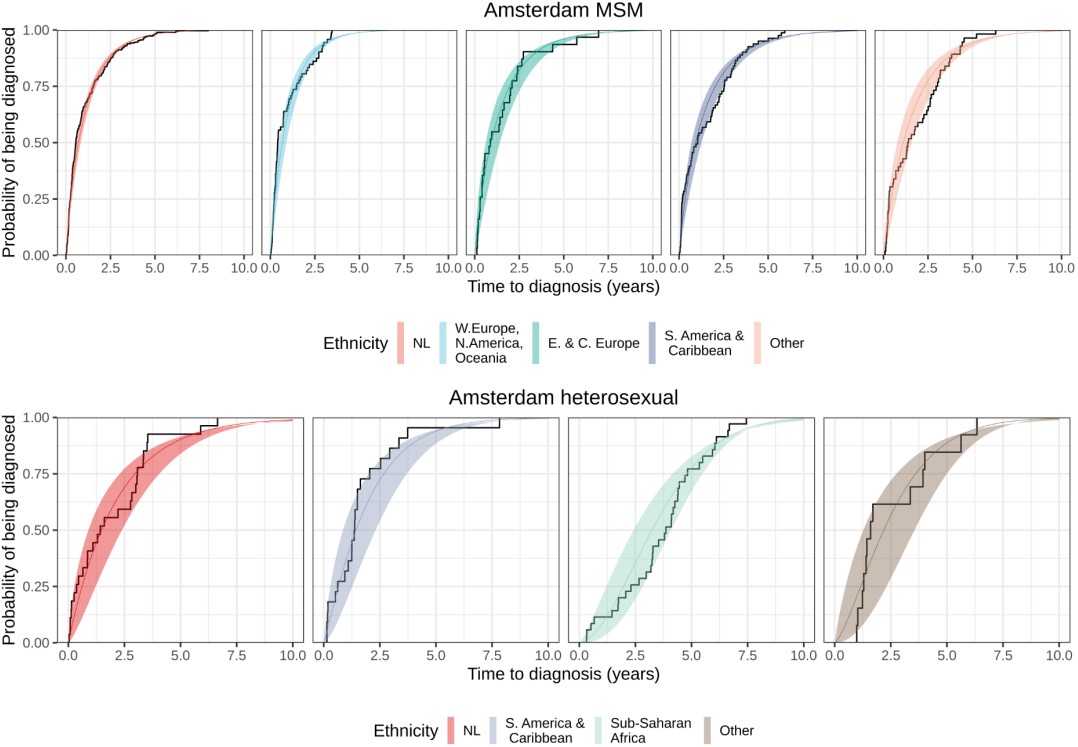

**Appendix 1—figure 34.** Posterior median cumulative distribution functions (CDFs) (line in colours) and 95% credible intervals (ribbon in colours) are shown along with the empirical CDF (steps in black).

*Appendix 1—figure 35* shows the total estimated number of infections acquired between 2014 and 2018 among Dutch-born and foreign-born individuals in Amsterdam, by risk group, alongside the number of infections by date of diagnosis, and the number and proportion of those with a sequence available.

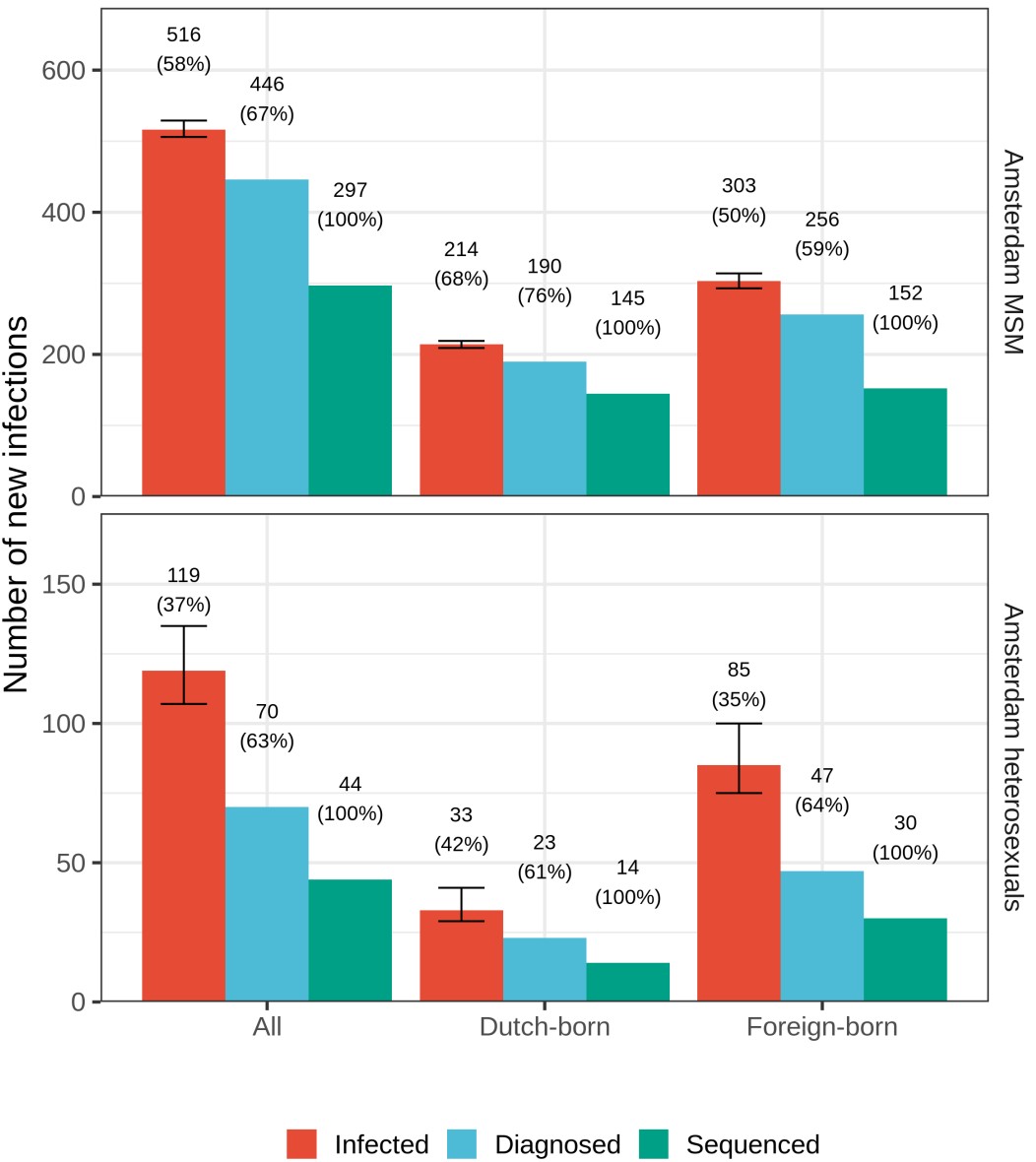

**Appendix 1—figure 35.** Estimated Amsterdam infections in 2014-2018. Estimates of the total number of individuals resident in Amsterdam that were infected in 2014-2018 are shown along with the subset of individuals that were diagnosed, and the subset of those for who at least one viral sequence is available. Posterior median estimates (bars, and number on top of bar) are shown along with 95% credible intervals. The posterior median proportion of individuals with a viral sequence is also shown (proportion on top of bar).

## Sensitivity analysis: Estimating the average undiagnosed from infections acquired in 2014-2018 with weights

Our approach to estimating the proportion undiagnosed fits the model to the times to diagnoses for infections acquired in 2010-2012, and calculates a weighted average to account for a change in incidence rates over the study period 2014-2018 using estimated infection rates. We compared this approach to where we assume constant rates of infection in 2014-2018 (no weights), and where we weight by diagnosis rates in 2014-2018.

*Appendix 1—table 7* compares the estimates for the undiagnosed population with the three approaches. There is evidence for declining incidence among MSM in both the diagnosis and estimated infection rates, so assuming constant weights may over-estimate the proportion

undiagnosed. However, whilst diagnosis rates appear to have also declined over time, estimated infection rates were relatively stable between 2014 and 2018, which leads to similar estimates of undiagnosed with equal weights.

**Appendix 1—table 7.** Estimated undiagnosed HIV infections in Amsterdam until May 2019 using equal weights, or weighting by diagnosis rates or estimated infection rates.

| | | Estimated undiagnosed HIV infections | | |
|---|---|---|---|---|
| Risk group | Region of birth | Equal weights | Weighted by diagnosis rates | Weighted by infection rates |
| Amsterdam MSM | Netherlands | 17% [15-20%] | 11% [9-13%] | 11% [9-13%] |
| | W. Europe, N. America, Oceania | 16% [11-21%] | 9% [6-13%] | 9% [6-14%] |
| | E. & C. Europe | 22% [16-32%] | 14% [9-22%] | 16% [11-24%] |
| | S. America and Caribbean | 23% [19-30%] | 19% [14-25%] | 17% [13-22%] |
| | Other | 27% [20-34%] | 23% [16-31%] | 20% [14-27%] |
| | All | 20% [18-22%] | 14% [13-17%] | 14% [12-16%] |
| Amsterdam heterosexual | Netherlands | 34% [23-47%] | 28% [18-39%] | 30% [21-44%] |
| | Sub-Saharan Africa | 60% [48-69%] | 48% [37-59%] | 57% [47-67%] |
| | S. America and Caribbean | 30% [19-45%] | 25% [16-38%] | 28% [19-42%] |
| | Other | 44% [31-59%] | 31% [18-50%] | 40% [25-57%] |
| | All | 44% [37-50%] | 34% [28-41%] | 41% [35-48%] |
| All | | 24% [22-27%] | 18% [16-20%] | 19% [17-21%] |

## Sensitivity analysis: Using only data on last negative tests

We carried out several sensitivity analyses to explore the impact of alternative approaches to estimating infection dates on the proportion of Amsterdam infections in 2014-2019 that are estimated to have remained undiagnosed by 2019. We first considered estimating the date of infection as the midpoint between last negative HIV test and first positive HIV test, where available. We therefore only considered patients with a last negative HIV test to fit the model for the time-to-diagnosis distributions. In contrast, the approach taken to estimating the infection date in the main analysis considers the time at risk to be either the time since last negative HIV test, or the time since the patient was 15 years old where a last negative test is not available. Based on the midpoint estimates, each individual was classified to have been infected before or after 2014 in analogy to Equation (S11). We had 266 patients across the synthetic cohorts defined by Equation (S12), compared with 632 when using the estimated date of infection. This is reflective of the fact many individuals do not have a reported last negative test date.

*Appendix 1—figure 36* compares the estimated proportion of undiagnosed Amsterdam infections obtained as in the main analysis from all biomarker data from all individuals (*Appendix 1—figure 36A*) to that obtained when using only midpoint estimates from seroconverters (*Appendix 1—figure 36B*). Estimates are compared by year of infection for each risk group. When using data only from the seroconverters, the estimated proportions of undiagnosed individuals are much smaller. This is likely driven by the fact we excluded patients without a last negative test, who may have typically had longer estimated times to diagnosis. This was also observed by *Ratmann et al., 2016*. There are also considerably fewer data points, particularly among heterosexuals, resulting in elevated uncertainty in these estimates. *Appendix 1—figure 37* shows our estimates for the total number of infected individuals in Amsterdam. Clearly, whilst the estimates are more conservative where we use midpoint estimates than we find using the estimated times to diagnosis (see *Appendix 1—figure 37*), we still find a substantial proportion of individuals to be undiagnosed by 2019.

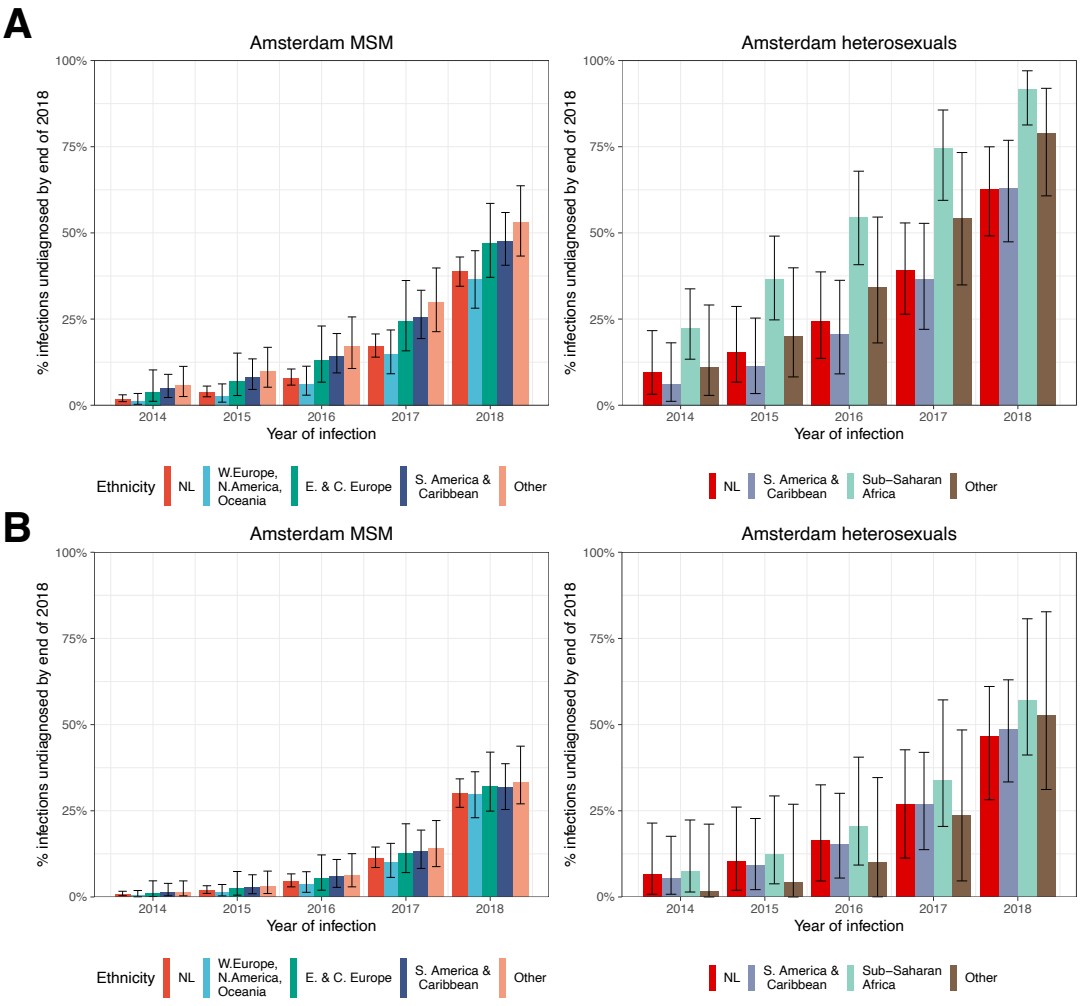

**Appendix 1—figure 36.** Estimated proportion of Amsterdam infections in 2014-2018 which remained undiagnosed by 2019, by year of infection. (**A**) Using all biomarker data from all individuals. (**B**) Using midpoint estimates from seroconverters.

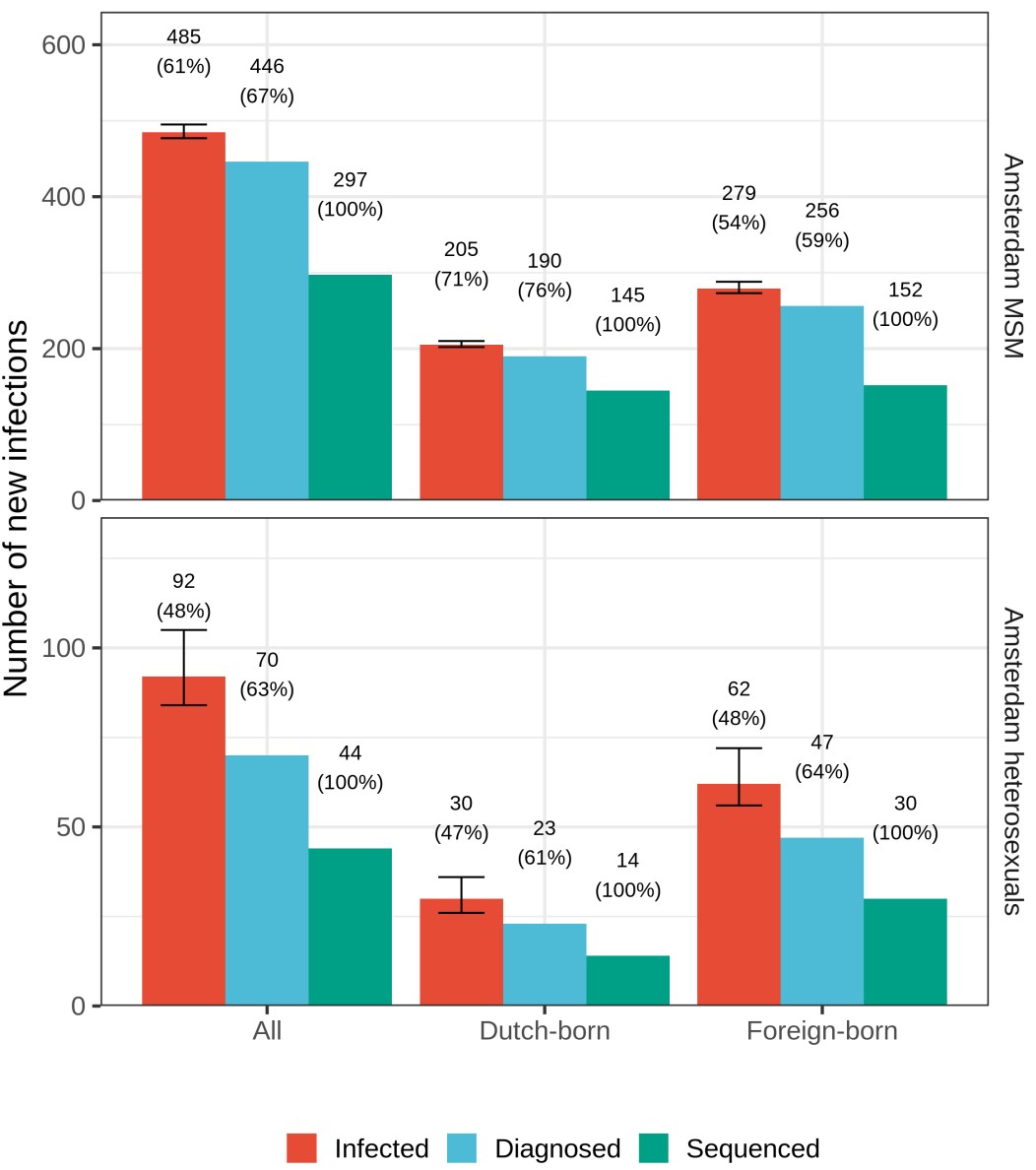

**Appendix 1—figure 37.** Estimated Amsterdam infections in 2014-2018, using midpoint estimates from seroconverters. Estimates of the total number of individuals resident in Amsterdam that were infected in 2014-2018 are shown along with the subset of individuals that were diagnosed, and the subset of those for who at least one viral sequence is available. Posterior median estimates (bars, and number on top of bar) are shown along with 95% credible intervals. The posterior median proportion of individuals with a viral sequence is also shown (proportion on top of bar).

## Sensitivity analysis: Estimates from ECDC modelling tool

Third, we considered utilising estimates for Amsterdam from the ECDC HIV modelling tool (**Stockholm: European Centre for Disease Prevention and Control, 2017**). We used model estimates of the estimated infections per year among MSM and heterosexuals, respectively, and the estimated proportion of infections diagnosed within $1, \ldots, 15$ years. Estimates were only available by transmission group, but not place of birth. Estimates also also only available by year, so we estimate the proportion of infections acquired between 2014 and 2018 undiagnosed by the end of 2019.

If $I_y$ are the estimated number of individuals infected in year $y$, and $\delta_{y,z}$ is the probability of an individual infected in year $y$ being diagnosed in year $z \in \{y, \cdots, 2018\}$. Then, the proportion of

individuals infected between 2014 and 2018 who are undiagnosed by the end of 2019, are estimated by:

$$\delta = \sum_{y \in \mathcal{Y}} \omega_y (1 - \sum_{z=y}^{2018} \delta_{y,z})$$
(S20)

where $\omega_y$ are weights according to the proportion of individuals infected in year $y$:

$$\omega_y = \frac{I_y}{\sum_{z \in \mathcal{Y}} I_z}$$
(S21)

Estimates of undiagnosed were similar for MSM (28.7%) of infections were undiagnosed by 2019, but considerably higher among heterosexuals compared to the estimates from the Weibull model (62.2%).

## Viral phylogenetic analyses

### Multiple sequence alignment

We used partial HIV pol sequences from Amsterdam and the rest of the Netherlands from the ATHENA cohort and aligned these to the reference genome HXB2 (*Ratner et al., 1985*) using Virulign (*Libin et al., 2019*). Sequences which failed to align were aligned globally using Mafft version 7 (*Katoh and Standley, 2013*). Nucleotide positions which were missing for most sequences, or not in the reference sequence HXB2 were removed. Known antiretroviral resistant mutations were masked using the R package big.phylo (*Ratmann, 2019*). The final alignment was 1302nt in length. We carried out some manual curation of the alignment, removing all gaps and resolving sequences which did not align correctly. We then classified sequences by subtype using COMET v2.3 (*Struck et al., 2014*) and verified any which were uncertain with REGA v3.0 (*Pineda-Peña et al., 2013*).

We downloaded 82,708 background sequences from the Los Alamos HIV-1 sequence database on 27th February 2020, specifying fragments in the POL region longer than 1300nt. We then used the Basic Local Alignment Search tool (BLAST, https://blast.ncbi.nlm.nih.gov/Blast.cgi) to identify the top 20 closest background sequences to each of the Dutch sequences, which we kept and aligned to the Dutch sequences using the HXB2 reference sequence. We created alignments by subtype, excluding the least common subtypes with fewer than 50 Dutch and background sequences.

### Reconstruction of city transmission chains

We used FastTree v2.1.8 to reconstruct phylogenetic trees for each subtype (*Price et al., 2010*). We then assigned labels to each sequence. Sequences from Amsterdam were labelled according to their risk group, sequences from the rest of the Netherlands (excluding Amsterdam) were labelled as such, and background sequences were labelled according to the country they originated from. The geographic regions for the MSM trees were,

$\mathcal{N} = \{$Amsterdam MSM, Amsterdam non-MSM, Netherlands, Africa, Western Europe,

Eastern Europe and Central Asia, North America, Latin America and the Caribbean,

Dutch Caribbean and Suriname, Middle East and North Africa, South and South-East Asia, and Oceania$\}$,
(S22)

and similarly for heterosexual trees,

$\mathcal{O} = \{$Amsterdam heterosexual, Amsterdam non-heterosexual, Netherlands, Africa,

Western Europe, Eastern Europe and Central Asia, North America,

Latin America and the Caribbean, Dutch Caribbean and Suriname,

Middle East and North Africa, South and South-East Asia, and Oceania$\}$,
(S23)

We then used phyloscanner v1.8.0 (*Wymant et al., 2018*) to assign one of the state labels to each viral lineage in the reconstructed phylogenies. *Appendix 1—figures 4–14* show the annotated phylogenetic trees for all major subtypes and circulating recombinant forms that circulate in Amsterdam. From the annotated trees, we extracted the viral phylogenetic subgraphs that were assigned to Amsterdam individuals. We assume that viral phylogenetics correctly assigns individuals into subgraphs, which we interpret as the observed parts of distinct city-level transmission chains.

## Branching process model of partially observed, growing transmission chains

In this section, we describe how we build on existing methods to model the growth of the existing and newly introduced transmission chains. Utilising the phylogenetic subgraph data described in Section 4.2, we show how we can model their growth from a point in time, rather than from the first introduction, by utilising the number of infectious cases in the subgraph at a given point in time, and how many new cases were generated from those infectious cases. We also describe the model likelihood of new transmission chains which emerged.

We model the growth of transmission chains using putative infection dates, estimated in 3.1. For individuals with no estimate for date of infection, due to missing clinical biomarker data after diagnosis, we subtracted the posterior median time-to-diagnosis for an individual estimated using the model described in equation (S15a) in the corresponding migrant group, defined by *Equation (S1a) and (S1b)*.

### Probability that $m$ index cases collectively generate $i$ new infections

We model the spread of HIV transmission chains that are characterised by reproduction numbers below one, through branching processes characterised by Negative Binomial offspring distributions (*Blumberg et al., 2014*). A central component of branching process theory is the probability generating function $Q(s) = \sum_{i=0}^{\infty} q_i s^i$, where $q_i$ is the probability that one individual generates $i$ new infections in one generation, and $q_0$ is the probability that one individual generates no further infections. For our purposes, we will use two fundamental formulae. First, the $k$ th derivative of $Q$ is

$$\frac{d}{d^k s} Q(s) = \sum_{i=k}^{\infty} \frac{i!}{(i-k)!} q_i s^{i-k},$$

(S24)

and so the probability $q_k$ is recovered through

$$q_k = \frac{1}{k!} \frac{d}{d^k s} Q(0).$$

(S25)

Second, the $k$ th coefficient of $Q^2(s)$ is

$$\sum_{j=0}^{k} q_j q_{k-j},$$

(S26)

which is the probability that two individuals collectively generate $k$ new infections. Thus, the probability that $m$ index cases collectively generate $i$ new infections is given by the $i$ th coefficient of $Q^m(s)$. We denote this probability by

$$p(i|m) = \frac{1}{i!} \frac{d}{d^i s} Q^m(0).$$

(S27)

We consider a Negative Binomial offspring distribution, parameterised in terms of the mean μ and dispersion parameter $\phi$, so that its variance is given by $\mu(1 + \mu/\phi)$. Thus, as $\phi$ tends to zero, $\mu/\phi$ increases, and so does the variance to mean ratio $(1 + \mu/\phi)$. This means that the Negative Binomial can simultaneously model average reproduction numbers as well as additional heterogeneity in the number of new infections per generation, that goes beyond the variation described by a Poisson offspring distribution. The probability generating function of the Negative Binomial offspring distribution is

$$Q(s) = \left(1 + \frac{\mu}{\phi}\right)^{-\phi}.$$

(S28)

Thus, we have that the probability that $m$ index cases generate $i$ new infections is

$$p(i|m, \mu, \phi) = \frac{1}{i!} \frac{d}{d^i s} Q^m(0)$$

(S29a)

$$= \frac{1}{i!} \left( \prod_{k=0}^{i-1} (\phi m + k) \left( \frac{\mu}{\phi} \right)^i \left( 1 + \frac{\mu}{\phi} \right)^{-\phi m - i} \right) \tag{S29b}$$

$$= \frac{(\phi m + i - 1)!}{i! gt(\phi m - 1)!} \left( \frac{\mu}{\phi} \right)^i \left( 1 + \frac{\mu}{\phi} \right)^{-\phi m - i} \tag{S29c}$$

$$= \frac{(\phi m + i - 1)!}{i!(\phi m - 1)!} \left( \frac{\phi}{\mu + \phi} \right)^{\phi m} \left( 1 - \frac{\phi}{\mu + \phi} \right)^i, \tag{S29d}$$

where $m = 1, 2, \ldots$ are fixed, and the number of new infections takes on values $i = 0, 1, \ldots$. It is helpful to note that *equation (S29a)-(S29d)* has an intuitive interpretation, it is a Negative Binomial with mean $\mu m$ and dispersion parameter $\phi m$, which we denote by

$$p(i|m, \mu, \phi) = \mathrm{NegBin}(i|\mu m, \phi m), \tag{S30}$$

where $m = 1, 2, \ldots$ are fixed, and the number of new infections takes on values $i = 0, 1, \ldots$.

Equivalently, we can express the probability that $m$ index cases result in a total number of $n$ cases through

$$\tilde{p}(n|m, \mu, \phi) = \frac{(\phi m + n - m - 1)!}{(n - m)!(\phi m - 1)!} \left( \frac{\phi}{\mu + \phi} \right)^{\phi m} \left( 1 - \frac{\phi}{\mu + \phi} \right)^{n - m}, \tag{S31}$$

or more simply

$$\tilde{p}(n|m, \mu, \phi) = \mathrm{NegBin}(n - m|\mu m, \phi m), \tag{S32}$$

where $m = 1, 2, \ldots$ are fixed, and the number of total cases are $n = m, m + 1, \ldots$.

## Probability that $m$ index cases result in a transmission chain with $i$ new infections

Transmission chains require that infections occur in a particular order, while in contrast *equation (S29a)-(S29d)* do not impose in what generation how many infections occur. For example, with one index case $m = 1$ and a total size $n$, *Equation S31* quantifies the probability that $n - 1$ new infections occur, but there is no constraint that the index case generates at least one new infection in the next generation.

*Dwass, 1969* derived the correction factor, and the probability that a transmission chain with $m$ index cases has $i$ new infections, or equivalently $n$ cases, is

$$c(i|m, \mu, \phi) = \frac{m}{m + i} p(i|m, \mu, \phi) \tag{S33a}$$

$$\tilde{c}(n|m, \mu, \phi) = \frac{m}{n} p(n - m|m, \mu, \phi), \tag{S33b}$$

where $m = 1, 2, \ldots$, $i = 0, 1, \ldots$, and $n = m, m + 1, \ldots$.

## Probability that $m$ index cases result in subgraphs with $i$ sampled, new infections

In practice, only a subset of new infections are captured in viral phylogenies because only a subset of infections are diagnosed, and of those only a subset have virus sequenced. We make two assumptions. First, infections are missing independently of each other with the same probability $1 - \rho$, so $\rho$ is the sampling probability of infections. Second, uncertainty in $\rho$ can be quantified within several percentage points through surveillance data and/or modelling; we use this assumption later to ensure that the remaining parameters are statistically identifiable.

Then, the probability of observing individuals in a subgraph that has known index cases is

$$
\begin{aligned}
c_{\mathrm{obs}}(i|m, \mu, \phi, \rho) &= \sum_{k=i}^{\infty} \left( \mathrm{Bin}(i|k, \rho) c(k|m, \mu, \phi) \right) \\
&= \sum_{k=i}^{\infty} \left( \mathrm{Bin}(i|k, \rho) \frac{m}{m + k} \mathrm{NegBin}(k|\mu m, \phi m) \right),
\end{aligned} \tag{S34}
$$

where $m = 1, 2, \ldots,$ $i = 0, 1, \ldots,$ and $c(k|m, \mu, \phi)$ is from **Equation S33a**. It is possible that an observed subgraph has $m$ index cases by a particular study start time $\psi_{\text{start}}$ and no new infections between $\psi_{\text{start}}$ and $\psi_{\text{end}}$, as defined in , **Equations S2a and S2b**, and the probability of observing one such subgraph is $c_{\text{obs}}(0|m, \mu, \phi, \rho)$.

## Probability that emergent subgraphs have $n$ sampled cases

Some observed subgraphs are emergent in the sense that they consist of individuals that were all diagnosed after the study start time $T$. In this case, **Equation (S34)** cannot be used because it assumes that subgraphs contain at least one index case prior to the study start time $T$. We assume that emergent subgraphs are seeded by one index case, which for example ignores the possibility that sexual partners infected each other and then moved to Amsterdam, and seeded a new transmission chain in Amsterdam. The probability of observing an emergent transmission chain of size $n$ is given by

$$
\begin{aligned}
\tilde{c}_{\text{obs}}(n|m = 1, \mu, \phi, \rho) &= \frac{\sum_{z=n}^{\infty}\left(\text{Bin}(n|z,\rho)\tilde{c}(z|m=1,\mu,\phi)\right)}{1-\sum_{z=1}^{\infty}\left(\text{Bin}(0|z,\rho)\tilde{c}(z|m=1,\mu,\phi)\right)} \\
&= \frac{\sum_{z=n}^{\infty}\left(\text{Bin}(n|z,\rho)\tilde{c}(z|m=1,\mu,\phi)\right)}{1-\sum_{z=1}^{\infty}\left((1-\rho)^z\tilde{c}(z|m=1,\mu,\phi)\right)} \\
&= \frac{\sum_{z=n}^{\infty}\left(\text{Bin}(n|z,\rho)\frac{1}{z}\text{NegBin}(z-1|\mu,\phi)\right)}{1-\sum_{z=1}^{\infty}\left((1-\rho)^z\frac{1}{z}\text{NegBin}(z-1|\mu,\phi)\right)},
\end{aligned}
\tag{S35}
$$

where unlike **Equation (S34)**, $n = 1, 2, \ldots$ may include in the count the index case (if it is sampled), and $\tilde{c}(z|m = 1, \mu, \phi)$ is from **Equation (S33b)**. The denominator corrects for the event that the index case and all new infections in an emergent chain are unsampled, which is possible with non-zero probability, but always unobserved.

## Likelihood of the growth distribution of phylogenetic subgraphs

We now describe the likelihood of the growth distribution of viral phylogenetic subgraphs, which throughout we identify as the observed parts of distinct city-level transmission chains. In what follows, for brevity, we only consider one transmission group and omit reference to this transmission group. All equations are analogous for the other transmission group.

We start with the viral phylogenetic subgraphs in the viral phylogeny of one subtype, and omit for brevity also any indication of that subtype. The data consist of a two-dimensional array $\mathbf{x}$, where $x_{mi}$ denotes the number of subgraphs that had $m$ index cases at the study start time $\psi_{\text{start}}$ and $i$ sampled, new infections by the study end time $\psi_{\text{end}}$. Here, $m = 1, \ldots, M$ and $i = 0, \ldots, I$ where $M$ denotes the largest number of index cases observed, and $I$ denotes the largest number of new infections observed. In addition, the data consist of a one-dimensional array $\tilde{x}$, where $\tilde{x}_n$ denotes the number of emergent subgraphs that have $n$ sampled cases during the study period. Here, $n = 1, \ldots, N$, because at least one case needs to be sampled in order to observe the corresponding subgraph.

Then, we associate the following log-likelihood to the growth distributions of pre-existing and emergent subgraphs,

$$
(x, \tilde{x}|\mu, \phi, \rho) = \left(\sum_{m=1}^{M}\sum_{i=0}^{I} x_{mi} \log c_{\text{obs}}(i|m, \mu, \phi, \rho)\right) + \left(\sum_{n=1}^{N} \tilde{x}_n \log \tilde{c}_{\text{obs}}(n|m = 1, \mu, \phi, \rho)\right).
\tag{S36}
$$

The log-likelihood thus involves infinite sums through **equations (S34) and (S35)**. We approximated these by summing up to the $10 * I * (N_D/N_S)$ th term, where $N_D$ are the number of diagnosed individuals and $N_S$ are the number of sequenced individuals, so $I * (N_D/N_S)$ is the expected number of individuals in the transmission chain that corresponds to the largest observed subgraph. In applying this log-likelihood, we assume that (1) all transmission chains have reached their final size by the end of the study period, i.e. that they are complete; (2) that all emergent transmission chains have one index case; (3) that each case has an equal and independent probability of being sampled.

Next we consider the joint likelihood that arises from consideration of viral subgraphs of the same transmission group (e.g. MSM or heterosexual individuals) across all HIV subtypes or circulating recombinant forms. Since the number of subgraphs and new cases acquired between 2014-2018 are very small for some subtypes, we aggregate the subgraph size distributions for non-B subtypes. We index subtypes B and non-B by $s = 1, \ldots, S$, where $S = 2$, and denote the corresponding subgraph growth distributions by $\mathbf{x}_s$, and $\tilde{x}_s$. Then, we model the log-likelihood of all the data for one transmission group through

$$
\begin{aligned}
ll \quad &= \sum_{s=1}^{S} l(x_s, \tilde{x}_s | \mu_s, \phi_s, \rho_s) \\
&= \sum_{s=1}^{S} \left[ \left( \sum_{m=1}^{M} \sum_{i=0}^{I} x_{mi} \log c_{obs}(i|m, \mu_s, \phi_s, \rho_s) \right) + \right. \\
&\quad \left. \left( \sum_{n=1}^{N} \tilde{x}_n \log \tilde{c}_{\mathrm{obs}}(n|m=1, \mu_s, \phi_s, \rho_s) \right) \right],
\end{aligned}
\tag{S37}
$$

where the $\mu_s$, $\phi_s$, $\rho_s$ are specific to the corresponding transmission group and subtype.

## Bayesian inference

We estimate city-level transmission dynamics, the growth distribution of transmission chains, and the proportion of locally acquired infections through the log-likelihood (*Equation S37*) of phylogenetically observed subgraphs.

## Preliminaries
### 6.1.1 Number of index cases
For each individual $i$ in the cohort $\mathcal{N}$, if $r_i$ is their last viral load measurement taken before 2014, we define them to be not virally suppressed by 2014 through,

$$
S_i = \begin{cases} 1, & \text{if } T_i^{\mathrm{infection}} < 2014 \cap r_i > 100 \\ 0, & \text{otherwise.} \end{cases}
\tag{S38}
$$

Then, for each observed subgraph $j$ where $(j = 1, ..., A)$, $m_j$ are the observed index cases, we count the number of individuals infected by, but who were not virally suppressed, by the start of 2014. For example for MSM, if $\mathcal{C}^{\mathrm{MSM}}$ is the subset of MSM in Amsterdam,

$$
\mathcal{C}^{\mathrm{MSM}} \subseteq \mathcal{N} : R_i = 1,
\tag{S39}
$$

the number of observed index cases in subgraph $j$ is,

$$
m_j = \sum_{k \in \mathcal{C}^{\mathrm{MSM}}} S_{jk},
\tag{S40}
$$

and $m_j > 0$. We count analogously for heterosexuals. The actual number of index cases $m_j^* \sim \mathrm{NegBinom}(m_j, \nu)$, where $\nu$ is the sampling fraction of individuals who were not virally suppressed by 2014. We estimate the true number of index cases under complete sampling $m_j^*$ by,

$$
E(m_j^*) = \frac{m_j}{\nu}, \quad i = 1, ..., A
\tag{S41}
$$

When $m_j = 0$, estimate $m_j^*$ from the mode of the pmf for the distribution $\mathrm{Binomial}(0; m_j^*, \nu)$.

### 6.1.2 Number of subgraphs with no new infections
For the subgraphs in which no individuals were not virally suppressed by 2014 (i.e. no observed index case), and no observed new case between 2014-2018, were not included in the subgraph sizes and assumed to have died out.

## Hierarchical model
The parameters of the model (*Equation S37*) are the subtype-specific mean parameters of the offspring distributions, $\mu_1, \ldots, \mu_S$, the dispersion parameters $\phi_s$ and the sampling parameter $\rho$. To estimate the $\mu_1, \ldots, \mu_S$, we borrow information across subtypes through a hierarchical prior

distribution. We interpret the mean parameters of the offspring distributions as the effective reproduction numbers during the study period for the corresponding subtype. The variance-to-mean ratio of the Negative Binomial offspring distribution is $1 + \mu_s/\phi_s$ and measures the degree of dispersion of the size distribution of the transmission chains. For ease of inference, we re-parameterize the dispersion parameter into the variance-to-mean ratio minus one and also specify a hierarchical prior distribution,

$$v_s = \mu_s/\phi_s. \tag{S42}$$

The log posterior density is given by

$$
\begin{aligned}
&\log p\left(\mu^s, v, \rho | \mathbf{x}_s, \tilde{x}_s, s = 1, \ldots, S\right) \\
&\propto \sum_{s=1}^{S} ll\left(\mathbf{x}_s, \tilde{x}_s | \mu_s, \mu_s v, \rho\right) + \sum_{s=1}^{S} \log p(\mu^s) + \log p(v) + \log p(\rho)
\end{aligned} \tag{S43}
$$

where the prior densities are specified as follows. For the effective reproduction numbers, we specified the normal-normal two-level

$$
\begin{aligned}
\log \mu_s &\sim N(\log \mu, \sigma^2) \\
\log \mu &\sim N(\hat{\mu}_{\log \mathrm{MLE}}, 0.3) \\
\sigma &\sim \mathrm{Exp}(0.1).
\end{aligned} \tag{S44}
$$

The hyperprior of the grand mean was centred on the maximum likelihood estimate $\log \hat{\mu}_{\mathrm{MLE}} = \log(1 - 1/\bar{x})$, where $\bar{x}$ is the average subgraph size (**Blumberg and Lloyd-Smith, 2013**). The hyperprior of the grand standard deviation $\sigma$ was specified by considering the differences in the log maximum likelihood estimates $\log \hat{\mu}_{\mathrm{MLE}}$ for each subtype.

For the variance-to-mean ratio, we specified

$$v_s \sim \mathrm{Exp}(v), \, v \sim \mathrm{Exp}(1), \tag{S45}$$

where 1 is the rate parameter for the exponential distribution. For the sampling parameter, we specified

$$\rho \sim \mathrm{Beta}(N_S + 0.5, (N_D/(1 - \delta) - N_S) + 0.5), \tag{S46}$$

where $N_S$ are the number of sequenced individuals, $N_D$ are the number diagnosed and $\delta$ are the proportion of undiagnosed individuals.

## Numerical inference

The joint posterior distribution was estimated using Stan version 2.21 across three chains, each with 2,000 samples.

The models mixed well; *Appendix 1—figures 40 and 41* show the trace plot for the parameter with the smallest effective sample size in each model, which was 1637 for the MSM model and 1622 for the heterosexual model. *Appendix 1—figures 42 and 43* shows the pairs plot of parameters for the MSM and heterosexual models, respectively. We note that we did not observe multiplicative non-identifiabilities (banana shape) between the reproduction rate R0 and the variance-to-mean ratio, as found by *Blumberg and Lloyd-Smith, 2013*.

## Target quantities derived from fitted model

### Estimated number of new cases in transmission chains since 2014

To estimate the actual number of new infections in transmission chains since 2014 from the phylogenetically observed subgraphs, we use the model fits in combination with the size equations (5.2) and (5.2) to obtain the posterior predictive number of new cases in a transmission chain with $m = 1, \ldots$ index cases in 2014. For emergent chains, we assume as before that there was one index case since 2014. Specifically, we have

$$p(i^* | x, \tilde{x}, m) = \int c(i^* | m, \mu, \phi) p(\mu, \phi | x, \tilde{x}) d(\mu, \phi) \tag{S47}$$

where $i^* = 0, 1, \ldots$, and for ease of notation we have dropped the suffixes for different subtypes, transmission groups, or time intervals. We approximate *Equation (S47)* numerically from $k = 1, \ldots, K$ Monte Carlo samples $\mu^{(k)}, \phi^{(k)}$ of the joint posterior distribution by generating samples from

$$i^{*(k)} \sim c(i^*|m, \mu^{(k)}, \phi^{(k)}), \quad k = 1, \ldots, K. \tag{S48}$$

This is easily done since the inference algorithm already tabulates the probabilities $c(i^*|m, \mu^{(k)}, \phi^{(k)})$ for $i^* = 0, 1, \ldots$.

*Equation (S48)* allows us to generate one Monte Carlo sample of the actual growth of all transmission chains. We denote the number of all pre-existing phylogenetically observed transmission chains with at least one index case by

$$|\mathbf{x}| = \sum_{m=1}^{M} \sum_{i=1}^{I} x_{mi}, \tag{S49}$$

and index each of them through $j^x = 1, \ldots, |\mathbf{x}|$. Correspondingly we denote the number of emergent subgraphs by

$$|\tilde{x}| = \sum_{n=1}^{N} \tilde{x}_n. \tag{S50}$$

A certain proportion of emergent transmission chains remains phylogenetically unobserved owing to incomplete sampling. In our model, the probability that an emergent transmission chain is entirely unobserved is given by

$$\rho_{\text{not-obs}}^e = \sum_{z=1}^{\infty} (1 - \rho)^z \frac{1}{z} c(z - 1|m = 1, \mu, \phi), \tag{S51}$$

as in *Equation (S35)*. Thus, the expected number of emergent transmission chains is $|\tilde{x}|/(1 - \rho_{\text{not-obs}}^e)$. We obtain a Monte Carlo estimate of (*Equation S51*) by plugging in our estimates of the joint posterior density,

$$\rho_{\text{not-obs}}^{e(k)} = \sum_{z=1}^{\infty} (1 - \rho^{(k)})^z \frac{1}{z} c(z - 1|m = 1, \mu^{(k)}, \phi^{(k)}). \tag{S52}$$

Using *Equation (S52)*, we can predict the number of completely unobserved, emergent subgraphs through

$$N_{\text{not-obs}}^{*(k)} \sim \text{NegBin}\left(|\tilde{x}|, \rho_{\text{not-obs}}^{e(k)}\right), \tag{S53}$$

where the Negative Binomial is specified in terms of the number of failures and success probabilities, with mean $\left(|\tilde{x}|(1 - \rho_{\text{not-obs}}^{e(k)})\right)/(1 - \rho_{\text{not-obs}}^{e(k)})$, so that the mean of $|\tilde{x}| + N_{\text{not-obs}}^{*(k)}$ equals as desired $|\tilde{x}|/(1 - \rho_{\text{not-obs}}^{e(k)})$. We index all emergent transmission chains through $j^e = 1, \ldots, |\tilde{x}| + N_{\text{not-obs}}^{*(k)}$, and note that the number of emergent transmission chains is uncertain.

Then, the actual number of new cases since 2014 in the chain corresponding to the observed, pre-existing subgraph $j^x$ is predicted by sampling $i_{j^x}^{*(k)} \sim c(\cdot|m_{j^x}, \mu^{(k)}, \phi^{(k)})$, where $m_{j^x}$ is the number of index cases in the corresponding subgraph. Similarly, the actual number of new cases since 2014 of the chain corresponding to the emerging transmission chain $j^e$ is predicted by sampling $i_{j^e}^{*(k)} \sim \tilde{c}(\cdot|1, \mu^{(k)}, \phi^{(k)})$, and then calculating $n_{j^e}^{*(k)} = i_{j^e}^{*(k)} + 1$. For the emergent subgraphs, we add 1 since we assume as before that the index case occurred after 2014. Aggregating these sizes, we predict the size distribution of the number of chains with $i$ new cases since 2014 by

$$x_i^{*(k)} = \sum_{j^x=1}^{|\mathbf{x}|} \mathbb{1}\left(i_{j^x}^{*(k)} == i\right) + \sum_{j^e=1}^{|\tilde{x}|+N_{\text{not-obs}}^{*(k)}} \mathbb{1}\left(1 + i_{j^e}^{*(k)} == i\right), \tag{S54}$$

where $i = 0, 1, \ldots$ and $\mathbb{1}$ is the indicator function that evaluates to 1 if $i_{j^x}^{*(k)}$ is equal to $i$, and otherwise to zero. The median estimate and 95% credible intervals of $x_i^*$ are obtained by drawing posterior samples $(k)$, repeating the calculation of (S54) for each $k$, and then summarising the set of samples.

*Appendix 1—figure 38* shows the observed growth of subgraphs in red next to the predicted actual growth of subgraphs in blue (with 95% credible intervals) for Amsterdam MSM and heterosexuals.

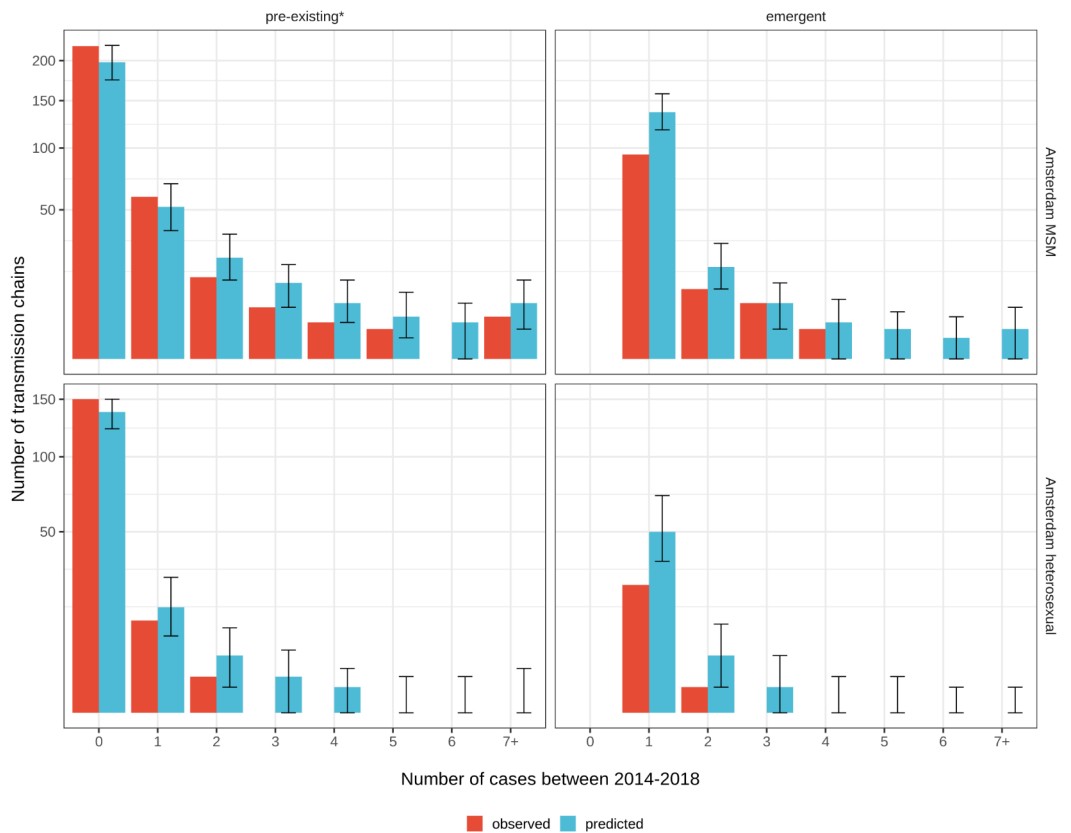

**Appendix 1—figure 38.** Growth of pre-existing (left) and emergent (right) phylogenetically observed subgraph sizes using estimated date of infection. Estimates generated from 203 phylogenetic subgraphs among Amsterdam MSM, containing 297 individuals, and 41 subgraphs among Amsterdam heterosexuals, containing 44 individuals. * pre-existing prior to 2014.

## Estimated origins of transmission chains between 2014 and 2018

If $\hat{\pi}_r = (\hat{\pi}_1, \ldots, \hat{\pi}_R)$ are the proportion of phylogenetically observed subgraphs since 2014 with geographic origin $r$, we can predict the origins of the pre-existing and emergent transmission chains, $y_j$, for each Monte Carlo sample through,

$$y_j^{(k)} \sim \text{Categorical}(\hat{\pi}r) \tag{S55}$$

We then denote the proportion of predicted emergent transmission chains since 2014 with Amsterdam origin $(A)$ by

$$\lambda^{(k)} = \sum_{j^e=1}^{|\bar{x}|+N_{\text{not-obs}}^{*(k)}} \frac{\mathbb{1}(y_{j^e}^{(k)}==A)}{\sum_{r=1}^{R} y_{j^e}^{(k)}}. \tag{S56}$$

*Appendix 1—table 5* reports the estimated ancestral origins of viral lineages in Amsterdam in the central phylogenetic analysis, and uncertainty estimates obtained from the bootstrapped analyses. The estimated origins predicted from the model are also reported, with 95% credible intervals.

## Estimated number of new cases between 2014 and 2018

From Section 7.1, we can use the posterior predictive distribution of the number of new cases for a chain of a given index size **Equation S47** to obtain a Monte Carlo prediction of the number of city-level cases since 2014. This is given by

$$x^{*(k)} = \sum_{j^x=1}^{|\mathbf{x}|} i_{j^x}^{*(k)} + \sum_{j^e=1}^{|\tilde{x}|+N_{\text{not-obs}}^{*(k)}} (1 + i_{j^e}^{*(k)}). \tag{S57}$$

## Estimated ethnicity of new cases between 2014 and 2018

For Amsterdam MSM, we consider geographic regions of birthplace of new cases $r \in \mathcal{M}$. For Amsterdam heterosexuals, we consider georegions $r \in \mathcal{H}$, defined in **Equation (S1a) and (S1b)**. Consider, for MSM, $\hat{\zeta}_r$ is a vector of the proportions of diagnosed individuals estimated to have been infected since 2014, born in each geographic region $r \in \mathcal{M}$. We then predict the birthplace regions of the total $x^{*(k)}$ new cases between 2014-2018, $\mathbf{z}_n$ , for each Monte Carlo sample through,

$$\mathbf{z}_n^{(k)} \sim \text{Multinomial}(\hat{\zeta}_r, x^{*(k)}) \tag{S58}$$

## Proportion of locally acquired infections

The proportion of locally acquired infections since 2014 is defined by the proportion of city-level cases since 2014 that acquired infections in Amsterdam. In the model, all secondary infections originating from index cases of pre-existing transmission chains are infections that were acquired locally. Similarly, all secondary infections originating from index cases of emergent transmission chains are infections that were acquired locally. The index cases of pre-existing transmission chains do not contribute to the denominator (S57) because they existed prior to 2014. This leaves the index cases of emergent transmission chains, for which we also have a Monte Carlo estimate,

$$|\tilde{x}| + N_{\text{not-obs}}^{*(k)}. \tag{S59}$$

A proportion of these index cases also acquired infection locally, from other risk groups in Amsterdam. We denote this proportion by $\lambda$ (**Equation S56**). This allows us to estimate the proportion of locally acquired infections since 2014 through

$$\gamma^{(k)} = 1 - \frac{\left(1 - \lambda^{(k)}\right)\left(|\tilde{x}| + N_{\text{not-obs}}^{*(k)}\right)}{x^{*(k)}}. \tag{S60}$$

The median estimate and 95% credible intervals of $\gamma$ are obtained by drawing posterior samples $(k)$, repeating the calculation of **Equation S60** for each $k$, and then summarising the set of samples.

**Appendix 1—table 8** presents the estimated proportion of locally acquired infections by subtype, and the quantities used to calculate from **Equation S60**.

**Appendix 1—table 8.** Input quantities used to estimate proportion of infections acquired locally in Amsterdam.

| Risk group | Subtype | Chains of non-Amsterdam origin $(1-\lambda)$ | Phylogenetically observed emergent subgraphs $(|\tilde{x}|)$ | Emergent transmission chains (unobserved) (N'not-obs) | Total emergent chains (partially observed + unobserved $|\tilde{x}| + N_{\text{not-obs}}^*$]) | Individuals in pre-existing and emergent chains $(x')$ | Proportion of infections that are importations $\left(\frac{(1-\lambda)(|\tilde{x}|+N_{\text{not-obs}}^*)}{x^*}\right)$ | External importations $\left(\frac{(1-\lambda)(|\tilde{x}|+N_{\text{not-obs}}^*)}{x^*}\right)$ | Locally acquired infections $(\gamma)$ |
|---|---|---|---|---|---|---|---|---|---|
| Amsterdam hetersexual | B | 78.6% [70.6-86.2%] | 12 [12-12] | 14 [5-30] | 26 [17-42] | 58 [35-95] | 0.47 [0.27-0.7] | 36.6% [21.1-55.6%] | 63.4% [44.4-78.9%] |
| Amsterdam hetersexual | Non-B | 93% [88.2-97.3%] | 14 [14-14] | 17 [7-35] | 31 [21-49] | 58 [37-93] | 0.55 [0.35-0.78] | 51% [32.1-72.5%] | 49% [27.5-67.9%] |
| Amsterdam MSM | B | 99.5% [98.6-100%] | 85 [85-85] | 45 [30-64] | 130 [115-149] | 412 [332-521] | 0.32 [0.25-0.4] | 31.5% [24.8-39.3%] | 68.5% [60.7-75.2%] |
| Amsterdam MSM | Non-B | 98.5% [94.1-100%] | 29 [29-29] | 13 [5-24] | 42 [34-53] | 106 [72-172] | 0.4 [0.24-0.58] | 38.7% [23.5-57.2%] | 61.3% [42.8-76.5%] |

We then seek to estimate the proportion of locally acquired infections by ethnicity for each transmission group model separately. Until now, all generated quantities are calculated for each subtype, without specific indexing. To obtain estimates of locally acquired infections by ethnicity, we apply weights to the subtype-specific estimates of locally acquired infections, using the proportion

of predicted individuals from each georegion with subtype B or non-B infections. To obtain estimates of locally acquired infections by place of birth, we first generate weights which correspond to the proportion of infected individuals from region of birth $r$ that were infected with subtype/circulating recombinant form $s$,

$$\nu_{sr}^{(k)} = \frac{\sum_{n=1}^{x_s^{*(k)}} 1\left(z_{sn}^{(k)}==r\right)}{\sum_{i \in S} \sum_{n=1}^{x_i^{*(k)}} 1\left(z_{in}^{(k)}==r\right)}. \tag{S61}$$

Then, if $\gamma_s^{(k)}$ is the proportion of locally acquired infections for subtype $s \in \{\text{B,non-B}\}$, we then calculate the proportion of locally acquired infections by place of birth $r$ as follows:

$$\gamma_r^{(k)} = \sum_{s \in S} \nu_{sr}^{(k)} \cdot \gamma_s^{(k)}. \tag{S62}$$

As before, the median estimate and 95% credible intervals of $\gamma$ are obtained by drawing posterior samples ($k$), repeating the calculation of *Equation S62* for each $k$, and then summarising the set of samples.

*Appendix 1—table 9* presents the characteristics of the observed phylogenetically observed subgraphs alongside the model estimates for the parameters of the branching process model, and the proportion of infections estimated to have been acquired through city-level transmissions by transmission group and subtype.

*Appendix 1—figure 39* presents the estimated $\gamma_r^{(k)}$ from *Equation S62* and the composition of subtypes among the predicted total new cases used to estimate locally acquired infections by ethnicity from the subtype-specific estimates.

**Appendix 1—table 9.** Empirical results from partially observed subgraphs in phylogenetic trees, and model estimates based on complete transmission chains, adjusting for sampling (those in study with a sequence available) for new infections since 2014.

Estimated reproduction number and proportion of locally acquired infections are also presented.

| | | Phylogenetically observed | | | | Model estimates | | | |
|---|---|---|---|---|---|---|---|---|---|
| Risk group | Subtype | New cases | Subgraphs | Average new cases | Transmission chains | Average new cases | Effective reproduction number | Variance-to-mean ratio | Infections acquired in Amsterdam |
| Amsterdam MSM | B | 241 | 368 | 0.65 | 413 [398-432] | 1.01 | 0.26 [0.22-0.31] | 1.69 [1.26-2.38] | 68.5% [60.7-75.2%] |
| Amsterdam MSM | Non-B | 65 | 55 | 1.18 | 68 [60-79] | 1.62 | 0.39 [0.28-0.53] | 1.33 [1.02-2.53] | 61.3% [42.8-76.5%] |
| Amsterdam heterosexual | Non-B | 21 | 105 | 0.2 | 122 [112-140] | 0.49 | 0.17 [0.09-0.26] | 1.26 [1.01-2.94] | 49% [27.5-67.9%] |
| Amsterdam heterosexual | B | 23 | 86 | 0.27 | 100 [91-116] | 0.59 | 0.19 [0.11-0.3] | 1.25 [1.01-2.59] | 63.4% [44.4-78.9%] |

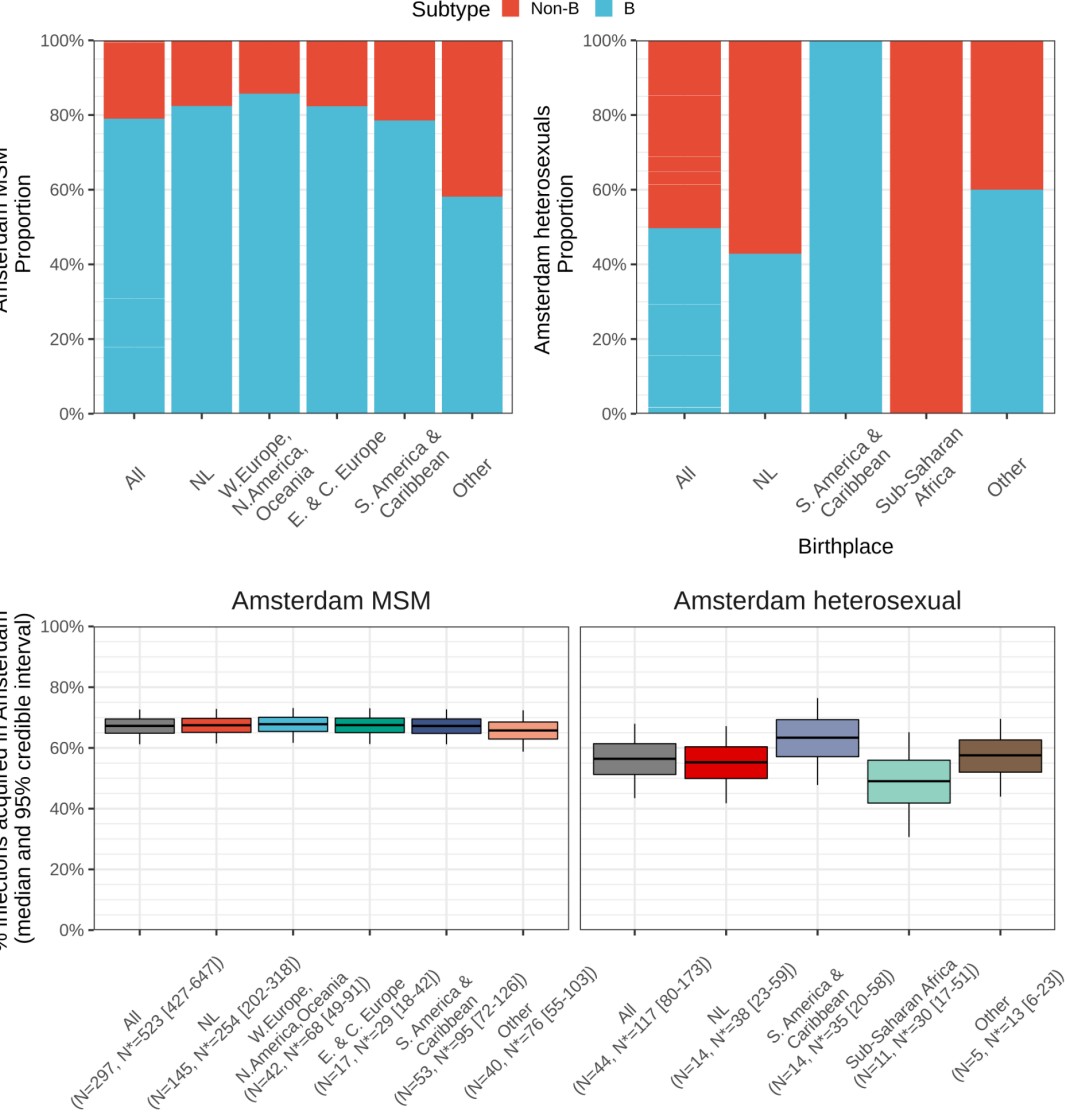

**Appendix 1—figure 39.** Top: Composition of subtypes among total predicted new cases. Bottom: Estimated local infections among MSM (left) and heterosexuals (right), stratified by place of birth between 2014-2018. N = number of sequences available, N* = estimated actual infections [95% credible interval].

## Assessing model fit

To assess model fit, we perform posterior predictive checks against the phylogenetically observed growth distribution of subgraphs for each transmission group and subtype. To keep notations simple, we drop in what follows the suffixes that indicate dependence on transmission group and subtype.

We previously described the phylogenetically observed growth distribution through the number of pre-existing subgraphs with $m$ index cases by 2014 and $i$ new cases since 2014, $x_{mi}$, and the number of emergent subgraphs since 2014 with new cases. To generate posterior predictions and , we index the pre-existing, phylogenetically observed subgraphs by . With regard to emergent transmission chains, for the purpose of assessing model fit, we index only the corresponding phylogenetically observed subgraphs, . In analogy to Equation (S47), we use the sampling-adjusted size equations (S34) and (S35), which lead to the posterior predictive densities

$$p_{\text{obs}}(i^*|\mathbf{x}, \tilde{x}, m) = \int c_{\text{obs}}(i^*|m, \mu, \phi, \rho)p(\mu, \phi, \rho|\mathbf{x}, \tilde{x})d(\mu, \phi, \rho) \tag{S63}$$

$$p_{\text{obs}}(n^*|\mathbf{x}, \tilde{x}) = \int \tilde{c}_{\text{obs}}(n^*|m = 1, \mu, \phi, \rho) p(\mu, \phi, \rho|\mathbf{x}, \tilde{x}) d(\mu, \phi, \rho). \tag{S64}$$

We use these posterior predictive densities to predict the (observed) growth of the pre-existing subgraphs, and the (observed) growth of the emergent subgraphs, and compare these predictions to the observed values. Specifically, given a Monte Carlo sample $\mu^{(k)}, \phi^{(k)}, \rho^{(k)}$ from the posterior distribution, we predict the growth of the pre-existing, phylogenetically observed subgraph $j^x$ through

$$i_{j^x}^{*(k)} \sim c_{\text{obs}}(\cdot|m_{j^x}, \mu^{(k)}, \phi^{(k)}, \rho^{(k)}). \tag{S65}$$

Similarly, we predict the growth of the emergent, phylogenetically observed subgraph $j^e$ through

$$n_{j^e}^{*(k)} \sim \tilde{c}_{\text{obs}}(\cdot|1, \mu^{(k)}, \phi^{(k)}, \rho^{(k)}). \tag{S66}$$

Finally, we aggregate ((S65)-(S66)) to obtain a posterior prediction of the observed growth distributions,

$$x_{mi}^{*(k)} = \sum_{j^x=1}^{|\mathbf{x}|} 1\left(i_{j^x}^{*(k)} == i \text{ and } m_{j^x} == m\right) \tag{S67}$$

$$\tilde{x}_n^{*(k)} = \sum_{j^e=1}^{|\tilde{x}|} 1\left(n_{j^e}^{*(k)} == n\right). \tag{S68}$$

The posterior predictive check then tests if the observed $x_{mi}$, $\tilde{x}_n$ lie within the 95% range of the posterior predictive samples $\{x_{mi}^{*(k)}, k = 1, \ldots, K\}$ and $\{\tilde{x}_n^{*(k)}, k = 1, \ldots, K\}$.

*Appendix 1—figure 27* shows the posterior predictive check for the MSM and heterosexual models, respectively, by subtype. 100% of the observed subgraph counts fall within the 95% credible intervals of the predicted subgraph size distribution, indicating very good model fit.

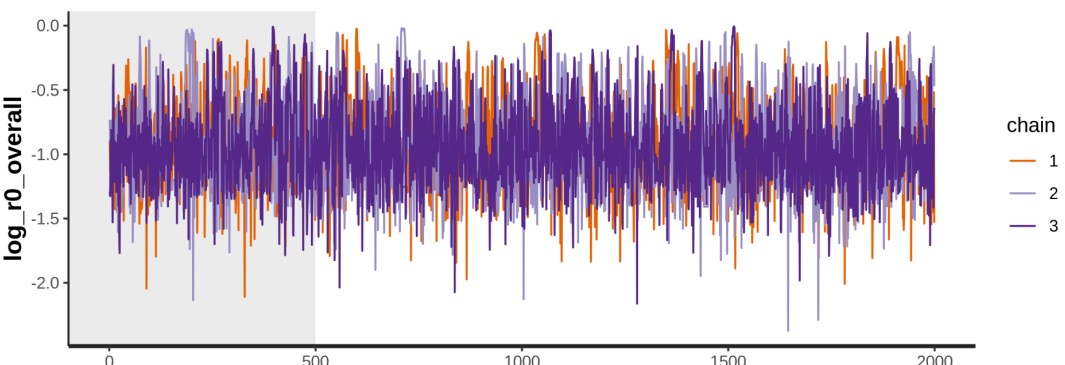

**Appendix 1—figure 40.** Trace plot of parameter with the smallest effective sample size for Amsterdam MSM model.

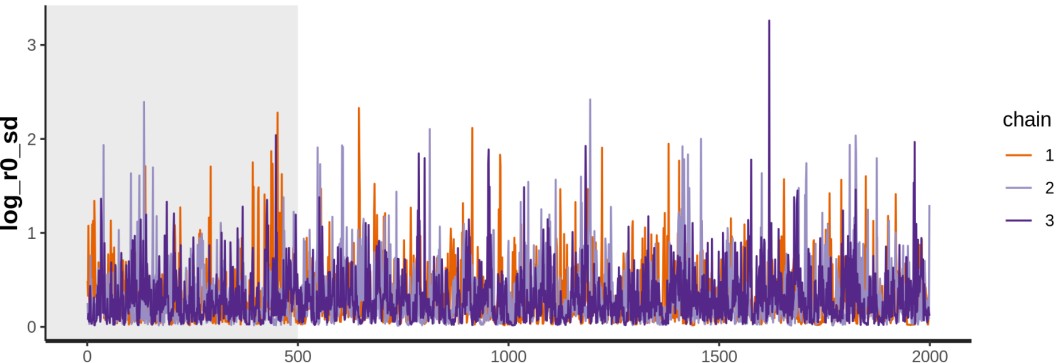

**Appendix 1—figure 41.** Trace plot of parameter with the smallest effective sample size for Amsterdam heterosexual model.

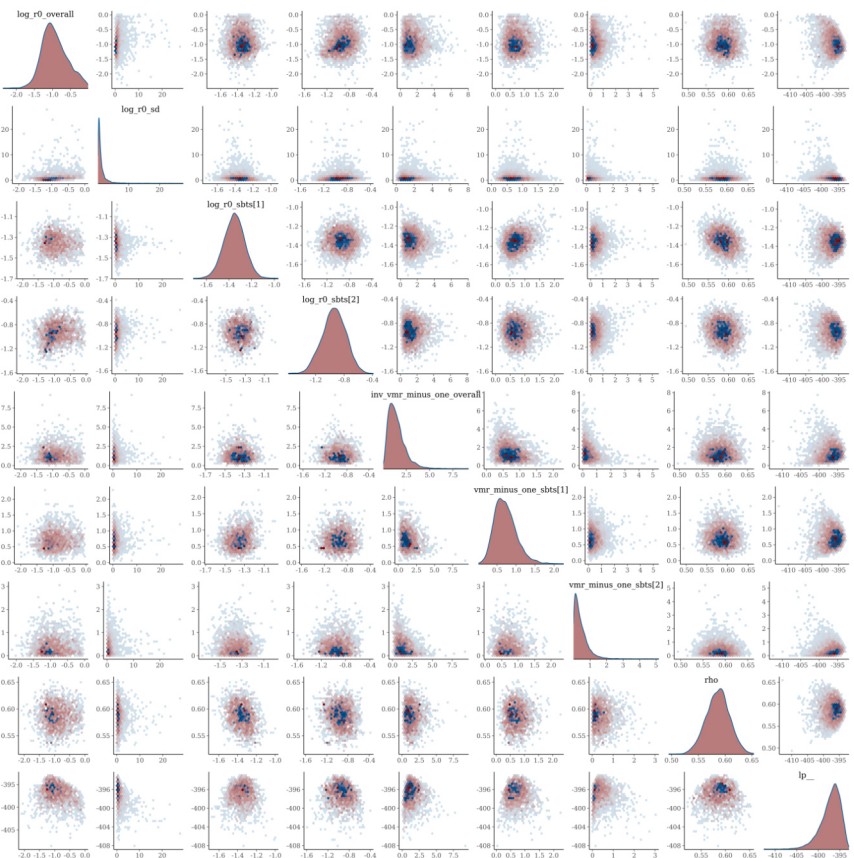

**Appendix 1—figure 42.** Pairs plot of the joint posterior density of the model parameters for Amsterdam MSM.

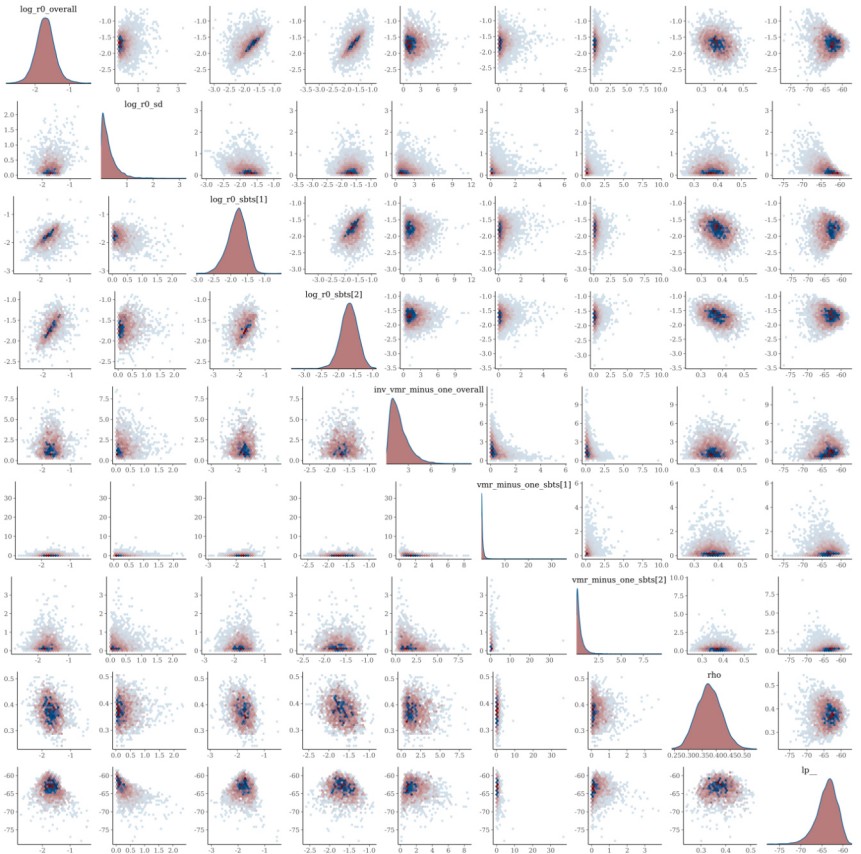

**Appendix 1—figure 43.** Pairs plot of the joint posterior density of the model parameters for Amsterdam heterosexuals.

## Sensitivity analysis

### Observed subgraph size distribution considering infection date

*Appendix 1—figure 44* shows how the observed growth distributions of the subgraphs compare when considering all diagnoses with a sequence between 2014 and 2018, and all diagnoses with a sequence estimated to have been infected between 2014 and 2018. There are fewer sequences in total when considering infection date, since some diagnoses since 2014 are estimated to be infections acquired before 2014.

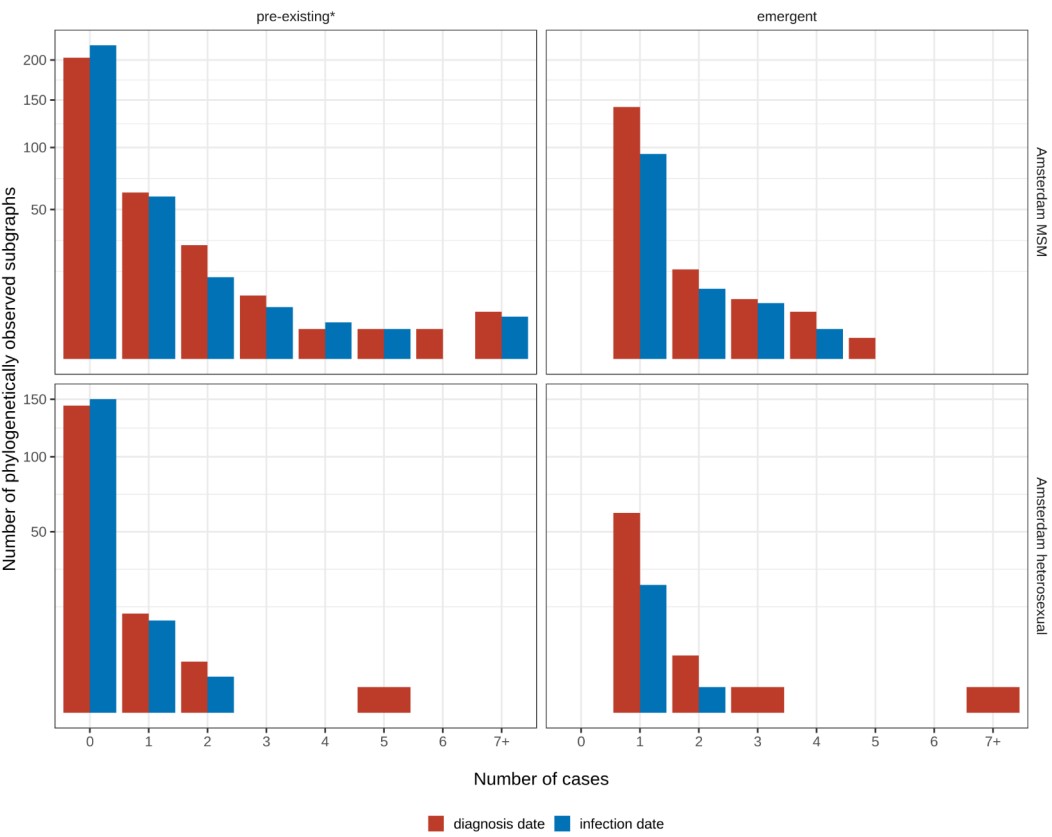

**Appendix 1—figure 44.** Growth of pre-existing and emergent phylogenetically observed subgraph sizes using diagnosis date and estimated date of infection. * pre-existing prior to 2014.

