## [Editor Report]

Congratulations on this impressive paper which combines clinical biomarker data, patient specific data and viral genetics data to estimate the proportion of HIV infections occurring within key subgroups of the population in Amsterdam. The work is methodologically impressive and also may be of high utility for understanding the spread of HIV and other viral infections through the population.

---

## [Decision Letter]

**Decision letter after peer review:**

Thank you for submitting your article "Estimating the potential to prevent locally acquired HIV infections in a UNAIDS Fast–Track City, Amsterdam" for consideration by *eLife*. Your article has been reviewed by 3 peer reviewers, and the evaluation has been overseen by a Reviewing Editor and David Serwadda as the Senior Editor. The following individual involved in review of your submission has agreed to reveal their identity: David A Rasmussen (Reviewer #2).

Essential revisions:

Please clarify certain assumptions about the model regarding the source of local infections, as described mostly by reviewer 2 and a bit by the other 2 reviewers

*Reviewer #1 (Recommendations for the authors):*

This manuscript describes an analysis of the transmission dynamics of HIV–1 in Amsterdam in the context of a rapid scale–up of prevention measures. Rather than relying on a single source of information, it integrates both virus sequence diversity (by phylogenetic analysis) and statistical modelling to derive estimates of the numbers of unsampled infections and to use clinical biomarkers to estimate dates of infection.

Specifically, the authors reconstructed phylogenetic trees relating large numbers of HIV–1 sequences collected in Amsterdam, outside Amsterdam, and from around the world. The distribution of risk groups and locations was extrapolated by maximum parsimony from the observed sequences at the tips of the tree down towards their common ancestors, and clusters associated with local transmission within Amsterdam were extracted. This process was repeated for 100 replicate trees, which is an important step for accommodating the uncertainty in reconstructing phylogenetic trees.

The authors used a Bayesian method to fit a hierarchical model of the time elapsed between HIV infection and diagnosis ("age" of infection), as predicted by viral load and CD4 measurements, and stratify the model by risk group and location. The resulting estimated dates of infection provide an important context to interpreting the transmission clusters extracted from the phylogenetic trees – many studies have endeavoured to obtain similar information.

The presence of undiagnosed infections that are epidemiologically related to clusters in the study was estimated by fitting a branching process model to the clusters augmented with estimated infection dates. This is a very noteworthy aspect of the analysis, especially because the failure to observe undiagnosed infections is a significant issue that makes it difficult to interpret HIV–1 clusters derived from the comparative analysis of sequence variation. In this case, clusters of recently diagnosed individuals who were infected for a long period of time, as indicated by clinical biomarkers, implies a larger number of unsampled infections related to that particular group.

The authors report that, despite the success of scaling up prevention measures in Amsterdam at limiting the onward transmission of HIV–1, there remains a substantial undiagnosed population associated with a disproportionately greater number of recent infections, and that this largely affects foreign–born men who have sex with men. These methodological advances and unique data resources distinguish this study from an abundant literature on HIV–1 transmission clusters. Most of my comments, provided as recommendations for the authors, about the manuscript concern language and the presentation of results.

– Page 2, "(HIV) is concentrated in metropolitan areas, […] representing 26% of the global HIV burden" – minor point, it would be easier to interpret this statistic if it were accompanied with a rough estimate of what share of the global population is contained in metropolitan areas

– Page 2, is there a URL or reference for the UNAIDS fast–track Cities initiative?

– Page 3, "among infections that are estimated to have occurred since 2014" – is the significance of this cutoff year associated with the fast track initiative? It would be helpful to spell this out for the reader.

– Page 4, I have heard of the Weibull distribution but not the Weibull likelihood – please clarify.

– Page 4, "we used the first available partial HIV–1 polymerase (pol) sequence" – please provide HXB2 nucleotide coordinates

– Page 4, "retrieved from the Los Alamos HIV–1 sequence database" – the LANL database requests authors to provide the URL (https://www.hiv.lanl.gov)

– Page 4, "All sequences were subtyped using Comet and Rega." Please provide version numbers and briefly explain how discordant subtype assignments were resolved between these two algorithms. (I found the explanation in the supplementary material and it's not that much longer – why not explain it in main text?)

– Page 4, did you use a double–precision version of FastTree?

– Page 4, I have a problem with the term "phylogenetic subgraphs". A subgraph is a subset of nodes in the network that is not necessarily a single connected component, whereas the authors are presumably referring to either subtrees (all descendants from an internal node), or subset trees (a contiguous subset of descendants from an internal node). In the HIV molecular epidemiology literature, we would refer to these as subtrees.

– Page 4, "Diagnosed Amsterdam patients in the same subgraph are interpreted as belonging to the same transmission chain" – this interpretation is not very accurate – while individuals in a distinct subtree are more epidemiologically related, they are not necessarily in the same "transmission chain" and it is misleading to make this association. The conventional terminology would be "transmission cluster".

– Page 4, "the estimated state of the root of the subgraph is interpreted as the origin of the transmission chain" – this ancestral state reconstruction is going to be quite uncertain in many cases – was this uncertainty propagated from the replicate bootstrap trees? – it would also be good to emphasize that "origin" refers to location and risk group, and not a single individual

– Figure 1, please check this figure for color accessibility – also the use of thin (low weight) font and narrow lines in the tree may make the figure difficult to read for the vision impaired

– Figure 3, I don't understand why there is so much unused space at the top half of the lower plots – I surmise that the authors intend to preserve the y–axis scales between the top and bottom plots for each of A and B columns, but why not truncate the lower plots? I also think the "zoom in" plots have limited utility and could be moved to supplementary material. Otherwise the plots are a rather effective visualization method.

– Page 12, please provide some details on the branching process model, since it plays an important role in this analysis – although it is well described in supplementary material, it would be helpful to give a little more info in the main text

– Figure 4, for what it's worth, these palettes render well in black and white

– Page 15, "the proportion of undiagnosed individual individuals infected with HIV are" – singular or plural?

– Code availability – this GitHub repository appears to be set to private – please make this available for peer review

– Figures S4–S14 – I'm not a fan of radial layouts for displaying phylogenies with node annotations – also, are these trees rooted? this layout method implies that they are, but I did not see any methodological description for rooting trees. Also, these appear to be vectorized images – it's actually more efficient to use a rasterized format like PNG or TIFF for large annotated trees because you're otherwise storing information about an excessive number of objects to draw.

– Figure S4, what's circulating recombinant form A1? do you mean subtype A1?

– Equation S12, please use or something similar to prevent the MSM and HSX superscripts from being rendered in math mode

– Please explain how the pairs plots (Figures S18, S19) were used to assess convergence of the Hamiltonian Monte Carlo chain samples. The figure legends don't provide any information.

*Reviewer #2 (Recommendations for the authors):*

While Amsterdam has done a tremendous job identifying and treating HIV infections and a large proportion of the HIV infected population is in care, the authors results highlight that there are often still substantial delays in diagnosis and that as many as 20% of individuals infected between 2014–2018 remained undiagnosed by the end of 2018. These results have important implications for HIV prevention strategies and policy: long–term goals aimed at diagnosing or treating a certain fraction of the overall infected population may need to be reconsidered in a more dynamic context focused on the delays between infection and diagnosis/treatment.

Overall, the study design is strong and the statistical methods used to estimate the time to diagnosis and the proportion of undiagnosed infections are sound.

However, it's not clear to me what assumptions are being made to estimate the proportion of new infections arising locally from individuals living in Amsterdam. It seems there may be several latent assumptions here, namely (1) there are no unobserved transmission events in the transmission chains between individuals living in/outside of Amsterdam (2) sampling more extensively outside of Amsterdam would not break apart the observed transmission chains into a larger number of smaller chains (with more external introductions from outside of Amsterdam). It seems to me these assumptions could bias the estimated proportion of new infections arising locally, and therefore the "preventable fraction", upwards. At the very least I think it would be helpful to be more transparent about the assumptions behind the estimator given in eq. 2.

Pg. 5 "predicted actual transmission chains" – not clear what is meant by this.

Pg. 6 "N_C is the estimated number of transmission chains which emerged between 2014–2018" –– is N_C the estimated number of transmission chains or rather the estimated number of individuals in those transmission chains? It seems like it would need to be the later to make N_C directly comparable to N_I.

*Reviewer #3 (Recommendations for the authors):*

The paper uses various sources of data to identify when and among whom new HIV transmissions are occurring in Amsterdam. The models use multiple data inputs and the conclusions are of high public health importance. The results appear to be largely supported by the analyses. The multidisciplinary approach could be of use for tracking infectious diseases in other settings.

A few areas of the paper could use clarification as described below

– I am a bit confused by the term upwards bias in time to diagnosis estimates. Please be specific in what is meant by this phrase.

– The data in the 2nd and 3rd to last columns in Table 1 would be easier to digest in a histogram.

– Tables 2 and 3 would also be easier to digest in graphical form.

– Please provide 1–2 sentences summarizing the Bayesian methods in Pantazis et al. in the Methods.

– Please justify the assumption that time to diagnosis did not change substantially over 2010–2019. It seems like that with lower true incidence, in theory a higher proportion of diagnoses could occur during later stages of infection.

– Please more precisely define "uninterrupted state label".

– Why were other subtypes or recombinants excluded?

– The blue and turquoise in Figure 3 are difficult to discriminate.

– Discussion: What is the ECDC model? Please specify.

---

## [Author Response]

Essential revisions:Please clarify certain assumptions about the model regarding the source of local infections, as described mostly by reviewer 2 and a bit by the other 2 reviewers

We have been very much encouraged by the overall positive feedback on our work from the editorial board and the reviewers. Thank you.

With regards to the essential revisions noted above, we carefully updated the entire manuscript as best as possible to describe our definitions, assumptions, and implications including possible caveats.

We further met one more time with the wider H-Team in Amsterdam (the PIs, physicians, patient representatives and city authorities involved in the HIV transmission elimination team Amsterdam) to clarify language around the notion of “Amsterdam transmission chains”, as we felt this is especially important in the wider context of concerns around using HIV phylogenetics methods.

We now define the term Amsterdam transmission chains more clearly in the Introduction,

“Here, we build on Amsterdam’s combined case and genomic surveillance data to reconstruct transmission chains at city level, defined as a single introduction of HIV into Amsterdam residents, followed by a direct infection chain among Amsterdam residents (Figure 1).”

We clarified the nature of our location data in the Methods section,

“We geolocated diagnosed infections to Amsterdam based on patients’ postcode of residence at time of first registration in ATHENA or the most recent registration update, which includes PLHIV that changed residence to Amsterdam prior to first registration and PLHIV that changed residence to another Dutch municipality after first registration.”

We further clarified on the exact search criteria to define the background sequence data set, which is crucial to be able to separate infected Amsterdam residents into distinct phylogenetic clades (or subgraphs using the preferred terminology associated with the phyloscanner software). We highlight that this was an iterative analysis process, involving manual inspection of identified Amsterdam clades with in-country experts to detect unlikely large clades and refine the search criteria,

“To reconstruct distinct HIV transmission chains among Amsterdam residents, we used the first available partial HIV-1 *polymerase* (*pol*) sequence from Amsterdam PLHIV, Dutch PLHIV from outside Amsterdam, and ~82,000 *pol* sequences from non-Dutch PLHIV. The non-Dutch viral sequences were retrieved from the Los Alamos HIV-1 sequence database subject to a length of at least 1300 in the *pol* gene on March 2, 2020 (www.hiv.lanl.gov). The basic local alignment search tool (BLAST v2.10.0) was used to select the top 20 closest background sequences to any Dutch sequence (Altschul et al. 1990).”

We further clarified our assumptions around our epidemiological interpretation of the observed phylogenetic data,

“In the labelled phylogeny, the lineage labels jump backwards in time, e.g. from Amsterdam MSM associated with a lineage ending in a tip observed in Amsterdam MSM to Western Europe. Thus, we can group lineages according to the same label between jumps, and we follow Wymant et al. in referring to these groups as *phyloscanner* subgraphs (Wymant et al. 2018). We assumed that we have sufficient background sequences such that no additional background sequences would further separate transmission chains among Amsterdam residents into more distinct chains. A subtle but important related point is that with the available location data at time of registration or a registration update, we are only able to phylogenetically reconstruct transmission chains by residence status rather than the location at which transmission actually occurred. For example, two Amsterdam residents appear in the same *phyloscanner* subgraph if they infected each other during a short-term visit in another Dutch, European or global location, if they were both infected from a common source during such a short-term visit and the source remained unsampled, if they infected each other before they began their residence in Amsterdam, or after they moved to another Dutch municipality. Diagnosed Amsterdam patients in the same subgraph were then interpreted as belonging to the same transmission chain, and the estimated state of the root of the subgraph was interpreted as the geographical origin of the transmission chain. Throughout, we refer to the subgraphs also as the phylogenetically observed (parts of) transmission chains. Using this approach, we note that unlike most phylogenetic clustering analyses (Brenner et al. 2017), every infected patient with a sequence is included in one subgraph, and all partially observed transmission chains of size one are included in the analysis to ensure that the entire distribution of observed transmission chains is represented in the analysis (Bezemer et al. 2022). To capture phylogenetic uncertainty, phylogenetic analyses were repeated on 100 bootstrap replicates drawn from each subtype alignment, and transmission chains were enumerated across these replicate analyses.”

We cut shorter descriptions of the directly observed data in the Results, section Growth of the phylogenetically observed parts of city-level transmission chains,

“The emerging chains thus outnumbered the growing pre-existing chains in both Amsterdam MSM and heterosexuals. However, the observed phylogenetic data are challenging to interpret directly because larger proportions of recent infections remain undiagnosed, approximately half of diagnosed individuals did not have a sequence sampled, and small chains are more likely to remain entirely unobserved (see Materials and methods).”

We more clearly describe the term ‘locally preventable infections’,

“From the emerging transmission chains, we can directly estimate the proportion of Amsterdam infections since 2014 that had an Amsterdam source (see Materials and methods). We interpret these infections as locally preventable, because they are within the reach of the HIV prevention efforts in Amsterdam.”

We added clarifications on the interpretability of the reconstructed transmission chains among Amsterdam residents at the start of the Discussion,

“More than 300 cities have by the end of 2021 signed the Fast-Track Cities Paris Declaration and committed to end the AIDS epidemic by 2030, addressing disparities in access to basic health and social services, social justice and economic opportunities. The city of Amsterdam reached the UNAIDS Fast-track Cities 95-95-95 targets before the onset of the COVID-19 pandemic, and has seen a decade of declines in city-level HIV diagnoses. Here, we characterised the number, size and growth of HIV transmission chains among Amsterdam residents, and quantified the further potential of preventing HIV infection at city level. It is important to recognize that through the analyses conducted here, the exact location of infection events cannot be identified. Rather, the available location data enable us to identify groups of Amsterdam residents with phylogenetically distinct HIV, which are the inferential basis for estimating the number, size, and growth of the actual unobserved transmission chains among Amsterdam residents. Regardless of the exact infection location, Amsterdam residents live in Amsterdam, and are thus within reach of Amsterdam public health and local prevention interventions.”

We further clarified the strength of evidence of our primary findings in the Discussion,

“Fourth, we quantified the locally preventable infections among Amsterdam residents in 2014-2018, defined as the infections in Amsterdam residents in 2014-2018 who are estimated to have as source another Amsterdam resident. Using the virus’ genetic code as an objective marker into infection events, we estimate that regardless of declining diagnoses and incidence, the majority of infections in Amsterdam residents in 2014-2018 remained locally preventable in all risk groups investigated. The statistical strength of evidence into this finding was strong for Amsterdam MSM (all 95% credible intervals for the proportion of locally preventable infections were above 50%), but more moderate for Amsterdam heterosexuals (wider credible intervals including 50%), reflecting that relatively few infections in Amsterdam heterosexuals in 2014-2018 were observed with a viral sequence by early 2019 due to frequent late diagnosis and incomplete viral sequencing.”

Reviewer #1 (Recommendations for the authors):This manuscript describes an analysis of the transmission dynamics of HIV–1 in Amsterdam in the context of a rapid scale–up of prevention measures. Rather than relying on a single source of information, it integrates both virus sequence diversity (by phylogenetic analysis) and statistical modelling to derive estimates of the numbers of unsampled infections and to use clinical biomarkers to estimate dates of infection.Specifically, the authors reconstructed phylogenetic trees relating large numbers of HIV–1 sequences collected in Amsterdam, outside Amsterdam, and from around the world. The distribution of risk groups and locations was extrapolated by maximum parsimony from the observed sequences at the tips of the tree down towards their common ancestors, and clusters associated with local transmission within Amsterdam were extracted. This process was repeated for 100 replicate trees, which is an important step for accommodating the uncertainty in reconstructing phylogenetic trees.The authors used a Bayesian method to fit a hierarchical model of the time elapsed between HIV infection and diagnosis ("age" of infection), as predicted by viral load and CD4 measurements, and stratify the model by risk group and location. The resulting estimated dates of infection provide an important context to interpreting the transmission clusters extracted from the phylogenetic trees – many studies have endeavoured to obtain similar information.The presence of undiagnosed infections that are epidemiologically related to clusters in the study was estimated by fitting a branching process model to the clusters augmented with estimated infection dates. This is a very noteworthy aspect of the analysis, especially because the failure to observe undiagnosed infections is a significant issue that makes it difficult to interpret HIV–1 clusters derived from the comparative analysis of sequence variation. In this case, clusters of recently diagnosed individuals who were infected for a long period of time, as indicated by clinical biomarkers, implies a larger number of unsampled infections related to that particular group.The authors report that, despite the success of scaling up prevention measures in Amsterdam at limiting the onward transmission of HIV–1, there remains a substantial undiagnosed population associated with a disproportionately greater number of recent infections, and that this largely affects foreign–born men who have sex with men. These methodological advances and unique data resources distinguish this study from an abundant literature on HIV–1 transmission clusters. Most of my comments, provided as recommendations for the authors, about the manuscript concern language and the presentation of results.

Thank you for this positive evaluation.

– Page 2, "(HIV) is concentrated in metropolitan areas, […] representing 26% of the global HIV burden" – minor point, it would be easier to interpret this statistic if it were accompanied with a rough estimate of what share of the global population is contained in metropolitan areas

Thank you for this comment. We have reviewed these estimates in the main paper as stated in the UNAIDS report, and found the actual basis of these estimates to be vague with little information on how they were generated. For this reason, we have decided to remove the quantitative part of this statement, and leave only the qualitative sub-sentence in the introduction. Specifically, we have changed the text in the Introduction as follows:

“Human immunodeficiency virus (HIV) is concentrated in metropolitan areas (Joint United Nations Programme on HIV/AIDS (UNAIDS) 2014).”

– Page 2, is there a URL or reference for the UNAIDS fast–track Cities initiative?

Thank you for highlighting this, we have added a reference to the UNAIDS Fast-Track cities website where it is first mentioned,

“In response, as of March 2021 over 300 cities have joined the Fast-Track Cities initiative (www.fast-trackcities.org) by signing the Paris Declaration, committing to end the AIDS epidemic by 2030, by addressing disparities in access to basic health and social services, social justice and economic opportunities (UNAIDS 2019).”

– Page 3, "among infections that are estimated to have occurred since 2014" – is the significance of this cutoff year associated with the fast track initiative? It would be helpful to spell this out for the reader.

Thank you for this comment. 2014 was the year Amsterdam joined the Fast-Track Cities network and the H-Team was founded. We have amended the text as follows:

“…among infections that are estimated to have occurred since Amsterdam joined the Fast-Track Cities network in 2014 and galvanised its local response with the inception of the H-Team on December 1, 2014.”

– Page 4, I have heard of the Weibull distribution but not the Weibull likelihood – please clarify.

Thanks for raising this. We have revised the text to clarify that we fit a model in which the times to diagnosis follow a Weibull distribution as follows:

“We next reconstructed characteristic time-to-diagnosis distributions for each of the 9 Amsterdam risk groups (MSM/heterosexual, and region of birth) with a Bayesian hierarchical model from the individual-level estimates, modelling the individual-level estimates with a Weibull distribution.”

– Page 4, "we used the first available partial HIV–1 polymerase (pol) sequence" – please provide HXB2 nucleotide coordinates

We have added the coordinates and referenced the HXB2 genome as follows in the Methods section, sub-section ‘Phylogenetic reconstruction of city-level transmission chains’:

“Subtype-specific alignments were generated with *Virulign (Libin et al. 2019)* (Appendix 1 Section 4.1) and sequences from other subtypes were added as outgroup for the purpose of phylogenetic rooting. The final alignments were trimmed to positions 2253-3870 in the reference genome HXB2 (Ratner et al. 1985).”

– Page 4, "retrieved from the Los Alamos HIV–1 sequence database" – the LANL database requests authors to provide the URL (https://www.hiv.lanl.gov)

Thank you for pointing this out. We have added in a reference to the URL:

“The non-Dutch viral sequences were retrieved from the Los Alamos HIV-1 sequence database subject to a length of at least 1300 in the pol gene on March 2, 2020 (www.hiv.lanl.gov).”

– Page 4, "All sequences were subtyped using Comet and Rega." Please provide version numbers and briefly explain how discordant subtype assignments were resolved between these two algorithms. (I found the explanation in the supplementary material and it's not that much longer – why not explain it in main text?)

Thank you for highlighting this. We have added version numbers and clarified how Rega was used to verify uncertain subtypes from Comet as follows:

“All sequences were subtyped using Comet v2.3 (Struck et al. 2014). Sequences with an uncertain subtype classification using Comet were analysed with Rega v3.0 (Pineda-Peña et al. 2013). Any remaining sequences for which a subtype could not be resolved were discarded from further analysis (n = 122).”

– Page 4, did you use a double–precision version of FastTree?

We did not use a double-precision version, and have noted this for future work. Thank you for pointing this out.

– Page 4, I have a problem with the term "phylogenetic subgraphs". A subgraph is a subset of nodes in the network that is not necessarily a single connected component, whereas the authors are presumably referring to either subtrees (all descendants from an internal node), or subset trees (a contiguous subset of descendants from an internal node). In the HIV molecular epidemiology literature, we would refer to these as subtrees.

Thanks for this query. We refer to subgraphs in the context of the *phyloscanner* software, and obtained the following clarifications from the lead authors: “Here, host subgraphs are the result of ancestral-host state reconstruction, defined as connected regions of the tree that are assigned the same host state. The background subgraphs do not necessarily include every descendant of a node, so we choose not to use the term ‘subtree’, which has a very well understood meaning in phylogenetics as you have defined it.”

To make the distinction clearer and honour this recommendation, the first time we refer to these subgraphs, we have corrected “phylogenetic subgraphs” to say “*phyloscanner* subgraphs”, as follows:

“In the labelled phylogeny, the lineage labels jump backwards in time, e.g. from Amsterdam MSM associated with a lineage ending in a tip observed in Amsterdam MSM to Western Europe. Thus, we can group lineages according to the same label between jumps, and we follow (Wymant et al. 2018) in referring to these groups as *phyloscanner* subgraphs.”

– Page 4, "Diagnosed Amsterdam patients in the same subgraph are interpreted as belonging to the same transmission chain" – this interpretation is not very accurate – while individuals in a distinct subtree are more epidemiologically related, they are not necessarily in the same "transmission chain" and it is misleading to make this association. The conventional terminology would be "transmission cluster".

Thank you for this comment. We have discussed this point at length amongst the writing team, and the wider H-Team in Amsterdam (the PIs, physicians, patient representatives and city authorities involved in the HIV transmission elimination team Amsterdam) to identify the best language to use. Ultimately, we decided against the word “transmission cluster” due to its similarity with the terms “phylogenetic cluster” and “phylogenetic clustering methods” that are ubiquitous in HIV phylogenetics, and which we have not adopted here.

Yet, your comment touches upon several important points and concepts, which we have modified throughout. We now define the term Amsterdam transmission chains more clearly in the Introduction,

“Here, we build on Amsterdam’s combined case and genomic surveillance data to reconstruct transmission chains at city level, defined as a single introduction of HIV into Amsterdam residents, followed by a direct infection chain among Amsterdam residents (Figure 1).”

We clarified the nature of our location data in the Methods section,

“We geolocated diagnosed infections to Amsterdam based on patients’ postcode of residence at time of first registration in ATHENA or the most recent registration update, which includes PLHIV that changed residence to Amsterdam prior to first registration and PLHIV that changed residence to another Dutch municipality after first registration.”

We further clarified our assumptions around our epidemiological interpretation of the observed phylogenetic data,

“In the labelled phylogeny, the lineage labels jump backwards in time, e.g. from Amsterdam MSM associated with a lineage ending in a tip observed in Amsterdam MSM to Western Europe. Thus, we can group lineages according to the same label between jumps, and we follow Wymant et al. in referring to these groups as *phyloscanner* subgraphs (Wymant et al. 2018). We assumed that we have sufficient background sequences such that no additional background sequences would further separate transmission chains among Amsterdam residents into more distinct chains. A subtle but important related point is that with the available location data at time of registration or a registration update, we are only able to phylogenetically reconstruct transmission chains by residence status rather than the location at which transmission actually occurred. For example, two Amsterdam residents appear in the same *phyloscanner* subgraph if they infected each other during a short-term visit in another Dutch, European or global location, if they were both infected from a common source during such a short-term visit and the source remained unsampled, if they infected each other before they began their residence in Amsterdam, or after they moved to another Dutch municipality. Diagnosed Amsterdam patients in the same subgraph were then interpreted as belonging to the same transmission chain, and the estimated state of the root of the subgraph was interpreted as the geographical origin of the transmission chain. Throughout, we refer to the subgraphs also as the phylogenetically observed (parts of) transmission chains. Throughout, we refer to the subgraphs also as the phylogenetically observed (parts of) transmission chains. Using this approach, we note that unlike most phylogenetic clustering analyses (Brenner et al. 2017), every infected patient with a sequence is included in one subgraph, and all partially observed transmission chains of size one are included in the analysis to ensure that the entire distribution of observed transmission chains is represented in the analysis (Bezemer et al. 2022). To capture phylogenetic uncertainty, phylogenetic analyses were repeated on 100 bootstrap replicates drawn from each subtype alignment, and transmission chains were enumerated across these replicate analyses.”

We more clearly describe the term ‘locally preventable infections’,

“From the emerging transmission chains, we can directly estimate the proportion of Amsterdam infections since 2014 that had an Amsterdam source (see Materials and methods). We interpret these infections as locally preventable, because they are within the reach of the HIV prevention efforts in Amsterdam.”

– Page 4, "the estimated state of the root of the subgraph is interpreted as the origin of the transmission chain" – this ancestral state reconstruction is going to be quite uncertain in many cases – was this uncertainty propagated from the replicate bootstrap trees? – it would also be good to emphasize that "origin" refers to location and risk group, and not a single individual

Thank you for this question. Indeed, we estimate the origins for the central trees and 100 replicate trees, generated from bootstrapped alignments, to capture phylogenetic uncertainty. We fully agree that there is uncertainty in exactly which of the location labels (the 11 non-Amsterdam locations, the Netherlands, Africa, Western Europe, Eastern Europe and Central Asia, North America, Latin America and the Caribbean, Dutch Caribbean and Suriname, Middle East and North Africa, and South and South-East Asia and Oceania). However please note that for our purposes and the calculation of the locally acquired infections, it is only relevant whether a transmission chain originated within or outside of Amsterdam, and this is associated with less phylogenetic uncertainty. Specifically, we find good agreement in Ams/non-Ams origin allocations across all 100 bootstrap replicate trees: well over 80% of the Amsterdam phyloscanner subgraphs have the Ams/non-Ams origin allocation (see Author response image 1). Please note that the tips in each subgraph may change across bootstrap trees, so we here mapped subgraphs by identifying the subgraph with the most common tips shared with each subgraph from the central tree. Note also that this uncertainty is reflected in our calculations, specifically equation S56 in Appendix 1 Section S7.2.

We have also clarified that “origin” refers to the geographic origin of the transmission chain in the Methods section:

“Diagnosed Amsterdam patients in the same subgraph are interpreted as belonging to the same transmission chain, and the estimated state of the root of the subgraph is interpreted as the geographical origin of the transmission chain”

**Author response image 1. sa2fig1:** 

– Figure 1, please check this figure for color accessibility – also the use of thin (low weight) font and narrow lines in the tree may make the figure difficult to read for the vision impaired

Thank you for highlighting this. We have modified the figure to improve colour accessibility, increased font size and revised the tree subfigure to improve legibility.

– Figure 3, I don't understand why there is so much unused space at the top half of the lower plots – I surmise that the authors intend to preserve the y–axis scales between the top and bottom plots for each of A and B columns, but why not truncate the lower plots? I also think the "zoom in" plots have limited utility and could be moved to supplementary material. Otherwise the plots are a rather effective visualization method.

Thank you for pointing this out. We have reduced the white space in the sub-figure of subgraphs among heterosexuals. Since it is important to see both the transmission chains which did not grow further since 2014 (panel A) and also the detail of the transmission chains which grew since 2014 (panel B), we have added some annotation to indicate that these are ‘zoomed in’ from panel A.

– Page 12, please provide some details on the branching process model, since it plays an important role in this analysis – although it is well described in supplementary material, it would be helpful to give a little more info in the main text

Thank you for raising this. We agree and have added more details in the model, including the probability densities for the actual chain sizes and observed chain sizes as follows:

“Specifically, given m=1,…,M index cases of a chain that pre-existed, the final size distribution of stuttering transmission chains is under a Negative Binomial branching process model given by

c(i|m,μ,ϕ) = mm+i NegBin

where NegBin is the Negative Binomial distribution characterised by mean μm and dispersion parameter ϕm, i=0,1,2,… is the number of new cases, and μ < 1. Incomplete sampling of new cases can be accommodated via, (3)c~obs(i|m,μ,ϕ,ρ)=∑k=1∞Bin(i|k,ρ)c(k|m,μ,ρ)=∑k=1∞Bin(i|k,ρ)mm+kNegBin(k|μm,ϕm)

where ρ denotes the probability that a new case in 2014-2018 is diagnosed and has a viral sequence sampled by database closure. In the model, the index cases are assumed to be infectious and defined by the number of unsuppressed members by 2014 in a pre-existing chain, adjusted for the sampling probability of such members. We further capped the infinite sum in (3) in the model, recognizing that the summands rapidly tend to zero. The corresponding equations for emergent transmission chains (since 2014 as defined above) is similar,. (6)c~obs(n|m=1,μ,ϕ,ρ)=∑z=n∞Bin(n|zρ)1zNegBin(z−1|μ,ϕ)1−∑z=n∞((1−ρ)z1zNegBin(z−1|μ,ϕ))

The likelihood then comprises the growth distributions of emerging chains, pre-existing chains that continued to grow, and pre-existing chains with unsuppressed members that did not grow, with the following log-likelihood,, (7)l(x,x~|μ,ϕ,ρ)=∑m=1M∑i=0Ixmilogcobs(i|m,μ,ϕ,ρ)+∑n=1Nx~nlog˜cobs(n|m=1,μ,ϕ,ρ).

where M is the largest number of index cases observed across the chains after adjusting for sampling, I is the largest number of new cases observed in pre-existing chains and N is the largest number of new cases observed in emergent chains, including the first case. Pre-existing chains for which all members were suppressed by 2014 and which did not grow were not included, because these chains had no unsuppressed index case. We fitted the branching process model under a Bayesian framework with Stan version 2.21 to MSM chains, borrowing information across subtypes, and similarly for heterosexual chains.”

– Figure 4, for what it's worth, these palettes render well in black and white

Thank you for the feedback.

– Page 15, "the proportion of undiagnosed individual individuals infected with HIV are" – singular or plural?

Thank you for highlighting this. We have amended the sentence to reflect that there are several proportions, by place of birth, as follows:

“First, when focusing on the denominator of recent infections that are estimated to have occurred in 2014-2018, the proportions of undiagnosed individuals infected with HIV are all between 9-20% in (self-identified) Amsterdam MSM risk groups, and between 28-57% in Amsterdam heterosexual risk groups.”

– Code availability – this GitHub repository appears to be set to private – please make this available for peer review

Thanks for raising this. The repository has now been made public at github.com/alexblenkinsop/locally.acquired.infections.

– Figures S4–S14 – I'm not a fan of radial layouts for displaying phylogenies with node annotations – also, are these trees rooted? this layout method implies that they are, but I did not see any methodological description for rooting trees. Also, these appear to be vectorized images – it's actually more efficient to use a rasterized format like PNG or TIFF for large annotated trees because you're otherwise storing information about an excessive number of objects to draw.

Thank you for raising this. We have added some further details on how the trees were rooted to the methods section as follows:

“Subtype-specific alignments were generated with *Virulign (Libin et al. 2019)* (Appendix 1 Section 4.1) and sequences from other subtypes were added as outgroup for the purpose of phylogenetic rooting. The final alignments were trimmed to positions 2253-3870 in the reference genome HXB2 (Ratner et al. 1985).

Subtype-specific HIV phylogenetic trees were generated for alignments with at least 50 Amsterdam sequences (subtypes and recombinant forms B, 01AE, 02AG, C, D, G, A1 or 06cpx) using FastTree v2.1.8 (Price, Dehal, and Arkin 2010). Following tree reconstruction, trees were manually rooted at the outgroup, and the outgroup taxa were then pruned from the phylogeny. Next, we attributed”

We have now also plotted the inferred, rooted trees in standard layout, and added these due to size as Supplementary Figures to the manuscript (Appendix 1 – Figures 4-25).

– Figure S4, what's circulating recombinant form A1? do you mean subtype A1?

Thank you for spotting this. We have amended the legend to say subtype A1.

– Equation S12, please use or something similar to prevent the MSM and HSX superscripts from being rendered in math mode

Thank you for highlighting this. We have amended the risk groups to not be formatted in math mode.

– Please explain how the pairs plots (Figures S18, S19) were used to assess convergence of the Hamiltonian Monte Carlo chain samples. The figure legends don't provide any information.

Thank you for raising this. For both the models for the undiagnosed and the branching process model we have added trace plots and reported the parameter with the smallest effective sample size as follows.

In section S3.3.2 of Appendix 1:

“The models mixed well, with no correlation between parameters, and had at most one divergence (Figures 30-33). The smallest effective sample size across parameters for the MSM model was 1562 and 1326 for the heterosexual model.”

In section S6.3 of Appendix 1:

“The models mixed well; Figures 40-41 show the trace plot for the parameter with the smallest effective sample size in each model, which was 2377 for the MSM model and 1125 for the heterosexual model. Figures 42-43 shows the pairs plot of parameters for the MSM and heterosexual models, respectively. ”

Reviewer #2 (Recommendations for the authors):While Amsterdam has done a tremendous job identifying and treating HIV infections and a large proportion of the HIV infected population is in care, the authors results highlight that there are often still substantial delays in diagnosis and that as many as 20% of individuals infected between 2014–2018 remained undiagnosed by the end of 2018. These results have important implications for HIV prevention strategies and policy: long–term goals aimed at diagnosing or treating a certain fraction of the overall infected population may need to be reconsidered in a more dynamic context focused on the delays between infection and diagnosis/treatment.Overall, the study design is strong and the statistical methods used to estimate the time to diagnosis and the proportion of undiagnosed infections are sound.However, it's not clear to me what assumptions are being made to estimate the proportion of new infections arising locally from individuals living in Amsterdam. It seems there may be several latent assumptions here, namely (1) there are no unobserved transmission events in the transmission chains between individuals living in/outside of Amsterdam (2) sampling more extensively outside of Amsterdam would not break apart the observed transmission chains into a larger number of smaller chains (with more external introductions from outside of Amsterdam). It seems to me these assumptions could bias the estimated proportion of new infections arising locally, and therefore the "preventable fraction", upwards. At the very least I think it would be helpful to be more transparent about the assumptions behind the estimator given in eq. 2.

Thank you for this positive and indeed very helpful public review. Please note our response to the editor at the beginning of this document for a detailed summary on how we clarified definitions, assumptions, and the interpretation of our findings.

Pg. 5 "predicted actual transmission chains" – not clear what is meant by this.

Thank you for raising this point. We agree. We have clarified the text as follows:

“Given estimates of the number and growth of both pre-existing and emergent transmission chains, it is straightforward to derive estimates of the proportion of HIV infections among Amsterdam residents in 2014-2018 that had an Amsterdam resident as source (which we denote by γ and refer to as the proportion of locally acquired infections). This is because all infections originating from an individual living in Amsterdam had a local source, except the index cases in the emerging chains that were introduced from outside of Amsterdam.”

Pg. 6 "N_C is the estimated number of transmission chains which emerged between 2014–2018" –– is N_C the estimated number of transmission chains or rather the estimated number of individuals in those transmission chains? It seems like it would need to be the later to make N_C directly comparable to N_I.

Thank you for highlighting this. N_C is the number of transmission chains. By definition, each chain has one index case, and so it is comparable to N_I. We have revised our text to clarify our reasoning behind this key equation as follows

“Given estimates of the number and growth of both pre-existing and emergent transmission chains, it is straightforward to derive estimates of the proportion of HIV infections among Amsterdam residents in 2014-2018 that had an Amsterdam resident as source (which we denote by and refer to as the proportion of locally acquired infections). This is because all infections originating from an individual living in Amsterdam had a local source, except the index cases in the emerging chains that were introduced from outside of Amsterdam. We have
(8)γ=NI−αNcNI,

where NI is the estimated number of new infections between 2014-2018 in both pre-existing chains and emerging transmission chains including the index case, NC is the estimated number of transmission chains which emerged between 2014-2018 and α is the proportion of emergent transmission chains with an Amsterdam origin. Since each transmission chain has one index case, αNC is the estimated number of infections with non-Amsterdam origin, and NI−αNc is the estimated number of infections that had an Amsterdam resident as a source.”

Reviewer #3 (Recommendations for the authors):The paper uses various sources of data to identify when and among whom new HIV transmissions are occurring in Amsterdam. The models use multiple data inputs and the conclusions are of high public health importance. The results appear to be largely supported by the analyses. The multidisciplinary approach could be of use for tracking infectious diseases in other settings.A few areas of the paper could use clarification as described below– I am a bit confused by the term upwards bias in time to diagnosis estimates. Please be specific in what is meant by this phrase.

Thank you for raising this. We agree. We have clarified our language in the Results and Discussion section as follows:

In the Results section:

“While the bivariate model of biomarker data that underpins the individual-level time-to-diagnosis estimates has been validated (Pantazis et al. 2019), our estimates of the proportion of undiagnosed infections in 2014-2018 depend further on the trends in the number of infections in each year as shown in Equation 2. The main analysis is based on trends in HIV infections in MSM and heterosexuals that were estimated with the ECDC HIV Modelling Tool for Amsterdam. The ECDC estimates account for late diagnoses, but apply to MSM and heterosexuals across Amsterdam. In sensitivity analyses we used instead trends in directly observed Amsterdam diagnoses, which apply to each Amsterdam risk group but do not account for confounding due to late diagnoses. In the sensitivity analysis, we estimate that 14% [13-17%] of infections in Amsterdan MSM in 2014-2018 remained undiagnosed, and 34% [28-41%] in Amsterdam heterosexuals. Further details are presented in Appendix 1, Section 3.3-3.5.”

In the Discussion section:

“We explored the impact of assumptions on incidence trends to the undiagnosed estimates and found some sensitivities (Appendix 1, Section 3.3), although estimates were all very similar as long as the assumed incidence trends reflected available data**.** Further sensitivity analyses are reported in Appendix 1 Section 3.4-3.5. We also further validated the time-to-diagnosis estimates by comparing the estimated proportion of recent HIV infections (≤6 months) with those estimated in an independent study in Amsterdam using avidity assays (Slurink et al. 2021), and found them to be similar (Appendix 1 Figure 28). The main limitation of our biomarker approach is thus that at present we cannot account for time trends in time-to-diagnosis.”

– The data in the 2nd and 3rd to last columns in Table 1 would be easier to digest in a histogram.

Thank you for this comment. We had indeed considered multiple different versions of Table 1, with the aim to place the derived estimates in the last column into a logical broader context. At the same time we appreciate that the table is busy, and several columns could also be presented in a figure.

In our revision we have opted to keep the column “Estimated undiagnosed” in the table, so that readers see the logical steps from the observed infections to the estimated total infections, and have direct access the numerical values, which are of interest. Based on this comment we have moved the column “Estimated time to diagnosis” into the new Appendix 1 – Figure 3:

We have further taken the opportunity to clarify the column names of the table as follows:

– Tables 2 and 3 would also be easier to digest in graphical form.

Thank you for this comment. We created the corresponding figures based on your input, however upon discussion with all co-authors, the majority felt that in this instance, it is important to explicitly list the numbers in favour of showing the overall trend in a figure. We have left tables 2 and 3 untouched.

– Please provide 1–2 sentences summarizing the Bayesian methods in Pantazis et al. in the Methods.

This is a good point. We realise our description has been very brief. We expanded the text in the Methods section as follows:

“Using longitudinal viral load and CD4 count data and further demographic and clinical information, we estimated time from infection to diagnosis for all HIV diagnosed patients with a Bayesian approach (Pantazis et al. 2019). Briefly, data from the CASCADE collaboration on 19,788 observed HIV seroconverters were used to parameterize a bivariate normal linear model of the joint time evolution of HIV viral load and CD4 cell count decline since time of infection in the context of additional covariates (sex, region of origin, mode of infection, age at time of diagnosis). Then we used the trained model to estimate infection times from longitudinal biomarker data for Amsterdam patients, with an average of 4 viral load observations and 6 CD4 cell count observations per patient.”

– Please justify the assumption that time to diagnosis did not change substantially over 2010–2019. It seems like that with lower true incidence, in theory a higher proportion of diagnoses could occur during later stages of infection.

Thank you, this is a good point. We have addressed this in two parts.

1/ At present our methodological approach cannot be used to derive time trends in the individual-level time-to-diagnosis estimates. Separate published modelling efforts suggest that this is not a major caveat, and we have added corresponding references in the Methods section:

“We next reconstructed characteristic time-to-diagnosis distributions for each of the 9 Amsterdam risk groups (MSM/heterosexual, and region of birth) with a Bayesian hierarchical model from the individual-level estimates, modelling the individual-level estimates with a Weibull distribution. To avoid censoring of infection-to-diagnosis times, we focused analyses on the subset of infections in 2010-2012 which were diagnosed by May 1, 2019 since most infections in this window would have been diagnosed by the close of study, and assume time to diagnosis did not change substantially in 2010-2019 (Ard van Sighem et al. 2017; Ard van Sighem 2017).”

We call out this caveat in the Discussion as well:

“The main limitation of our biomarker approach is thus that at present we cannot account for time trends in time to diagnosis.”

2/ Regardless, even with an assumed time to diagnosis that is constant in 2014-2018 (importantly: separately for each pop strata), we find that among infections occurring in year y=2014, 2015, …, 2018, the proportion of infections that remains undiagnosed is increasing by year. This is a simple consequence of the fact that for each infection year cohort, we are progressively observing lower quantiles in the time-to-diagnosis distribution (please recall Figure 2). Now, to obtain an overall estimate of the undiagnosed infections, we need to weight the annualised estimates, and we realised based on your very helpful comment that we did not account for the declining incidence trends in our calculations. This had a substantive impact on our estimates – thank you again for sharing your insight here.

We now make this methodological step clear in the Methods section:

“We then calculated the proportion of infections in each year y=2014,…,2018 in each of the 9 Amsterdam risk groups that were not diagnosed by database closure (which we denote by δy) from the fitted model. To adjust for trends in incidence over time, the annual estimates were weighted by the estimated number of HIV infections in each year among Amsterdam MSM and heterosexual individuals without stratifiction by inmigrant status, according to the European Centre for Disease Control and Prevention (ECDC) HIV modelling tool for Amsterdam, version 1.3.0 (Stockholm: European Centre for Disease Prevention and Control 2017) through weights**,**, (1)ωy=NyInf−ECDC∑z∈YNzInf−ECDC

where y = 2014,…,2018 and NyInf−ECDC are the estimated total number of infections in year y in Amsterdam MSM or heterosexuals. We then estimated the overall estimate of the proportion of undiagnosed infections in 2014-2018, δ, by applying these weights to the yearly proportions through. (2)δ=∑y∈Yωyδy

Recognizing the limitations in applying weights that do not account for differences by place of birth, we used in sensitivity analyses as weights the observed trends in the number of annual HIV diagnoses in the corresponding Amsterdam risk group. ”

Considering incidence rates that decline at the estimated incidence rate from the ECDC modelling for the Dutch population, the weights are:

**Author response table 1. sa2table1:** 

Year	MSM	Heterosexuals
2014	0.26	0.19
2015	0.23	0.19
2016	0.20	0.20
2017	0.17	0.21
2018	0.13	0.22

Assuming incidence rates that decline as the observed diagnosis rates in Amsterdam, the weights are:

**Author response table 2. sa2table2:** 

	MSM	HSX		
year	origin	weight	origin	weight
2014	W.Europe, N.America, Oceania	0.28	Sub-Saharan Africa	0.25
2015		0.27		0.30
2016		0.19		0.20
2017		0.13		0.16
2018		0.13		0.09
2014	E and C. Europe	0.25	S. America and Caribbean	0.24
2015		0.25		0.16
2016		0.29		0.27
2017		0.12		0.19
2018		0.08		0.14
2014	S. America and Caribbean	0.20	NL	0.24
2015		0.20		0.24
2016		0.20		0.20
2017		0.25		0.14
2018		0.16		0.18
2014	NL	0.28	Other	0.20
2015		0.24		0.20
2016		0.20		0.35
2017		0.14		0.25
2018		0.15		0.00
2014	Other	0.23		
2015		0.15		
2016		0.23		
2017		0.21		
2018		0.19		

Table 7 (added to Appendix 1) summarises the estimates of the proportion of undiagnosed infections using the various approaches, which indicates that our estimates into the proportion of undiagnosed infections is relatively insensitive provided the weighting used in the calculations reflects the declines in incidence rates:

We have modified the Results section as follows:

“Local estimates of the continuum of care indicate that Amsterdam has surpassed the 95-95-95 targets, with an estimated 5% of all people in Amsterdam living with HIV that remained undiagnosed by the end of 2019 (A. van Sighem et al. 2020; UNAIDS 2019). Based on the time-to-diagnosis estimates in our cohort, we can focus here at the forefront of ongoing transmission chains and quantify the proportion of recent Amsterdam infections in 2014-2018 that remained undiagnosed by May 1, 2019. Figure 2 shows that the estimated undiagnosed proportions are considerably higher when we focus on infections acquired since 2014. Accounting for declining diagnosis and infection trends (see Methods), an estimated 14% [12-16%] of infections in Amsterdan MSM in 2014-2018 remained undiagnosed, and 41% [35-48%] in Amsterdam heterosexuals (Table 1). The highest proportion of undiagnosed Amsterdam infections in 2014-2018 are in heterosexuals born in Sub-Saharan Africa, with 57% [47-67%].

While the bivariate model of biomarker data that underpins the individual-level time-to-diagnosis estimates has been validated (Pantazis et al. 2019), our estimates of the proportion of undiagnosed infections in 2014-2018 depend further on the trends in the number of infections in each year as shown in Equation 2. The main analysis is based on trends in HIV infections in MSM and heterosexuals that were estimated with the ECDC HIV Modelling Tool for Amsterdam. The ECDC estimates account for late diagnoses, but apply to MSM and heterosexuals across Amsterdam. In sensitivity analyses we used instead trends in directly observed Amsterdam diagnoses, which apply to each Amsterdam risk group but do not account for confounding due to late diagnoses. In the sensitivity analysis, we estimate that 14% [13-17%] of infections in Amsterdan MSM in 2014-2018 remained undiagnosed, and 34% [28-41%] in Amsterdam heterosexuals. Further details are presented in Appendix 1, Section 3.3-3.5.”

We have also clarified the Discussion:

“We explored the impact of assumptions on incidence trends to the undiagnosed estimates and found some sensitivities (Appendix 1, Section 3.3), although estimates were all very similar as long as the assumed incidence trends reflected available data**.** We explored the impact of assumptions on incidence trends to the undiagnosed estimates and found some sensitivities (Appendix 1, Section 3.3), although estimates were all very similar as long as the assumed incidence trends reflected available data**.** Further sensitivity analyses are reported in Appendix 1 Section 3.4-3.5. We also further validated the time-to-diagnosis estimates by comparing the estimated proportion of recent HIV infections (≤6 months) with those estimated in an independent study in Amsterdam using avidity assays (Slurink et al. 2021), and found them to be similar (Appendix 1 Figure 28). The main limitation of our biomarker approach is thus that at present we cannot account for time trends in time-to-diagnosis.”

– Please more precisely define "uninterrupted state label".

Thank you for this comment. We decided to refer to this relatively minor detail through a citation, rather than explain it here once more, as follows:

“Next, we attributed to all viral lineages in the phylogenies a ‘state’ label that included information on the transmission risk group (MSM, heterosexual, other) and location with *phyloscanner* version 1.8.0 (Wymant et al. 2018); see (Bezemer et al. 2022) for details. Locations were classified into Amsterdam (for ATHENA patients with an Amsterdam postcode at time of registration or a registration update), the Netherlands (for other ATHENA patients), and the 10 world regions Africa, Western Europe, Eastern Europe and Central Asia, North America, Latin America and the Caribbean, Dutch Caribbean and Suriname, Middle East and North Africa, and South and South-East Asia and Oceania (for non-Dutch sequences). In the labelled phylogeny, the lineage labels jump backwards in time, e.g. from Amsterdam MSM associated with a lineage ending in a tip observed in Amsterdam MSM to Western Europe. Thus, we can group lineages according to the same label between jumps, and we follow (Wymant et al. 2018) in referring to these groups as *phyloscanner* subgraphs.”

– Why were other subtypes or recombinants excluded?

In principle, we can perform phylogeographic analyses and subgraph identification for trees of all identified sequences. However from an analytical point of view this becomes increasingly laboursome while the epidemiological scientific insights at population-level are diminishing as sample sizes are decreasing. Our Bayesian hierarchical branching process model can handle subgraph data from all subtypes, but information is increasingly borrowed from the subtypes that are heavily represented in the data. In particular, given we have >1000 taxa associated with subtype B, the subtype specific parameters in the model associated with a rare recombinant will essentially look like those for subtype B. While logical from a statistical point, such estimates are at best uninformative and at worst misleading to public health practitioners. We have thus decided to exclude subtypes or CRFs that are at population level represented by less than 50 sequences, and state that for these subtypes/CRFs, we are unable to provide estimates.

We have modified the text as follows. In the Methods section:

“Subtype-specific HIV phylogenetic trees were generated for alignments with at least 50 Amsterdam sequences using FastTree v2.1.8. For the excluded subtypes/ CRFs, sequence sample sizes were too low to characterise transmission chains and chain growth at population-level.”

And in the Results section:

“43 individuals were excluded from further analysis as their subtype identification was inconclusive, or they were associated with other subtypes or recombinant forms with fewer than 50 sequences in Amsterdam. ”

– The blue and turquoise in Figure 3 are difficult to discriminate.

Thank you for pointing this out. We have amended the colour scheme to improve colour accessibility.

– Discussion: What is the ECDC model? Please specify.

This is a good point, we clarified the text and added a reference as stated below.

“We then calculated the proportion of infections in each year y from 2014 to 2018 that were not diagnosed by database closure (which we denote by δy) from the fitted model. To adjust for trends in incidence over time, the annual estimates were weighted by the estimated number of HIV infections in each year among Amsterdam MSM and heterosexual individuals, according to the European Centre for Disease Control and Prevention (ECDC) HIV modelling tool for Amsterdam, version 1.3.0 (Stockholm: European Centre for Disease Prevention and Control 2017) through weights**,**, (1)ωy=NyInf−ECDC∑z∈YNzInf−ECDC

where y = 2014,…,2018 and NyInf−ECDC are the estimated total number of infections in year y in Amsterdam.”